# END-TO-END PROBABILISTIC FRAMEWORK FOR LEARNING WITH HARD CONSTRAINTS

**Utkarsh**[*]
MIT CSAIL
utkarsh5@mit.edu

**Danielle C. Maddix**[†]
AWS
dmmaddix@amazon.com

**Ruijun Ma**
Amazon SCOT
ruijunma@amazon.com

**Michael W. Mahoney**
Amazon SCOT
zmahmich@amazon.com

**Yuyang Wang**
AWS
yuyawang@amazon.com

## ABSTRACT

We present `ProbHardE2E`, a probabilistic forecasting framework that incorporates hard operational/physical constraints, and provides uncertainty quantification. Our methodology uses a novel differentiable probabilistic projection layer (DPPL) that can be combined with a wide range of neural network architectures. DPPL allows the model to learn the system in an end-to-end manner, compared to other approaches where constraints are satisfied either through a post-processing step or at inference. `ProbHardE2E` optimizes a strictly proper scoring rule, without making any distributional assumptions on the target, which enables it to obtain robust distributional estimates (in contrast to existing approaches that generally optimize likelihood-based objectives, which can be biased by their distributional assumptions and model choices); and it can incorporate a range of non-linear constraints (increasing the power of modeling and flexibility). We apply `ProbHardE2E` in learning partial differential equations with uncertainty estimates and to probabilistic time-series forecasting, showcasing it as a broadly applicable general framework that connects these seemingly disparate domains. Our code is available at https://github.com/amazon-science/probharde2e.

## 1 INTRODUCTION

Recently, machine learning (ML) models have been applied to a variety of engineering and scientific tasks, including probabilistic time series forecasting (Rangapuram et al., 2021; Hyndman et al., 2011; Taieb et al., 2017; Olivares et al., 2024b) and scientific applications (Krishnapriyan et al., 2021; Hansen et al., 2023; Négiar et al., 2023; Mouli et al., 2024). Exact enforcement of hard constraints can be essential in domains where any violation of operational or physical requirements (e.g., coherency in hierarchical forecasting, conservation laws in physics, and non-negativity in economics and robotics) is unacceptable (Gould et al., 2022; Hansen et al., 2023; Donti et al., 2021). Limitations of data-driven ML approaches arise in various disciplines where constraints need to be satisfied exactly (Rangapuram et al., 2021; Hansen et al., 2023). Within ML, constraints are typically incorporated as soft penalties, e.g., with a regularization term added to the loss function (Raissi et al., 2019; Li et al., 2024); but they are sometimes incorporated via post-training correction mechanisms, e.g., to enforce a hard constraint (Hansen et al., 2023; Mouli et al., 2024; Cheng et al., 2025). Some methods have managed to enforce hard constraints "end-to-end" in a general framework as a differentiable solver (Négiar et al., 2023; Chalapathi et al., 2024; Rackauckas et al., 2020), as a differentiable optimization layer (Amos & Kolter, 2017; Agrawal et al., 2019; Min et al., 2024), or as an auxiliary procedure (Donti et al., 2021).

The aforementioned hard-constrained models typically provide point estimates without uncertainty quantification (UQ), limiting their use cases in operational and physical domains requiring probabilistic forecasts. Generating output distribution statistics under hard constraints is often computationally

---

[*]Work done during internship at AWS.
[†]Correspondence to: Danielle C. Maddix <dmmaddix@amazon.com>.

expensive or yields only approximate solutions (Robert et al., 1999; Szechtman & Glynn, 2001; Girolami & Calderhead, 2011). There have been domain-specific works in hierarchical probabilistic time series forecasting, which enforce coherency constraints using end-to-end deep learning models (Olivares et al., 2024b; Rangapuram et al., 2021). However, these works either apply only to linear constraints, or they require a computationally expensive sampling procedure in training. Similar approaches have been proposed for computing probabilistic solutions to partial differential equations (PDEs) that satisfy constraints (Hansen et al., 2023; Mouli et al., 2024; Cheng et al., 2025; Gao et al., 2023; Utkarsh et al., 2025). In these works, however, the constraints are only applied as a post-processing step, and they do not lead to an end-to-end solution (of interest in the common situation that one wants to incorporate a hard-constrained model within a larger model, and then optimize the larger model) that optimizes the evaluation accuracy. In both the forecasting and PDE application domains, none of this prior UQ work can handle complex nonlinear (hard) constraints.

In this work, we propose a novel probabilistic framework, `ProbHardE2E`, that integrates a broad class of hard constraints (including non-linear constraints) in an end-to-end fashion, while incorporating UQ. By leveraging key results from statistics and optimization in a novel way, we predict both the mean and covariance of the output data, moving beyond point estimate predictions. `ProbHardE2E` enforces nonlinear constraints with an efficient sampling-free method to generate distribution statistics. Our probabilistic approach enables the effective handling of exogenous spikes and jumps (or other discontinuities) by leveraging data heteroscedasticity, enhancing the model's robustness and flexibility under varying data conditions.

We summarize our key contributions as follows.

- We introduce `ProbHardE2E`, as a general framework to learn a function in an end-to-end manner by optimizing an objective under hard constraints. The framework enables UQ by learning parameters of a multivariate probabilistic distribution. We show that `ProbHardE2E` can incorporate a broad class of deep learning backbone models.
- The key technical novelty of `ProbHardE2E` is a *differentiable probabilistic projection layer* (DPPL) that extends standard projection methods to accommodate UQ while enforcing hard constraints. `ProbHardE2E` can handle constraints ranging from linear equality to general nonlinear equality to convex inequality constraints.
- We use the DPPL to impose constraints directly on the marginals of the multivariate distribution for an efficient sampling-free approach for posterior distribution estimation, which reduces the computational overhead by up to $3$–$5\times$ during training.
- We show that `ProbHardE2E` is effective in two (seemingly-unrelated, but technically-related) tasks, where hard constraints are important: probabilistic time series forecasting; and solving challenging PDEs in scientific machine learning (SciML). We provide an extensive empirical analysis demonstrating that `ProbHardE2E` results in up to $15\times$ lower mean-squared error (MSE) in mean forecast and $2.5\times$ improved uncertainty estimates, measured by the Continuous Ranked Probability Score (CRPS), compared to the baseline methods.
- We show that training with the continuous-ranked probability score (CRPS), rather than negative log-likelihood (NLL) leads to better predictive performance. While the need for this is well-known in, e.g., time series forecasting, previous PDE learning works commonly use NLL-based metrics for UQ.

## 2 RELATED WORK

There is a large body of related work from various communities, ranging from imposing constraints on neural networks for point estimates (Min et al., 2024; Donti et al., 2021), to probabilistic time series forecasting with constraints (Rangapuram et al., 2021; 2023; Olivares et al., 2024b), to imposing constraints on deep learning solutions to PDEs (Négiar et al., 2023; Hansen et al., 2023). Table 5 in Appendix A summarizes some advantages and disadvantages of these methods that are motivated by enforcing hard constraints in these domains. (See Appendix A for additional details.)

## 3 PROBHARDE2E: A UNIFIED PROBABILISTIC OPTIMIZATION FRAMEWORK

In this section, we introduce `ProbHardE2E`. See Algorithm 1 for a summary. (See also Appendix B for a universal approximation guarantee.) In Section 3.1, we discuss the proper evaluation metric

---

**Algorithm 1** `ProbHardE2E`: Training and Inference

---

**Require:** Training data $\{(\phi^{(i)}, u^{(i)})\} \sim \mathcal{D}$, test data $\phi$ and constraints $g(\cdot) \leq 0$, $h(\cdot) = 0$.
**Ensure:** Learnable function $\hat{f}_\theta : \Phi \to \mathcal{Y}$ that outputs constrained distribution parameters.
 1: Pick a model class $\Theta$, initialize weights $\theta \in \Theta$ for probabilistic unconstrained model $f_\theta : \Phi \to \mathcal{Z}$.
 2: **while** $\theta$ not converged **do**
 3:    Predict unconstrained distribution parameters $(\mu_\theta(\phi^{(i)}), \Sigma_\theta(\phi^{(i)}))$.
 4:    **Training Mode**: Project parameters $(\hat{\mu}_\theta(\phi^{(i)}), \hat{\Sigma}_\theta(\phi^{(i)})) = \text{DPPL}((\mu_\theta(\cdot), \Sigma_\theta(\cdot)), g(\cdot), h(\cdot))$.
 5:    Update $\theta \in \bar{\Theta}$ by minimizing the CRPS loss $\ell(\mathbf{Y}_\theta(\phi^{(i)}), u^{(i)})$.
 6: **end while**
 7: **Inference Mode**: Project random variable $\mathbf{Y}_\theta(\phi) = \text{DPPL}(\mathbf{Z}_\theta(\phi), g(\cdot), h(\cdot))$, where $\mathbf{Z}_\theta(\phi)$ and $\mathbf{Y}_\theta(\phi)$ denote the unconstrained and unconstrained random variables, respectively.
 8: **Return** Feasible predicted sample $u^*(z_\theta(\phi)) \sim \mathbf{Y}_\theta(\phi)$, where $z_\theta(\phi) \sim \mathbf{Z}_\theta(\phi)$.

---

for a constrained probabilistic learner, and we define our objective function that corresponds to that evaluation metric. In Section 3.2, we propose our differentiable probabilistic projection layer (DPPL) that enforces the hard constraints. In Section 3.3, we describe how to compute the parameters of the resulting constrained posterior distribution. In Section 3.4, we discuss update rules for various types of constraints (linear equality, nonlinear equality, and convex inequality constraints). In Section 3.5, we propose a sample-free formulation for satisfying the constraints while optimizing for the objective.

### 3.1 PROBABILISTIC EVALUATION METRICS AND OBJECTIVE FUNCTION

We formulate the problem of *probabilistic learning under constraints*. The goal of this problem is to learn a function $\hat{f}_\theta : \Phi \to \mathcal{Y}$, where $\Phi \subset \mathbb{R}^m$ denotes the input space, $\theta \in \bar{\Theta} \subseteq \Theta$ denotes the feasible parameter space, and $\mathcal{Y} \subset \mathbb{R}^k$ denotes the space of predicted distribution parameters that meet the constraints. Given a multivariate distribution class, these learned parameters induce a predictive multivariate random variable $\mathbf{Y}_\theta(\phi^{(i)})$, where $(\phi^{(i)}, u^{(i)}) \sim \mathcal{D}$, where $\phi^{(i)} \in \Phi$, $u^{(i)} \in \mathbb{R}^n$, and $\mathcal{D}$ denotes training data from a distribution $\mathcal{D}$. Each realization of $\hat{u}(\phi^{(i)}) \sim \mathbf{Y}_\theta(\phi^{(i)}) \in \mathbb{R}^n$ is required to satisfy predefined hard constraints of the form $g(\hat{u}(\phi^{(i)})) \leq 0$ and $h(\hat{u}(\phi^{(i)})) = 0$. We can formulate this constrained optimization problem as follows:

$$\underset{\theta \in \Theta,\ g(\mathbf{Y}_\theta(\phi^{(i)})) \leq 0,\ h(\mathbf{Y}_\theta(\phi^{(i)})) = 0}{\arg\min} \mathbb{E}_{(\phi^{(i)}, u^{(i)}) \sim \mathcal{D}}\, \ell\big(\mathbf{Y}_\theta(\phi^{(i)}), u^{(i)}\big), \tag{1}$$

where denotes a proper scoring rule.

One widely-used (strictly) proper scoring rule for continuous distributions is the continuous ranked probability score (CRPS) (Gneiting & Raftery, 2007). The CRPS simultaneously evaluates sharpness (how concentrated or "narrow" the distribution is) and calibration (how well the distributional coverage "aligns" with actual observations). More formally, for an observed scalar outcome $y$ and a corresponding probabilistic distributional estimate, $Y$, the CRPS is defined as:

$$\text{CRPS}(Y, y) = \mathbb{E}_Y |Y - y| - \tfrac{1}{2}\mathbb{E}_Y |Y - Y'|, \tag{2}$$

where $Y'$ denotes an i.i.d. copy of $Y$. Compared to other scoring rules, e.g., the log probability scoring rules, which require assumptions on the outcome variable, the CRPS is robust to probabilistic model mis-specification. Because of this unique property, the CRPS is widely used as the evaluation metric in many applications, e.g., probabilistic time series forecasting (Gasthaus et al., 2019; Rangapuram et al., 2021; Park et al., 2022; Olivares et al., 2024b), quantile regression (Fakoor et al., 2023), precipitation nowcasting (Ravuri et al., 2021; Gao et al., 2023) and weather forecasting (Rasp & Lerch, 2018; Kochkov et al., 2024; Price et al., 2025).

We align our training objective with the proposed evaluation metric above, by directly optimizing the CRPS in Eq. (2) in Problem 1. We define the loss as the sum of the univariate CRPS:

$$\ell\big(\mathbf{Y}_\theta(\phi^{(i)}), u^{(i)}\big) = \sum_{j=1}^{n} \text{CRPS}((\mathbf{Y}_\theta(\phi^{(i)}))_j, u_j^{(i)}). \tag{3}$$

The CRPS naturally aligns with the goal of producing feasible and well-calibrated predictions, as the metric rewards distributions that closely match observed outcomes. Enforcing our constraints in the distribution space guarantees that every sample from the predicted distribution is physically or operationally valid. Consequently, modeling the loss through the CRPS provides a principled way to reconcile domain constraints with distributional accuracy.

## 3.2 DIFFERENTIABLE PROBABILISTIC PROJECTION LAYER (DPPL)

We transform the constrained Problem 1 into the unconstrained optimization problem:

$$\arg\min_{\theta \in \bar{\Theta}} \mathbb{E}_{(\phi^{(i)}, u^{(i)}) \sim \mathcal{D}} \, \ell\big(\mathbf{Y}_\theta(\phi^{(i)}), u^{(i)}\big), \tag{4}$$

where $\bar{\Theta} \subseteq \Theta$ denotes the feasible parameter space that ensures constraint satisfaction, and $\ell$ denotes the loss function in Eq. (3). We solve this using a two-step procedure: first define a predictive output distribution, then project it onto the constraint manifold using a *differentiable probabilistic projection layer (DPPL)* for end-to-end optimization.

Our framework begins with an established probabilistic backbone model. This can be a Gaussian Process (Rasmussen & Williams, 2006), neural process (Kim et al., 2019), DeepVAR (Salinas et al., 2019; Rangapuram et al., 2021), or ensembles of neural networks or operators (Mouli et al., 2024). The base model $f_\theta : \Phi \to \mathbb{R}^k$ predicts the distribution parameters (mean $\mu_\theta(\phi^{(i)})$ and covariance $\Sigma_\theta(\phi^{(i)})$, for $\theta \in \Theta$) – without constraint awareness. We then use a reparameterization function $r : \mathbb{R}^k \times \mathbb{R}^n \to \mathbb{R}^l$ to define the distribution in one of two ways: either as an identity map, where $l = k$, that returns $f_\theta(\phi^{(i)}) = \big(\mu_\theta(\phi^{(i)}), \Sigma_\theta(\phi^{(i)})\big)$ for our efficient sample-free paradigm during training; or as a map, where $l = n$, that combines the distribution parameters with noise $\xi \sim p(\xi) \in \mathbb{R}^n$, where $p$ denotes a tractable sampling distribution, and gives a sample $z_\theta(\phi^{(i)}) \sim \mathbf{Z}_\theta(\phi^{(i)}) \in \mathbb{R}^n$ from the predicted distribution to generate constrained samples at inference. This dual-mode design balances training efficiency with strict constraint feasibility at inference.

The reparameterization function induces the base (unconstrained) distribution parameters or predictive random variable as:

$$r(f_\theta(\phi^{(i)}), \xi) = \begin{cases} \big(\mu_\theta(\phi^{(i)}), \Sigma_\theta(\phi^{(i)})\big), & \text{(Training)} \\ \mathbf{Z}_\theta(\phi^{(i)}), & \text{(Inference)} \end{cases} \tag{5}$$

Following this Predictor Step above, we use the DPPL in the Corrector Step to restrict the parameter space to $\bar{\Theta} \subseteq \Theta$, such that for all $\hat{u}_\theta(\phi^{(i)}) \sim \mathbf{Y}_\theta(\phi^{(i)})$, the constraints $g(\hat{u}_\theta(\phi^{(i)})) \leq 0$ and $h(\hat{u}_\theta(\phi^{(i)})) = 0$ are satisfied. The DPPL is our core architecture innovation for leveraging the base model to learn predictions that satisfy the given constraints. We define the projected distribution parameters or projected predictive random variable as:

$$\text{DPPL}(r(f_\theta(\phi^{(i)}), \xi), g(\cdot), h(\cdot)) = r(\hat{f}_\theta(\phi^{(i)}), \xi) = \begin{cases} \big(\hat{\mu}_\theta(\phi^{(i)}), \hat{\Sigma}_\theta(\phi^{(i)})\big), & \text{(Training)} \\ \mathbf{Y}_\theta(\phi^{(i)}), & \text{(Inference)} \end{cases} \tag{6}$$

for $r(f_\theta(\phi^{(i)}), \xi)$ in Eq. (5), where $\hat{f}_\theta : \Phi \to \mathcal{Y} \subset \mathbb{R}^k$ denotes the probabilistic model that outputs the constrained distribution parameters $(\hat{\mu}_\theta(\phi^{(i)}), \hat{\Sigma}_\theta(\phi^{(i)}))$. Our DPPL yields a constraint-satisfying realization $u^* \sim \mathbf{Y}_\theta(\phi^{(i)})$ as the final predictive random variable.

This two-step approach mirrors predictor-corrector methods (Boyd & Vandenberghe, 2004; Bertsekas, 1997), with the DPPL serving as our key architectural innovation for ensuring constraint satisfaction. Equivalently, the DPPL can be formulated as a constrained least squares problem on the samples of $\mathbf{Z}_\theta(\phi^{(i)})$. (See Appendix C for details.) Prior works on imposing hard constraints in time series and solving PDEs (Rangapuram et al., 2021; Hansen et al., 2023) reduce to special cases of our method with linear constraints. (See Appendix D for details.) We draw $z_\theta(\phi^{(i)}) \sim \mathbf{Z}_\theta(\phi^{(i)})$, and we solve the following constrained optimization problem:

$$u^*(z_\theta(\phi^{(i)})) := \arg\min_{\hat{u}_\theta(\phi^{(i)}) \in \mathbb{R}^n, \, g(\hat{u}_\theta(\phi^{(i)})) \leq 0, \, h(\hat{u}_\theta(\phi^{(i)})) = 0} \|\hat{u}_\theta(\phi^{(i)}) - z_\theta(\phi^{(i)})\|_Q^2, \tag{7}$$

where $u^*(z_\theta(\phi^{(i)}))$ denotes a predicted sample of $\mathbf{Y}_\theta(\phi^{(i)})$, and where $\|x\|_Q = \sqrt{x^\top Q x}$ for some symmetric positive semi-definite matrix $Q$. (See Appendix E for details on the flexibility of learning various forms of $Q$.)

### 3.3 DPPL ON THE DISTRIBUTION PARAMETERS FOR LOCATION-SCALE DISTRIBUTIONS

In this subsection, we detail how to directly compute the parameters for the constrained distribution by applying our DPPL on the base distribution parameters for an efficient, sampling-free during training. To do so, we can assume that the prior distribution $\mathcal{F}$ belongs to a multivariate, location-scale family, i.e., a distribution such that any affine transformation $\mathbf{Y}$ of a random variable $\mathbf{Z} = \mu + \Sigma^{1/2}\xi \sim \mathcal{F}(\mu, \Sigma)$ and $\xi \sim \mathcal{F}(0, 1)$, remains within the same distribution family $\mathcal{F}$. This is an example of how to compute the random variable in Eq. (5) for a multivariate location-scale distribution. A familiar case of this is when $\mathbf{Z} \sim \mathcal{N}(\mu, \Sigma)$ and $\mathbf{Y} = A\mathbf{Z} + B$ is an affine transformation; in which case $\mathbf{Y} \sim \mathcal{N}(A\mu + B, A\Sigma A^\top)$. Alternatively, we can show that when $\mathbf{Y}$ is a nonlinear transformation of $\mathbf{Z}$, it has approximately (to first-order) the same distribution $\mathbf{Z}$, with an appropriately-chosen set of parameters (given in Eq. (8) below). We state this result more formally in Theorem 3.1. The proof, given in Appendix F, uses a first-order Taylor expansion to linearize the nonlinear function transformation, and is similar to the Multivariate Delta Method (Casella & Berger, 2001).

**Theorem 3.1.** *Let $\mathbf{Z} \sim \mathcal{F}(\mu, \Sigma)$ be a random variable, where the underlying distribution $\mathcal{F}$ belongs to a multivariate location-scale family of distributions, with mean $\mu$ and covariance $\Sigma$; and let $\mathcal{T}$ be a function with continuous first derivatives, such that $J_\mathcal{T}(\mu)\Sigma J_\mathcal{T}(\mu)^\top$ is symmetric positive semi-definite. Then, the transformed distribution $\mathbf{Y} = \mathcal{T}(\mathbf{Z})$ converges in distribution with first-order accuracy to $\mathcal{F}(\hat{\mu}, \hat{\Sigma})$ with mean $\hat{\mu} = \mathcal{T}(\mu)$ and covariance $\hat{\Sigma} = J_\mathcal{T}(\mu)\Sigma J_\mathcal{T}(\mu)^\top$, where $J_\mathcal{T}(\mu) = \nabla\mathcal{T}(\mu)^\top$ denotes the Jacobian of $\mathcal{T}$ with respect to $z$ evaluated at $\mu$.*

Let $\mathbf{Z} \sim \mathcal{F}(\mu, \Sigma)$ denote the prior distribution and $z \sim \mathbf{Z}$. We apply Theorem 3.1 with $\mathcal{T}(z) = u^*(z)$, where $u^*(z)$ denotes the solution of the constrained least squares problem in Problem (7). In this case, the projected random variable satisfies $\mathbf{Y} \sim \mathcal{F}(\hat{\mu}, \hat{\Sigma})$ with updated parameters:

$$\hat{\mu} = \mathcal{T}(\mu), \qquad \hat{\Sigma} = J_\mathcal{T}(\mu)\,\Sigma\,J_\mathcal{T}(\mu)^\top. \tag{8}$$

### 3.4 DPPL FOR VARIOUS CONSTRAINT TYPES

In this subsection, we discuss how to compute the DPPL for various constraint types (linear equality, nonlinear equality, and convex inequality) for both train and inference modes. Table 1 shows these constraints types require different treatments: linear equality have closed-form projections, nonlinear equality can be solved with iterative methods, and convex inequality require optimization solvers.

Table 1: Summary of DPPL in `ProbHardE2E` for various constraint types. For linear equality constraints, the oblique projection $P_{Q^{-1}} = I - Q^{-1}A^\top(AQ^{-1}A^\top)^{-1}A$; for nonlinear equality constraints, $R$ denotes the first-order optimality conditions.

| Constraint Type | Solution $u^*(z)$ | Solver Type | Jacobian $J_\mathcal{T}$ |
|---|---|---|---|
| **Linear Equality** | $P_{Q^{-1}}z + (I - P_{Q^{-1}})A^\dagger b$ | closed-form | $P_{Q^{-1}}$ |
| **Nonlinear Equality** | $(u^*, \lambda^*)$ s.t. $R(u^*, \lambda^*; z) = 0$ | nonlinear | implicit differentiation |
| **Convex Inequality** | $\underset{h(\hat{u})=0,\,g(\hat{u})\leq 0}{\mathrm{argmin}} \|\hat{u} - z\|_Q^2$ | convex opt. | sensitivity analysis; argmin differentiation |

#### 3.4.1 LINEAR EQUALITY CONSTRAINTS

For linear equality constraints, we have an underdetermined linear system $h(\hat{u}) = A\hat{u} - b = 0$, where $A \in \mathbb{R}^{q \times n}, q < n$, and has full row rank $q$. In this case, we can derive a closed-form solution to the constrained least squares Problem (7). In this case, both training and inference modes are equivalent since the DPPL projection is exact. (See Appendix C.1.)

#### 3.4.2 NONLINEAR EQUALITY CONSTRAINTS

For nonlinear equality constraints, $h(\hat{u}) = 0$, we can no longer derive the exact closed-form expression for the solution. Instead, we can provide an expression which is satisfied by the optimal solution. In particular, we approximate the parameter-level projection at training time. This can then be

solved for the posterior mean $\hat{\mu} = u^*(\mu)$ in Eq. (8) with the nonlinear transformation $\mathcal{T}(\mu) = u^*(\mu)$ with iterative optimization methods, e.g., Newton's Method. (We can then compute the posterior covariance $\hat{\Sigma}$ in Eq. (8) by estimating the Jacobian $J_{\mathcal{T}}(\mu)$ by differentiating the nonlinear equations $u^*(z) = z - Q^{-1}\nabla h(u^*(z))^\top \lambda$, $h(u^*(z)) = 0$ with respect to $z$ via the implicit function theorem (Blondel et al., 2022), and evaluating it at $\mu$. (See Appendix C.2.) At inference, we project each sample exactly with our custom, batched optimization solver to ensure strict constraint feasibility.

### 3.4.3 (NONLINEAR) CONVEX INEQUALITY CONSTRAINTS

For convex inequality constraints, $\hat{u}$ in Problem (7) is in a convex set, $\mathcal{C} \subset \mathbb{R}^n$. Closed-form expressions (such as those in previous subsections for linear and nonlinear equality constraints) do not exist (Boyd & Vandenberghe, 2004). Instead, we rely on convex optimization solvers to ensure computational efficiency and scalability to compute the solution $u^*$ in training. The gradients of the convex program can be calculated efficiently using sensitivity analysis (Bertsekas, 1997; Bonnans & Shapiro, 2013), argmin differentiation (Sun et al., 2022; Agrawal et al., 2019; Amos & Kolter, 2017; Gould et al., 2016), and/or variational analysis (Rockafellar & Wets, 2009). These techniques provide a means to compute the Jacobian $J_{\mathcal{T}}(\mu)$, which represents the sensitivity of the optimal solution $u^*$ to changes in the input vector $\mu$, whose projection we are essentially computing to the convex constraints space. During inference, we solve the convex program per sample. (See Appendix C.3.)

### 3.5 SAMPLE-FREE WITH CLOSED-FORM CRPS

We use a closed-form expression for the CRPS to enable a computationally efficient and sample-free approach for evaluating the CRPS in the loss function $\ell$ in Eq. (3). Calculating the CRPS for an arbitrary distribution requires generating samples (Rangapuram et al., 2021; Gneiting & Raftery, 2007), but closed-form expressions for the CRPS exist for several location-scale distributions (Gaussian, logistic, student's t, beta, gamma, uniform). Most notably, for the univariate Gaussian, the closed-form CRPS is given as: $\text{CRPS}_{\mathcal{N}}(z) = \left[ z \cdot (2P(z) - 1) + 2p(z) - \frac{1}{\sqrt{\pi}} \right]$, where $p(z) = \frac{1}{\sqrt{2\pi}} \exp\left(-z^2/2\right)$ denotes the standard normal probability density function (PDF), and $P(z) = \int_{-\infty}^{z} p(y)dy$ denotes the standard normal cumulative distribution function (CDF) for $z \sim \mathcal{N}(0, 1)$ (Gneiting et al., 2005; Taillardat et al., 2016). This sample-free formulation is especially beneficial when the DPPL is computationally intensive, e.g., in the presence of nonlinear constraints.

## 4 EMPIRICAL RESULTS

In our empirical evaluations, we aim to answer the following five questions about `ProbHardE2E`:

(Q1) Does training end-to-end with the imposed hard constraints improve upon the performance of imposing them only at inference time?

(Q2) Is using a general oblique projection more beneficial than using the commonly-used orthogonal projection, and if so when?

(Q3) Does training with the distribution-agnostic proper scoring rule, CRPS instead of NLL, improve performance?

(Q4) What are the computational savings of projecting directly on the distribution parameters and using the closed form CRPS vs. projecting on the samples?

(Q5) How does `ProbHardE2E` perform when extended to more general constraints, e.g., nonlinear equality and convex inequality constraints?

See Appendix G for details on the test datasets, Appendix H for implementation details, and Appendix I for additional empirical results.

**Test Cases.** We demonstrate the efficacy of `ProbHardE2E` in two constrained optimization applications: PDEs; and hierarchical forecasting. We show that our methodology with DPPL is model-agnostic, as demonstrated through its high-performance integration with different base models across applications. We first consider a series of PDE problems with varying levels of difficulty in learning their solutions, following the empirical evaluation from Hansen et al. (2023). These PDEs are categorized as "easy," "medium," and "hard," with the difficulty level determined by the smoothness or sharpness of the solution. (See Appendix G.1 for details.) In addition to PDEs, we

also evaluate `ProbHardE2E` on five hierarchical time-series forecasting benchmark datasets from Alexandrov et al. (2019), where the goal is to generate probabilistic predictions that are coherent with known aggregation constraints across cross-sectional hierarchies (Rangapuram et al., 2021). (See Appendix G.2 for details.)

**Baselines.** We compare two variants of `ProbHardE2E`, i.e., `ProbHardE2E-Ob`, which uses a general oblique projection ($Q = \Sigma^{-1}$) projection and is our default unless otherwise specified, and `ProbHardE2E-Or`, which uses an orthogonal projection ($Q = I$), against several probabilistic deep learning baselines commonly used for uncertainty quantification in constrained PDEs and probabilistic time series forecasting. For PDEs, `ProbHardE2E` uses `VarianceNO` (Mouli et al., 2024), which is a probabilisitic extension of the Fourier Neural Operator (FNO) (Li et al., 2021) as the unconstrained model. We compare `ProbHardE2E` with: (i) `HardC`, which is based on Négiar et al. (2023); Hansen et al. (2023), and which imposes the orthogonal projection only on the mean, but does not update the covariance; (ii) `ProbConserv` (Hansen et al., 2023), which applies the oblique projection only at inference time, and works only with linear constraints (in the nonlinear constraint case, we compare with `ProbHardInf`, which is a variant of `ProbConserv` that imposes the nonlinear constraint at inference time only); (iii) `SoftC` (Hansen et al., 2023), which introduces a soft penalty on constraint violation à la PINNs (Raissi et al., 2019; Li et al., 2024) during training but does not guarantee constraint satisfaction at inference; and (iv) the unconstrained model backbone `VarianceNO`. For hierarchical time-series forecasting, `ProbHardE2E` uses `DeepVAR` (Salinas et al., 2019) as the probabilistic base model. We compare `ProbHardE2E` with: (i) `ProbConserv`; (ii) `HierE2E` (Rangapuram et al., 2021), which enforces linear constraints via an end-to-end orthogonal projection; classical statistical approaches including (iii) `ARIMA-NaiveBU`, (iv) `ETS-NaiveBU` (Hyndman et al., 2011; 2025), (v) `PERMBU-MINT` (Taieb et al., 2017; Olivares et al., 2022); and (vi) the unconstrained model backbone `DeepVAR`.

**Evaluation.** We evaluate `ProbHardE2E` using the following metrics: Mean Squared Error (MSE), which measures the mean prediction accuracy; Constraint Error (CE), which measures the constraint errors on the samples (conservation law for PDEs and coherency for hierarchical time series forecasting); and Continuous Ranked Probability Score (CRPS), which measures performance in uncertainty quantification (UQ). (See Appendix H.3 for details on the metrics.) For each model, we report these metrics when trained with either CRPS or Negative Log-Likelihood (NLL) as the loss. Although originally optimized with NLL, we also train a CRPS-based variant of `ProbConserv` to ensure a fair comparison. The experiments are conducted on a single NVIDIA V100 GPU in the PDEs case, and on an Intel(R) Xeon(R) CPU E5-2603 v4 @ 1.70GHz in the time series forecasting case. To ensure scalability, we use a diagonal covariance matrix $Q$ in Problem 7, following prior work (Hansen et al., 2023; Mouli et al., 2024). (See Appendix E for low-rank and full covariance structures.)

## 4.1 LINEAR CONSERVATION AND HIERARCHICAL CONSTRAINTS

In this subsection, we test `ProbHardE2E` on linear constraints. Table 2 presents our comparative evaluation results across multiple PDE datasets under linear constraints, and Table 3 presents our evaluation results across multiple time series forecasting datasets. We use these results to answer questions (Q1)-(Q4) raised above.

**Q1.** Regarding (Q1) on the benefits of training end-to-end with a hard constraint, the results demonstrate that our method achieves superior performance compared to existing approaches. Specifically, when measured against two accuracy metrics across four PDE datasets in Table 2, our method with either oblique (`ProbHardE2E-Ob`) or orthogonal (`ProbHardE2E-Or`) projection consistently outperforms both `ProbConserv`, which applies constraints only during post-processing, and `SoftC`, which implements constraints as soft penalties, as measured by the desired CRPS metric. This performance advantage directly addresses research question (Q1), showing that our end-to-end approach is more effective than methods that treat constraints as either post-processing steps or soft penalties. In addition, across five diverse hierarchical time-series datasets, our method achieves state-of-the-art CRPS on the LABOUR, TOURISM-L, and WIKI datasets. On the TOURISM and TRAFFIC datasets, it remains highly competitive, outperforming traditional approaches, e.g., ARIMA and ETS, and offering comparable performance to specialized methods, e.g., `PERMBU-MINT` and `HierE2E`.

Table 2: Test metrics on constrained PDEs across four datasets, which are ordered top to bottom in their learning difficulty. Metrics include MSE $\times 10^{-5}$, constraint (conservation) error (CE) $\times 10^{-3}$, and CRPS $\times 10^{-3}$. Each algorithm is trained with either CRPS or NLL. Best values per row are highlighted in bold.

| Dataset | Metric | ProbHardE2E-Ob | | ProbHardE2E-Or | | HardC | | ProbConserv | | SoftC | | VarianceNO (base) | |
| --- | --- | --- | --- | --- | --- | --- | --- | --- | --- | --- | --- | --- | --- |
| | | CRPS | NLL | CRPS | NLL | CRPS | NLL | CRPS | NLL | CRPS | NLL | CRPS | NLL |
| Heat | MSE | 0.036 | 0.047 | 0.031 | 0.301 | 0.031 | 0.090 | **0.027** | 1.26 | 0.051 | 0.156 | 0.029 | 2.01 |
| | CE | **0** | **0** | **0** | **0** | **0** | **0** | **0** | **0** | 0.852 | 4.806 | 1.76 | 34.3 |
| | CRPS | 0.304 | 0.37 | **0.271** | 0.713 | 0.275 | 0.452 | 0.392 | 4.27 | 0.354 | 1.129 | 0.396 | 4.39 |
| PME | MSE | 9.59 | **6.16** | 9.01 | 11.08 | 8.870 | 10.55 | 8.801 | 10.5 | 8.187 | 7.362 | 7.945 | 8.132 |
| | CE | **0** | **0** | **0** | **0** | **0** | **0** | **0** | **0** | 17.091 | 29.31 | 20.19 | 27.2 |
| | CRPS | 2.01 | 2.65 | 1.798 | 1.80 | 1.785 | **1.667** | 2.03 | 2.49 | 2.065 | 2.444 | 2.02 | 2.43 |
| Advection | MSE | 131 | 262 | **88.09** | 310.82 | 103.78 | 458.38 | 134 | 277 | 148.11 | 599.11 | 149 | 605 |
| | CE | **0** | **0** | **0** | **0** | **0** | **0** | **0** | **0** | 19.334 | 182.99 | 18.9 | 182 |
| | CRPS | 4.19 | 7.03 | **2.94** | 8.669 | 3.236 | 11.23 | 3.88 | 7.90 | 3.963 | 9.702 | 3.98 | 10.1 |
| Stefan | MSE | **186** | 207 | 394.84 | 432.92 | 394.29 | 433.28 | 303 | 273 | 431.89 | 429.06 | 425 | 425 |
| | CE | **0** | **0** | **0** | **0** | **0** | **0** | **0** | **0** | 166.93 | 168.75 | 180 | 169 |
| | CRPS | **7.52** | 7.85 | 14.147 | 14.67 | 14.432 | 14.539 | 7.85 | 8.33 | 9.878 | 10.062 | 9.51 | 10.2 |

**Q2.** Regarding (Q2) on the effectiveness of the oblique vs. orthogonal projections, Table 2 shows while both oblique (`ProbHardE2E-Ob`) and orthogonal (`ProbHardE2E-Or`) variants of `ProbHardE2E` enforce zero constraint error, their impact on predictive fidelity varies significantly, depending on the difficulty of the PDE problem. For the "easy" (smooth) Heat equation and "medium" tasks (PME and Advection), orthogonal projection reduces CRPS by $10 - 30\%$ relative to oblique projection and improves MSE by up to $33\%$. However, in the "hard" (sharp) nonlinear Stefan problem, the oblique projection-based method improves CRPS by more than $50\%$ compared to the orthogonal projection. Table 3 shows that `ProbHardE2E-Or` generally performs better on the time series datasets with cross-sectional hierarchies, as it improves CRPS on LABOUR and TOURISM datasets, compared to `ProbHardE2E-Ob`. This addresses (Q2) that correcting predictions along covariance-weighted (oblique) directions better preserves the true uncertainty structure around shocks and spikes, performing more effectively on problems with heteroscedastic data.

Table 3: CRPS $\times 10^{-3}$ for hierarchical time-series datasets across various probabilistic forecasting algorithms. Constraint (coherency) error (CE) is given in parenthesis and is equal to 0 for all methods except the base unconstrained `DeepVAR`. `PERMBU-MINT` is not available for TOURISM-L, because the dataset has a nested hierarchical structure, and `PERMBU-MINT` is not well-defined on this type of dataset (Rangapuram et al., 2021). `ARIMA-NaiveBU`, `ETS-NaiveBU` and `PERMBU-MINT` are deterministic models with no model uncertainty.

| Dataset | ProbHardE2E-Ob | ProbHardE2E-Or | ProbConserv | HierE2E | ARIMA-NaiveBU | ETS-NaiveBU | PERMBU-MINT | DeepVAR (base) |
| --- | --- | --- | --- | --- | --- | --- | --- | --- |
| LABOUR | 36.1± 2.7 (0) | **28.6**±6.5 (0) | 45.8±6.5 (0) | 50.5±20.6 (0) | 45.3 (0) | 43.2 (0) | 39.3 (0) | 38.2±4.5 (0.215) |
| TOURISM | 98.9±13.0 (0) | 82.4±6.6 (0) | 100.7±7.7 (0) | 103.1±16.3 (0) | 113.8 (0) | 100.8 (0) | **77.1** (0) | 92.5±2.2 (2818.01) |
| TOURISM-L | **155.2**±3.6 (0) | 156.4±9.4 (0) | 176.9±21.5 (0) | 161.3±10.9 (0) | 174.1 (0) | 169.0 (0) | – | 158.1±10.2 (70000) |
| TRAFFIC | 55.0±10.6 (0) | 60.6±7.8 (0) | 71.0±3.9 (0) | **41.8±7.8 (0)** | 80.8 (0) | 66.5 (0) | 67.7 (0) | 40.0±2.6 (0.192) |
| WIKI | **212.1**±29.4 (0) | 215.8±16.9 (0) | 264.7± 30.7 (0) | 216.5±26.7 (0) | 377.2 (0) | 467.3 (0) | 281.2 (0) | 229.4±15.8 (8398.6) |

**Q3.** Regarding (Q3) on the training objective, Table 2 shows that training with the proper scoring rule, such as CRPS, improves UQ (measured by CRPS) across nearly all PDE datasets compared to training with the commonly-used NLL in previous SciML works. The only exception is `HardC` in the PME. The CRPS training objective also improves mean accuracy (measured by MSE) in approximately three-quarters of the dataset-model experiments. In addition, Table 3 shows that on four out of five time series datasets, we improve upon the results of `HierE2E`, which uses the `DeepVAR` base model with an orthogonal projection on the samples, and which optimizes the sampling-based quantile loss by projecting directly on the distribution parameters.

**Q4.** Regarding (Q4) on the computational efficiency of our sampling-free approach, Fig. 1(a) shows the training time per epoch for the hierarchical time series datasets. Models trained for time series and PDEs (see Appendix I.1) with 100 posterior samples per training step incur a 3.3–4.6× increase in epoch time relative to `ProbHardE2E`, which avoids sampling altogether by using a closed-form CRPS loss. Note that the computational overhead of `ProbHardE2E` is approximately 2× that of the unconstrained model. However, compared to the sampling-based probabilistic baselines, our

approach with the DPPL layer that directly projects distribution parameters and a closed-form CRPS objective offers significant training-time speed-ups across all forecasting and PDE testbeds.

## 4.2 GENERAL NONLINEAR EQUALITY AND CONVEX INEQUALITY CONSTRAINTS (Q5)

In this subsection, we test `ProbHardE2E` on nonlinear equality and convex inequality constraints to address question (Q5) on PDE datasets, as (current) time series forecasting applications usually need predictions to satisfy only linear (e.g., hierarchical) constraints.

### 4.2.1 NONLINEAR EQUALITY CONSTRAINTS

We test `ProbHardE2E` with general nonlinear constraints using nonlinear conservation laws from PDEs. (See Appendix I.2 for details.) Importantly, Table 4 shows that even in this challenging case of nonlinear constraints, the constraint error (CE) on the samples is 0 so that we ensure strict feasibility on the samples. In addition, we see superior performance of enforcing nonlinear constraints with `ProbHardE2E`. We see an even larger MSE performance improvement of at most $\approx 15 - 17\times$ when trained on CRPS, and CRPS performance improvement of at most $\approx 2.5\times$ for $m \in [2, 3]$ over the various baselines that apply the nonlinear constraint just at inference time (`ProbHardInf`), as a reduced linear constraint (`ProbConserv`) at inference time, and the unconstrained model (`VarianceNO`). These results highlight the benefit of training end-to-end with the constraint in the nonlinear case. In particular, this validates our dual mode training and inference approach, where we obtain the computational benefits of projecting on the parameters and approximately satisfying the constraint during training, and exact constraint sanctification at inference time while achieving state-of-the-art performance measured in CRPS. In addition, Fig. 1(b) shows that `ProbHardE2E` is able to significantly better capture the shock and has tighter uncertainty estimates, leading to lower CRPS values than the baselines.

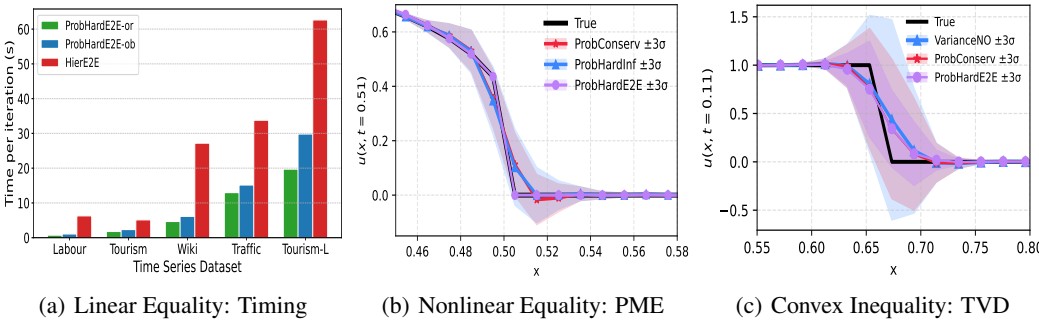

| (a) Linear Equality: Timing | (b) Nonlinear Equality: PME | (c) Convex Inequality: TVD |

Figure 1: `ProbHardE2E` on the various constraint types. (a) **Linear Equality**: Average time per iteration (in seconds) for `ProbHardE2E`, compared to the `HierE2E` on five hierarchical time-series datasets; (b) **Nonlinear Equality**: Mean $\pm 3$ standard deviation for the PME with conservation constraint at time $t = 0.51$, with PDE parameter $m_{\text{train}} \in [3, 4]$ and $m_{\text{test}} = 3.88$; (c) **Convex Inequality**: Mean $\pm 3$ standard deviation for linear advection with TVD constraint at time $t = 0.51$, with PDE parameter $\beta_{\text{train}} \in [1, 2]$ and $\beta_{\text{test}} = 1.5$. The horizontal axes in (b)-(c) are zoomed in to highlight the uncertainty near the propagating front.

### 4.2.2 (NONLINEAR) CONVEX INEQUALITY CONSTRAINTS

We impose a convex relaxation of the total variation diminishing (TVD) constraint that has been commonly used in numerical methods for PDEs to minimize artificial oscillations (LeVeque, 2002). In particular, we impose $\text{TVD} = \sum_{n=1}^{N_t} \sum_{i=1}^{N_x} |u(t_n, x_{i+1}) - u(t_n, x_i)|$ as a regularization term. (See Appendix I.3 for details.) Note that this discrete form is analogous to total variation denoising in signal processing (Rudin et al., 1992; Boyd & Vandenberghe, 2004). Fig. 1(c) illustrates that imposing this TVD relaxation improves the shock location prediction, compared to the unconstrained model `VarianceNO`. We see that `ProbHardE2E` has smaller variance, compared to both `ProbConserv`, which only enforces the conservation law, and `VarianceNO`. Most importantly, we see that `ProbHardE2E` is less likely to predict non-physical negative samples, which violates

Table 4: Test metrics on the nonlinear PME with PDE coefficient $k(u) = u^m$, which controls the sharpness of the solution (larger values are "harder"), for NLL and CRPS training. The training and test parameters are sampled from this range of $m$. Metrics include MSE $\times 10^{-6}$, calibration error (CE) $\times 10^{-3}$, and CRPS $\times 10^{-4}$. Best values per row and metric are bolded.

| PME Dataset | Metric | ProbHardE2E-Ob | | ProbHardE2E-Or | | ProbHardInf | | ProbConserv | | VarianceNO (base) | |
| | | CRPS | NLL | CRPS | NLL | CRPS | NLL | CRPS | NLL | CRPS | NLL |
|---|---|---|---|---|---|---|---|---|---|---|---|
| $m \in [2,3]$ | MSE | **5.04** | 106.638 | 9.38 | 43.5 | 78.185 | 86.147 | 88.539 | 94.467 | 80.342 | 89.212 |
| | CE | **0** | **0** | **0** | **0** | **0** | **0** | **0** | **0** | 0.020 | 0.028 |
| | CRPS | **8.648** | 18.867 | 11.34 | 14.8 | 19.005 | 32.977 | 20.672 | 36.724 | 20.779 | 37.140 |
| $m \in [3,4]$ | MSE | 296.4 | 471.3 | **3.19** | 134.7 | 157.8 | 200.2 | 184.5 | 276.4 | 162 | 201.5 |
| | CE | **0** | **0** | **0** | **0** | **0** | **0** | **0** | **0** | 14.8 | 34.1 |
| | CRPS | 11.23 | 16.9 | **7.10** | 11.18 | 22.60 | 30.4 | 24.7 | 35.1 | 23.7 | 48.5 |
| $m \in [4,5]$ | MSE | 424.8 | 716.8 | **1.59** | 206.49 | 280.4 | 332.3 | 276.7 | 619.9 | 199.2 | 341.7 |
| | CE | **0** | **0** | **0** | **0** | **0** | **0** | **0** | **0** | 22.8 | 59.7 |
| | CRPS | 10.8 | 19.9 | **5.62** | 9.36 | 23.3 | 32.4 | 25.4 | 41.3 | 27.2 | 35.9 |

the positivity of the true solution. In addition, the variance of the ProbHardE2E solution also has a smaller peak above the shock, and hence it is less prone to oscillations than the other baselines.

## 5 CONCLUSION

In this work, we propose ProbHardE2E, which seamlessly incorporates constraints ranging from commonly used linear constraints to challenging nonlinear constraints into black-box probabilistic deep learning models using a differentiable probabilistic projection layer (DPPL). We show that ProbHardE2E is applicable in a wide range of scientific and operational domains, ranging from linear coherency constraints in time series forecasting to nonlinear conservation laws in solving PDEs. Contrary to prior works (Hansen et al., 2023), which only support linear global conservation constraints, ProbHardE2E supports nonlinear conservation constraints. This paves the way for future work to enforce conservation locally over sub-regions or over control volumes à la finite volume methods (LeVeque, 2002). Future work also includes extending our method to handle general non-convex, nonlinear inequality constraints using advanced optimization techniques or relaxations, to richer covariance parameterizations, e.g., low-rank or dense, and to empirical distributions other than location-scale families by sample projection.

### ACKNOWLEDGMENTS

The authors thank Jiayao Zhang, Pedro Eduardo Mercado Lopez and Gaurav Gupta for discussions on this work and related early investigations. The authors would additionally like to thank Chris Rackauckas and Alan Edelman for their insights and support of the project.

### REPRODUCIBILITY STATEMENT

Code used to produce the experiments in this paper is available at https://github.com/amazon-science/probharde2e.

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

# A    RELATED WORK

In this section, we review works that have been motivated by dealing with hard constraints in various domains including imposing constraints in neural networks (Min et al., 2024; Donti et al., 2021), probabilistic time series forecasting (Rangapuram et al., 2023; 2021; Olivares et al., 2024b) and scientific machine learning (Négiar et al., 2023; Hansen et al., 2023). Table 5 summarizes several of these methods. We see that existing general methododologies, e.g., HardNet (Min et al., 2024) and DC3 (Donti et al., 2021), work across various domains and different types of constraints— HardNet handles convex constraints, and DC3 tackles nonlinear ones. The biggest limitation of these methods is that they provide point estimates only. Despite having the point forecast satisfying the constraints, they are unsuitable for PDEs and forecasting applications, which generally now require variance estimates. Hier-E2E (Rangapuram et al., 2021) and CLOVER (Olivares et al., 2024b) are specialized solutions for forecasting problems, which both deal with probabilistic forecasting under linear constraints. Linear constraints are common in the time series forecasting domain. Both methods require sampling during training, which can be computationally intensive. Within the PDE-focused methods, ProbConserv (Hansen et al., 2023) and HardC (Hansen et al., 2023) handle linear constraints and include variance estimates in their probabilistic models. The training procedure with the constraint is not end-to-end since the constraint is only applied at inference time. PDE-CL (Négiar et al., 2023) handles nonlinear constraints and supports end-to-end training, but at the cost of not supporting variance estimation.

Table 5: Summary of methods motivated by dealing with hard constraints in various domains: imposing constraints in neural networks (Min et al., 2024; Donti et al., 2021), probabilistic time series forecasting (Rangapuram et al., 2023; 2021; Olivares et al., 2024b) and scientific machine learning (Négiar et al., 2023; Hansen et al., 2023). For models that only provide point estimates, we evaluate their capabilities on sampling-free training and satisfying constraints on distributions, while treating the point estimate as a degenerate probabilistic distribution.

| Method | Domain | Constraint Type | End-to-End | Prob. Model w/ Variance Estimate | Sampling-free Training | Constraint on Dstbn. |
|---|---|---|---|---|---|---|
| HardNet (Min et al., 2024) | General | Convex | ✓ | ✗ | ✓ | ✓ |
| DC3 (Donti et al., 2021) | General | Nonlinear | ✓ | ✗ | ✓ | ✓ |
| Hier-E2E (Rangapuram et al., 2021; 2023) | Forecasting | Linear | ✓ | ✓ | ✗ | ✗ |
| CLOVER (Olivares et al., 2024b) | Forecasting | Linear | ✓ | ✓ | ✗ | ✓ |
| PDE-CL (Négiar et al., 2023) | PDEs | Nonlinear | ✓ | ✗ | ✓ | ✓ |
| ProbConserv (Hansen et al., 2023) | PDEs | Linear | ✗ | ✓ | ✓ | ✓ |
| HardC (Hansen et al., 2023) | PDEs | Linear | ✗ | ✓ | ✓ | ✓ |
| ProbHardE2E (Ours) | General | Nonlinear | ✓ | ✓ | ✓ | ✓ |

Our proposed method `ProbHardE2E` bridges the gaps left by its predecessors. It combines the flexibility of general domain application with the ability to handle nonlinear constraints, and it maintains end-to-end training capability. Perhaps most notably, it achieves this while incorporating probabilistic modeling with variance estimates, supporting sampling-free training, and maintaining constraints on distributions.

## A.1    IMPOSING DETERMINISTIC CONSTRAINTS ON NEURAL NETWORKS

Enforcing constraints in neural networks (NNs) has been explored in various forms. In fact, activation functions, e.g., sigmoid, ReLU, and softmax, inherently impose implicit constraints by restricting outputs to specific intervals. Another well-established method for enforcing constraints in NNs involves differentiating the Karush-Kuhn-Tucker (KKT) conditions, which enables backpropagation through optimization problems. This technique has led to the development of differentiable optimization layers (Amos & Kolter, 2017; Agrawal et al., 2019) and projected gradient descent methods (Rosen, 1960).

Most commonly, soft constraint methods, e.g., Lagrange duality based methods, are often employed in ML to balance minimizing the primary objective with satisfying the constraints. These methods typically do so by adding the constraint as a penalty term to the loss function (Battaglia et al., 2018). For example, Lagrange dual methods and relaxed formulations are frequently used to allow flexibility

in the optimization process, while still guiding the model toward feasible solutions. These methods encourage—but do not strictly enforce—adherence to the constraints during training; and this lack of strict enforcement can be undesirable in some scientific disciplines, where known constraints must be satisfied exactly (Hansen et al., 2023; Rangapuram et al., 2021).

More recently, there have been approaches that have been motivated by satisfying hard constraints. DC3 (Donti et al., 2021) is a general method for learning a family of constrained optimization problems using a correction and variable completion procedure. The variable completion approach has a strong theoretical and practical foundation. A limitation is that it does require knowledge of the structure of the matrix $A$ to identify these corresponding predicted and completed variables, which hinders its generalizability. In addition for inequality constraints, it only achieves hard constraint satisfaction asymptotically; that is, the "correction" procedure to enforce inequality constraints is carried out through gradient-descent optimization algorithms (Min et al., 2024; Donti et al., 2021).

Projection-based methods are an alternate method for enforcing hard constraints in NNs. Min et al. (2024) identify cases where the aforementioned DC3 framework (Donti et al., 2021) is outperformed by their proposed HardNet projection layer approach. Additionally, Min et al. (2024) investigate the expressiveness of projection layers, which builds on the foundational work in Agrawal et al. (2019); Amos & Kolter (2017), to further advance the understanding of constraint enforcement in NNs. Projection-based methods have also been used to enforce constraints on specific architectures, e.g., neural ordinary differential equations (Neural ODEs) (Kasim & Lim, 2022; Matsubara & Yaguchi, 2023). In particular, White et al. (2024) use a closed-form projection operator to enforce a nonlinear constraint $g(u) = 0$ in a Neural ODE, using techniques from Boumal (2024). A common limitation of these works is that they impose the constraint deterministically, on point estimates rather than on an entire probability distribution.

## A.2 Probabilistic Time Series Forecasting

Probabilistic time series forecasting extends beyond predicting point estimates, e.g., the mean or median, by providing a framework to capture uncertainty, with practical application in estimating high quantiles, e.g., P99. Classical statistical models, e.g., autoregressive integrated moving average (ARIMA) models (Box et al., 2015), state-space models (Kalman, 1960), and copula-based models (Joe, 1997) are prominent examples. More recently, deep learning models, e.g., DeepAR (Salinas et al., 2020) and its multivariate extension DeepVAR (Salinas et al., 2019), multivariate quantile regression-based models (Wen et al., 2018; Eisenach et al., 2022; Park et al., 2022), temporal fusion transformers (TFT) (Lim et al., 2021), and foundational models based on large language models (LLMs) (Ansari et al., 2024; Hoo et al., 2025; Das et al., 2023; Woo et al., 2024) have shown success. See Benidis et al. (2022) for an overview.

Linear constraints are important in hierarchical time series forecasting, where coherent aggregation constraints are required over regions (Rangapuram et al., 2021; Olivares et al., 2024b) and over temporal hierarchies (Rangapuram et al., 2023). This constraint is critical in scenarios where higher-level forecasts must be aggregates of lower-level ones, which is a common requirement in time-series forecasting. Early works in hierarchical forecasting focus on mean forecasts under linear/hierarchical constraints, starting from the naive bottom-up and top-down approaches (Hyndman & Athanasopoulos, 2018). More recently, Hyndman et al. (2011) show that the Middle-Out projection-based method yields better forecast accuracy. Since then, projection-based reconciliation methods, e.g., GTOP (Van Erven & Cugliari, 2015), MinT (Wickramasuriya et al., 2019), and ERM (Ben Taieb & Koo, 2019) have been developed. These methods leverage generic time series models, e.g., ARIMA and exponential smoothing (ETS), to derive the unconstrained mean forecast, and then they use a linear projection to map these forecasts to the consistent space. Taieb et al. (2017) further extend the reconciliation method (MinT) to probabilistic forecasting by developing a method called PERMBU that constructs cross-sectional dependence through a sequence of permutations. A more thorough review of forecasting reconciliation is provided in Athanasopoulos et al. (2024).

To better handle the trade-off between forecast accuracy and coherence within the model, several works have proposed end-to-end methodologies. For example, Rangapuram et al. (2021) propose Hier-E2E, which is an end-to-end learning approach that imposes constraints via an orthogonal projection on samples from the distribution. Hier-E2E produces coherent probabilistic forecasts without requiring explicit post-processing reconciliation. One limitation is that Hier-E2E relies on

expensive sampling techniques to achieve this coherence, by projecting directly on the samples rather than on the distribution itself, which has a closed-form expression.

Separately, DPMN (Olivares et al., 2024a) adopts an equality constraint completion approach similar to that in DC3 (Donti et al., 2021), rather than a projection-based approach, for satisfying the coherency constraint. DPMN assumes that the bottom-level series follow a Poisson mixture distribution, with a multivariate discrete distribution on the Poisson rates across bottom-level series. Compared to Hier-E2E, DPMN uses a CNN-based encoder rather than DeepVAR, and it shows improved forecast accuracy over Hier-E2E. As follow-up work to DPMN, Olivares et al. (2024b) propose CLOVER, a framework which enforces coherency as a hard constraint in probabilistic hierarchical time series forecasting models using a CNN encoder. In particular, CLOVER only predicts the base forecasts in the first step, and it solves for the aggregate forecasts by leveraging the constraint relation. Finally, CLOVER models the joint distribution of all the forecasts in the scoring function calculation. Similar to Hier-E2E, CLOVER also relies on sampling to enforce hierarchical coherency and to generate uncertainty estimates. This affects the training time, and it requires tuning of the number of samples for an accurate approximation of this scoring function. Although CLOVER admits constraint satisfaction, the exact provably convergent procedure only exists for linear equality constraints (Donti et al., 2021), and it has not been applied to nonlinear equality or convex inequality constraints.

### A.3 SCIENTIFIC MACHINE LEARNING (SCIML)

Partial differential equations (PDEs) are ubiquitous in science and engineering applications, and have been used to model various physical phenomena, ranging from nonlinear fluid flows with the Navier-Stokes equations to nonlinear heat transfer. Classical numerical methods to solve PDEs include finite difference (LeVeque, 2007), finite element (Hughes, 2003), and finite volume methods (LeVeque, 2002). These numerical methods discretize the solution on a spatio-temporal mesh, and the accuracy increases as the mesh becomes finer. For this reason, numerical methods can be computationally expensive on real-world, time dependent, 3D spatial problems that require fine meshes for high accuracy.

Recently, Scientific Machine Learning (SciML) methods aim to alleviate the high computational requirement of numerical methods. State-of-the-art data-driven methods include operator-based methods, which aim to learn a mapping from PDE parameters or initial/boundary conditions to the PDE solution, e.g., Neural Operators (NOs) (Li et al., 2020; 2021; Gupta et al., 2021) and DeepONet (Lu et al., 2021), and message-passing Graph Neural Networks (GNNs)-based MeshGraphNets (Pfaff et al., 2021; Fortunato et al., 2022). These data-driven methods are not guaranteed to satisfy the PDE or known physical laws exactly, e.g., conservation laws (Hansen et al., 2023; Mouli et al., 2024) or boundary conditions (Saad et al., 2023; Cheng et al., 2025) since they only implicitly encode the physics through the supervised training simulation data (Kadambi et al., 2023).

Similar to imposing constraints on NNs, most work on imposing constraints in SciML has been focused on soft constraints. One well-known type of approach is Physics-Informed Neural Networks (PINNs) (Raissi et al., 2019; Karniadakis et al., 2021), which approximates the solution of a PDE as a NN. PINNs and similarly Physics-Informed Neural Operators (PINOs) (Li et al., 2024) impose the PDE as an additional term in the loss function, akin to the aforementioned soft constraint regularization. Krishnapriyan et al. (2021); Edwards (2022) identify limitations of this approach on problems with large PDE parameter values, where adding this regularization term can actually make the loss landscape sharp, non-smooth and more challenging to optimize. In addition, Hansen et al. (2023) show that adding the constraint to the loss function does not guarantee exact constraint enforcement, which can be critical in the case of conservation and other physical laws. This constraint violation primarily happens because the Lagrangian duals of the constrained optimization problem are typically non-zero, i.e., the physical constraint is not exactly satisfied.

Recent work has studied imposing physical knowledge as hard constraints on various SciML methods. Négiar et al. (2023) propose PDE-CL, which uses differentiable programming and the implicit function theorem (Krantz & Parks, 2002) to impose nonlinear PDE constraints directly. Chalapathi et al. (2024), extend this work by leveraging a mixture-of-experts (MoE) framework to better scale the method. Similarly, Beucler et al. (2021) impose analytical constraints in NNs with applications to climate modeling. In addition, Universal Differential Equations (UDEs) (Rackauckas et al., 2020; Utkarsh et al., 2024) provide a GPU-compatible and end-to-end differentiable way to learn PDEs

while also enforcing implicit constraints. Chen et al. (2024) propose KKT-hPINN to enforce linear equality constraints by using a projection layer that is derived from the KKT conditions. These works show the benefit of imposing the PDE as a hard constraint rather than as a soft constraint. A limitation of these methods is that they impose the constraints deterministically, and they do not provide estimates of the underlying variance or uncertainties. To address this, Hansen et al. (2023) propose ProbConserv, which incorporates linear conservation laws as hard constraints on probabilistic models by performing an oblique projection to update the unconstrained mean and variance estimates. Limitations are that this projection is only applied as a post-processing step at inference time, and the method only supports linear constraints.

## B  UNIVERSAL APPROXIMATION GUARANTEES

In this section, we prove a universal approximation result for our differentiable probabilistic projection layer (DPPL) in the case of convex constraints. As a consequence of this result, our `ProbHardE2E` in Algorithm 1 is a universal approximator, and can approximate any continuous target function that satisfies the given constraints. Our proof of this result generalizes the analysis of Min et al. (2024) from the case $Q = I$ to our broader framework with arbitrary $Q$. Since $Q$ is symmetric positive definite, we compute its Cholesky factorization $Q = LL^T$, where $L$ denotes a lower triangular matrix. We then show that if $f_\theta$ is a universal approximator, i.e., a sufficiently wide and deep neural network, then our DPPL preserves this universal approximation capability. Hence, `ProbHardE2E` retains the expressiveness of neural networks, both in its probabilistic formulation and in enforcing hard constraints.

Our DPPL in Problem (7) is formulated in terms of projecting the samples $z_\theta(\phi^{(i)}) \sim \mathbf{Z}_\theta(\phi^{(i)})$, where $\mathbf{Z}_\theta(\phi^{(i)}) \sim \mathcal{F}(\mu_\theta(\phi^{(i)}), \Sigma_\theta(\phi^{(i)}))$ for some multivariate location-scale distribution $\mathcal{F}$, and where $(\phi^{(i)}, u^{(i)}) \sim \mathcal{D}$ denotes training data from a distribution $\mathcal{D}$. The mean $\mu_\theta(\phi^{(i)}) \in \mathbb{R}^n$ and covariance $\Sigma_\theta(\phi^{(i)}) \in \mathbb{R}^{n \times n}$ are the output from a deep neural network, $f_\theta : \Phi \to \mathbb{R}^k$. The value of $k$ depends on the approximation for $\Sigma_\theta(\phi^{(i)})$, e.g., $k = n + n^2$ for dense $\Sigma_\theta(\phi^{(i)})$, $k = 2n$ for $\Sigma_\theta(\phi^{(i)}) = \text{diag}(\sigma_1^2, \ldots, \sigma_n^2)$, or $k = n$ for $\Sigma_\theta(\phi^{(i)}) = I$. For notational simplicity, we assume in this section that $f_\theta : \Phi \to \mathbb{R}^n$ corresponds to the components that output the mean $\mu_\theta(\phi^{(i)})$. By setting $z_\theta(\phi^{(i)}) = \mu_\theta(\phi^{(i)}) = f_\theta(\phi^{(i)})$ in Problem (7), our DPPL can also be formulated in terms of projecting the mean $\mu_\theta(\phi^{(i)})$ as the following constrained least squares problem:

$$\hat{\mu}_\theta(\phi^{(i)}) := \underset{\substack{\tilde{\mu}_\theta(\phi^{(i)}) \in \mathbb{R}^n \\ g(\tilde{\mu}_\theta(\phi^{(i)})) \leq 0 \\ h(\tilde{\mu}_\theta(\phi^{(i)})) = 0}}{\arg\min} \|\tilde{\mu}_\theta(\phi^{(i)}) - f_\theta(\phi^{(i)})\|_Q^2, \tag{9}$$

where $Q$ denotes a symmetric positive definite matrix and $g(\cdot) \leq 0, h(\cdot) = 0$ denote the convex constraints. In particular, we show that the projected mean $\hat{\mu}_\theta(\phi^{(i)})$ is a universal approximator of the true solution $u \in \mathbb{R}^n$. We now state the theorem and provide its proof below.

**Theorem B.1.** *Consider Problem 9 with the projection step defined using a symmetric positive definite (SPD) matrix $Q \in \mathbb{R}^{n \times n}$, a deep neural network that is a universal approximator, $f_\theta : \Phi \to \mathbb{R}^n$, where $\Phi \subset \mathbb{R}^m$ denotes a compact set, convex constraints $g(\cdot) \leq 0, h(\cdot) = 0$, and training data $(\phi^{(i)}, u^{(i)}) \sim \mathcal{D}$ from a distribution $\mathcal{D}$. For any continuous target function that satisfies the constraints, i.e., the true solution $u : \Phi \to \mathcal{C} \subseteq \mathbb{R}^n$, $u \in C(\Phi)$, where $\mathcal{C}$ denotes the convex set of feasible points defined by the convex constraints and $C(\Phi)$ denotes the space of continuous functions on $\Phi$, there exists a choice of network parameters for $f_\theta(\phi^{(i)}) = \mu_\theta(\phi^{(i)}) \in \mathbb{R}^n$, such that the projected mean, which is composition of $f_\theta$ with the projection step, i.e., $\Pi_\mathcal{C}^Q\big(f_\theta(\phi^{(i)})\big) = \hat{f}_\theta(\phi^{(i)}) = \hat{\mu}_\theta(\phi^{(i)}) \in \mathcal{C} \subseteq \mathbb{R}^n$, approximates the target function arbitrarily well, where $\hat{f}_\theta : \Phi \to \mathcal{C} \subseteq \mathbb{R}^n$. Hence, under these conditions, `ProbHardE2E` is a universal approximator for constrained mappings.*

*Proof.* Let $\mathcal{C} \subseteq \mathbb{R}^n$ denote the convex set of feasible points defined by the convex constraints. Consider the projection operator onto $\mathcal{C}$ with respect to the $Q$-norm:

$$\Pi_\mathcal{C}^Q(v) = \arg\min_{\tilde{\mu}_\theta(\phi^{(i)}) \in \mathcal{C}} \|\tilde{\mu}_\theta(\phi^{(i)}) - v\|_Q^2. \tag{10}$$

Since $Q$ is symmetric positive definite (SPD), it has the following Cholesky factorization,

$$Q = LL^\top,$$

where $L \in \mathbb{R}^{n \times n}$ denotes a lower triangular matrix with strictly positive diagonal entries, and hence is invertible. By the definition of the $Q$-norm, and then substituting in its Cholesky factorization, we have

$$
\begin{aligned}
\|\tilde{\mu}_\theta(\phi^{(i)}) - v\|_Q^2 &= (\tilde{\mu}_\theta(\phi^{(i)}) - v)^\top Q(\tilde{\mu}_\theta(\phi^{(i)}) - v) \\
&= (\tilde{\mu}_\theta(\phi^{(i)}) - v)^\top LL^\top(\tilde{\mu}_\theta(\phi^{(i)}) - v) \\
&= \big((\tilde{\mu}_\theta(\phi^{(i)}) - v)^\top L\big)(L^\top(\tilde{\mu}_\theta(\phi^{(i)}) - v)) \\
&= \big(L^\top(\tilde{\mu}_\theta(\phi^{(i)}) - v)\big)^\top(L^\top(\tilde{\mu}_\theta(\phi^{(i)}) - v)) \\
&= \|L^\top(\tilde{\mu}_\theta(\phi^{(i)}) - v)\|_2^2.
\end{aligned}
\tag{11}
$$

This shows that the $Q$-norm is equivalent to the standard Euclidean norm after the linear transformation $L^\top$.

We define the invertible linear mapping $\Psi : \mathbb{R}^n \to \mathbb{R}^n$ by $\Psi(v) = L^\top v$. Then using Eq. (11), the $Q$-norm in Eq. (10) can be written as the Euclidean norm as follows:

$$
\begin{aligned}
\Pi_{\mathcal{C}}^Q(v) &= \mathrm{argmin}_{\tilde{u}_\theta(\phi^{(i)}) \in \mathcal{C}} \|L^T(\tilde{\mu}_\theta(\phi^{(i)}) - v)\|_2^2 \\
&= \mathrm{argmin}_{\tilde{u}_\theta(\phi^{(i)}) \in \mathcal{C}} \|\Psi(\tilde{\mu}_\theta(\phi^{(i)})) - \Psi(v)\|_2^2 \\
&= \Psi^{-1}\Big(\mathrm{argmin}_{w \in \Psi(\mathcal{C})} \|w - \Psi(v)\|_2^2\Big),
\end{aligned}
\tag{12}
$$

where $w = \Psi(\tilde{\mu}_\theta(\phi^{(i)}))$. Hence, the projection can be expressed as the Euclidean projection onto the transformed set $\Psi(\mathcal{C})$. It is well known that the Euclidean projection onto a closed convex set is nonexpansive and is Lipschitz continuous. (See, e.g., Min et al. (2024).)

Now, suppose that $f_\theta(\phi^{(i)})$ is a deep neural network that is a universal approximator, i.e., for any continuous function $u : \Phi \to \mathbb{R}^n$, $u \in C(\Phi)$, and for any $\epsilon > 0$, there exists parameters $\theta$ such that

$$\sup_{\phi^{(i)} \in \Phi} \|u(\phi^{(i)}) - f_\theta(\phi^{(i)})\| < \epsilon,$$

where $\Phi \subset \mathbb{R}^m$ denotes a compact set and $C(\Phi)$ denotes the space of continuous functions on $\Phi$. Let $u : \Phi \to \mathcal{C} \subseteq \mathbb{R}^n$, $u \in C(\Phi)$, be any continuous target function whose outputs satisfy the constraints. Since $\Pi_{\mathcal{C}}^Q$ is continuous (as the composition of the continuous mapping $\Psi$, the Euclidean projection onto $\Psi(\mathcal{C})$, and $\Psi^{-1}$), it follows by the universal approximation theorem and properties of continuous functions that the projected mean $\Pi_{\mathcal{C}}^Q(f_\theta(\phi^{(i)})) = \hat{\mu}(\phi^{(i)})$ can uniformly approximate $u(\phi^{(i)})$ arbitrarily well on $\Phi$. In other words, for every $\epsilon > 0$, there exists a choice of network parameters $\theta$ such that

$$\sup_{\phi^{(i)} \in \Phi} \|u(\phi^{(i)}) - \Pi_{\mathcal{C}}^Q(f_\theta(\phi^{(i)}))\| < \epsilon.$$

Thus, the composition of the neural network $f_\theta$ with the $Q$-norm projection retains the universal approximation property for any continuous target function satisfying the constraints. $\qquad\square$

## C   COMPUTATION OF POSTERIOR DISTRIBUTION FOR VARIOUS CONSTRAINT TYPES

In this section, we discuss how to compute the differentiable probabilistic projection layer (DPPL) that projects the distribution parameters (Eq. (6)) in `ProbHardE2E` for various constraint types, which are summarized in Table 1.

### C.1   LINEAR EQUALITY CONSTRAINTS

In this subsection, we provide the closed-form expressions for the constrained posterior distribution parameters, i.e., the mean $\hat{\mu}$ and covariance $\hat{\Sigma}$ in Eq. (8), from the DPPL in `ProbHardE2E` for linear

equality constraints. Linear equality constraints occur in a wide range of applications, including coherency constraints in hierarchical time series forecasting (Hyndman et al., 2011; Rangapuram et al., 2021; Petropoulos et al., 2022; Olivares et al., 2024b), divergence-free conditions in incompressible fluid flows (Raissi et al., 2019; Richter-Powell et al., 2022), boundary conditions in PDEs (Saad et al., 2023), and global linear conservation law constraints (Hansen et al., 2023; Mouli et al., 2024).

**Proposition C.1.** *For linear equality constraints, $h(\hat{u}) = A\hat{u} - b = 0$, with $A \in \mathbb{R}^{q \times n}$, with full row rank $q$, where $q < n$, and $b \in \mathbb{R}^q$, the optimal solution $u^*$ to Problem (7) is given as $u^*(z) = P_{Q^{-1}} z + (I - P_{Q^{-1}}) A^\dagger b$, where $P_{Q^{-1}} = I - Q^{-1} A^\top (AQ^{-1}A^\top)^{-1} A$, denotes an oblique projection operator, and $A^\dagger$ denotes the Moore-Penrose inverse. In addition, if $\mathbf{Z} \sim \mathcal{F}(\mu, \Sigma)$ and $z \sim \mathbf{Z}$ for multivariate, location-scale distribution $\mathcal{F}$, then $u^* \sim \mathbf{Y}$, where $\mathbf{Y} \sim \mathcal{F}(\hat{\mu}, \hat{\Sigma})$ and $\hat{\mu}, \hat{\Sigma}$ are given in Eq. (8) with $\mathcal{T}(z) = u^*(z)$, which simplifies to the closed-form expressions, $\hat{\mu} = P_{Q^{-1}} \mu + (I - P_{Q^{-1}}) A^\dagger b$ and $\hat{\Sigma} = P_{Q^{-1}} \Sigma P_{Q^{-1}}^\top$.*

*Proof.* Using the Lagrange multiplier $\lambda \in \mathbb{R}^q$, we can form the Lagrangian of Problem (7) with linear constraints to obtain:

$$L(\hat{u}, \lambda; z) = \frac{1}{2}\hat{u}^\top Q\hat{u} - z^\top Q\hat{u} + \lambda^\top (A\hat{u} - b).$$

The sufficient optimality conditions to obtain $(u^*, \lambda^*)$ are the first-order gradient conditions:

$$\nabla_{\hat{u}} L(\hat{u}, \lambda; z)|_{u^*, \lambda^*} = Qu^* - Q^\top z + A^\top \lambda^* = 0, \tag{13a}$$
$$\nabla_{\lambda} L(\hat{u}, \lambda; z)|_{u^*, \lambda^*} = Au^* - b = 0. \tag{13b}$$

Since $Q$ is SPD, $Q = Q^\top$ and $Q^{-1}$ exists. Then from Eq. (13a), we obtain:

$$Q(u^* - z) + A^\top \lambda^* = 0,$$

which simplifies to the following expression for $u^*$:

$$u^* = z - Q^{-1}A^\top \lambda^*. \tag{14}$$

We solve Eq. (13b) for $u^*$ using the Moore-Penrose inverse, i.e., $u^* = A^\dagger b$, where $A^\dagger = A^\top (AA^\top)^{-1}$. Note that $AA^\top \in \mathbb{R}^{q \times q}$ is invertible with full rank $q$ since $A \in \mathbb{R}^{q \times n}$ has full row rank $q \leq n$. Substituting this expression into Eq. (14) for $u^*$ gives:

$$A^\top (AA^\top)^{-1} b = z - Q^{-1}A^\top \lambda^*.$$

Rearranging for the optimal Lagrange multiplier $\lambda^*$, and multiplying both sides by $A$ gives:

$$(AQ^{-1}A^\top)\lambda^* = Az - \underbrace{(AA^\top)(AA^\top)^{-1}}_{I} b.$$

Now, $AQ^{-1}A^\top \in \mathbb{R}^{q \times q}$ is invertible since $A$ has full row rank $q$. Then we obtain:

$$\lambda^* = (AQ^{-1}A^\top)^{-1}(Az - b).$$

Substituting in the expression for $\lambda^*$ into Eq. (14) gives the following expression for the optimal solution:

$$\begin{aligned} u^* &= z - Q^{-1}A^\top (AQ^{-1}A^\top)^{-1}(Az - b), \\ &= (I - Q^{-1}A^\top (AQ^{-1}A^\top)^{-1}A)z + Q^{-1}A^\top (AQ^{-1}A^\top)^{-1}b. \end{aligned} \tag{15}$$

Let

$$P_{Q^{-1}} = I - Q^{-1}A^\top (AQ^{-1}A^\top)^{-1}A, \tag{16}$$

be an oblique projection. To see that this is a projection, observe that

$$\begin{aligned} P_{Q^{-1}}^2 &= (I - Q^{-1}A^\top (AQ^{-1}A^\top)^{-1}A)(I - Q^{-1}A^\top (AQ^{-1}A^\top)^{-1}A) \\ &= I - 2Q^{-1}A^\top (AQ^{-1}A^\top)^{-1}A + Q^{-1}A^\top (AQ^{-1}A^\top)^{-1}AQ^{-1}A^\top (AQ^{-1}A^\top)^{-1}A \\ &= I - 2Q^{-1}A^\top (AQ^{-1}A^\top)^{-1}A + Q^{-1}A^\top (AQ^{-1}A^\top)^{-1}A \\ &= I - Q^{-1}A^\top (AQ^{-1}A^\top)^{-1}A \\ &= P_{Q^{-1}}. \end{aligned}$$

Then, the expression for $u^*$ in Eq. (15) simplifies to:

$$\begin{aligned}
u^*(z) &= P_{Q^{-1}}z + Q^{-1}A^\top(AQ^{-1}A^\top)^{-1}(AA^\dagger)b, \\
&= P_{Q^{-1}}z + (Q^{-1}A^\top(AQ^{-1}A^\top)^{-1}A)A^\dagger b, \\
&= P_{Q^{-1}}z + (I - P_{Q^{-1}})A^\dagger b,
\end{aligned} \tag{17}$$

since $AA^\dagger = AA^\top(AA^\top)^{-1} = I$.

Since the expression for $u^*$ in Eq. (17) is a linear transformation $\mathcal{T}$ of $z \sim \mathcal{F}(\mu, \Sigma)$, we can use Theorem 3.1 with $\mathcal{T}(z) = u^*(z)$ to write the expression for $u^* \sim \mathcal{F}(\hat{\mu}, \hat{\Sigma})$, where:

$$\hat{\mu} = \mathcal{T}(\mu) = u^*(\mu) = P_{Q^{-1}}\mu + (I - P_{Q^{-1}})A^\dagger b, \tag{18a}$$

$$\hat{\Sigma} = J_\mathcal{T}(\mu)\Sigma J_\mathcal{T}(\mu)^\top = P_{Q^{-1}}\Sigma P_{Q^{-1}}^\top. \tag{18b}$$

It can easily be verified that $J_\mathcal{T}(\mu) = P_{Q^{-1}}$ by differentiating Eq. (17) with respect to $z$. We note that Eq. (18) holds exactly in the case of linear constraints since $\mathcal{T}$ is a linear transformation of $z$. $\quad\square$

We note that our probabilistic method applies to underdetermined linear systems when $q < n$, where there is existence of many solutions. When $q = n$ and $A$ has full row rank, the solution is unique. In this case, the projection $P_Q^{-1}$ has the following deterministic solution,

$$\hat{\mu} = A^\dagger b, \qquad \hat{\Sigma} = 0.$$

In this case, it reduces to a non-probabilistic point prediction methods, similar to the `HardC` baseline (Hansen et al., 2023), where only the mean is updated.

## C.2   Nonlinear Equality Constraints

In this subsection, we describe how to compute the DPPL in `ProbHardE2E` for general nonlinear equality constraints. Nonlinear equality constraints naturally arise in applications that involve structural, physical, or geometric consistency. These include closed-loop kinematics in robotics (Toussaint et al., 2019), nonlinear conservation laws (LeVeque, 1990) in PDE-constrained surrogate modeling (Biegler et al., 2003; Zahr & Persson, 2016; Négiar et al., 2023) with applications in climate modeling (Bolton & Zanna, 2019; Zanna & Bolton, 2020; Beucler et al., 2021), compressible flows in aerodynamics (Tezaur et al., 2017) and atomic modeling (Müller, 2022; Sturm & Wexler, 2022).

**Proposition C.2.** *For nonlinear equality constraints, $h(\hat{u}) = 0 \in \mathbb{R}^q$, where $h : \mathbb{R}^n \to \mathbb{R}^q$, the optimal solution $u^*(z)$ to Problem (7) forms a pair $(u^*(z), \lambda^*)$ which satisfies $u^*(z) = z - Q^{-1}\nabla h(u^*(z))^\top \lambda^*$ and $h(u^*(z)) = 0$. In addition, if $\mathbf{Z} \sim \mathcal{F}(\mu, \Sigma)$ and $z \sim \mathbf{Z}$ for multivariate, location-scale distribution $\mathcal{F}$, then $u^* \sim \mathbf{Y}$, where $\mathbf{Y} \sim \mathcal{F}(\hat{\mu}, \hat{\Sigma})$ and $\hat{\mu}, \hat{\Sigma}$ are given in Eq. (8) with $\mathcal{T}(z) = u^*(z)$.*

*Proof.* Using the Lagrange multiplier $\lambda \in \mathbb{R}^q$, we can form the Lagrangian of Problem (7) with nonlinear equality constraints to obtain:

$$L(\hat{u}, \lambda; z) = \frac{1}{2}\hat{u}^\top Q\hat{u} - z^\top Q\hat{u} + \lambda^\top h(\hat{u}).$$

The sufficient optimality conditions to obtain $(u^*, \lambda^*)$ are the first-order gradient conditions:

$$R(u^*, \lambda^*; z) = \begin{cases} \nabla_{\hat{u}}L(\hat{u}, \lambda; z)|_{u^*, \lambda^*} = Q(u^* - z) + \nabla h(u^*)\lambda^* = 0, \\ \nabla_\lambda L(\hat{u}, \lambda; z)|_{u^*, \lambda^*} = h(u^*) = 0. \end{cases} \tag{19}$$

We solve Eq. (19) via root-finding methods, e.g., Newton's method for $(u^*, \lambda^*)$ to obtain $u^*(z) = \arg\{\hat{u} : R(\hat{u}, \lambda^*; z) = 0\}$, where the root-finding solution $u^*$ is implicitly dependent on $z$. Since the expression for $u^*$ is a nonlinear transformation $\mathcal{T}$ of $z \sim \mathcal{F}(\mu, \Sigma)$, we can use Theorem 3.1 with $\mathcal{T}(z) = u^*(z)$ to write the expression for $u^* \sim \mathcal{F}(\hat{\mu}, \hat{\Sigma})$, where:

$$\hat{\mu} = \mathcal{T}(\mu) = u^*(\mu), \tag{20a}$$

$$\hat{\Sigma} = J_\mathcal{T}(\mu)\Sigma J_\mathcal{T}(\mu)^\top, \tag{20b}$$

hold to first-order accuracy. In the following Proposition C.3, we detail the iterative algorithm to compute the terms $u^*(\mu)$ and $J_\mathcal{T}(\mu)$ in Eq. (20). $\quad\square$

**Proposition C.3.** *Let $h(\hat{u}) = 0 \in \mathbb{R}^q$ be a smooth nonlinear equality constraint, where $h : \mathbb{R}^n \to \mathbb{R}^q$. Consider the constrained projection problem from Problem (7) with $z = \mu$:*

$$u^*(\mu) = \arg\min_{\substack{\hat{u} \in \mathbb{R}^n \\ h(\hat{u}) = 0}} f(\hat{u}), \tag{21}$$

*where $Q \succ 0$ and $f(\hat{u}) = \frac{1}{2} \|\hat{u} - \mu\|_Q^2$ denotes our quadratic objective.*

1. *At each iteration, we solve the linearized Karush-Kuhn-Tucker (KKT) system using the Schur complement to obtain:*

$$\lambda^{(i+1)} = \left(J^{(i)} Q^{-1} J^{(i)\top}\right)^{-1} \left(h(\hat{u}^{(i)}) - J^{(i)}(\hat{u}^{(i)} - \mu)\right), \tag{22a}$$

$$\hat{u}^{(i+1)} = \mu - Q^{-1} J^{(i)\top} \lambda^{(i+1)}, \tag{22b}$$

   *where $J^{(i)} = \nabla h(\hat{u}^{(i)})^\top \in \mathbb{R}^{q \times n}$. At the first iteration with $\hat{u}^{(0)} = \mu$, Eq. (22b) simplifies to:*

$$\hat{u}^{(1)} = \mu - Q^{-1} J^\top \left(J Q^{-1} J^\top\right)^{-1} h(\mu), \tag{23}$$

   *where $J = \nabla h(\mu)^\top$.*

2. *At convergence, the Jacobian $J_{\mathcal{T}}(\mu)$ of the projection map $\mathcal{T}(\mu) := u^*(\mu)$ is given by:*

$$J_{\mathcal{T}}(\mu) := \frac{\partial u^*(\mu)}{\partial \mu} = I - Q^{-1} J^{*\top} (J^* Q^{-1} J^{*\top})^{-1} J^* \in \mathbb{R}^{n \times n}, \tag{24}$$

*where $J^* = \nabla h(u^*)^\top$.*

*Proof.* We begin with the matrix form of the KKT system derived in Eq. (19):

$$R(u^*, \lambda^*; \mu) = \begin{bmatrix} \nabla_{\hat{u}} L(u^*, \lambda^*) \\ \nabla_{\lambda} L(u^*, \lambda^*) \end{bmatrix} = \begin{bmatrix} \nabla f(u^*) + J^{*\top} \lambda^* \\ h(u^*) \end{bmatrix} = 0, \tag{25}$$

with quadratic objective $f$ defined in Problem 21.

**1. Iteration Update.** We use Newton's Method to linearize the KKT system in Eq. (25) evaluated at $(\hat{u}^{(i+1)}, \lambda^{(i+1)})$ about the past iterate $(\hat{u}^{(i)}, \lambda^{(i)})$. For the stationarity condition, which is the first component of $R(u^*, \lambda^*; \mu)$ in Eq. (25), we use the first-order Taylor expansion of $R_0(\hat{u}^{(i+1)}, \lambda^{(i+1)}; \mu)$ about the past iterate $(\hat{u}^{(i)}, \lambda^{(i)})$ to obtain:

$$\begin{aligned}
R_0(\hat{u}^{(i+1)}, \lambda^{(i+1)}; \mu) &= R_0(\hat{u}^{(i)}, \lambda^{(i)}; \mu) + \nabla_{\hat{u},\lambda} R_0(\hat{u}^{(i)}, \lambda^{(i)}; \mu)^\top \begin{bmatrix} \Delta \hat{u}^{(i+1)} \\ \Delta \lambda^{(i+1)} \end{bmatrix} \\
&= \nabla_{\hat{u}} L(\hat{u}^{(i)}, \lambda^{(i)}) + \nabla_{\hat{u},\lambda}(\nabla f(\hat{u}^{(i)}) + J^{(i)\top} \lambda^{(i)})^\top \begin{bmatrix} \Delta \hat{u}^{(i+1)} \\ \Delta \lambda^{(i+1)} \end{bmatrix} \\
&= \nabla_{\hat{u}} L(\hat{u}^{(i)}, \lambda^{(i)}) + \left[(\nabla^2 f(\hat{u}^{(i)}) + \nabla^2 h(\hat{u}^{(i)}) \lambda^{(i)}) \quad J^{(i)\top}\right] \begin{bmatrix} \Delta \hat{u}^{(i+1)} \\ \Delta \lambda^{(i+1)} \end{bmatrix} \\
&= \nabla_{\hat{u}} L(\hat{u}^{(i)}, \lambda^{(i)}) + \left[\nabla_{\hat{u}\hat{u}}^2 L(\hat{u}^{(i)}, \lambda^{(i)}) \quad J^{(i)\top}\right] \begin{bmatrix} \Delta \hat{u}^{(i+1)} \\ \Delta \lambda^{(i+1)} \end{bmatrix} = 0,
\end{aligned} \tag{26}$$

where $\Delta \hat{u}^{(i+1)} = \hat{u}^{(i+1)} - \hat{u}^{(i)}$, $\Delta \lambda^{(i+1)} = \lambda^{(i+1)} - \lambda^{(i)}$, and $\nabla^2 h(\hat{u}^{(i)}) \in \mathbb{R}^{n \times n \times q}$ denotes the Hessian of the constraints. Solving for the increments we obtain:

$$\left[\nabla_{\hat{u}\hat{u}}^2 L(\hat{u}^{(i)}, \lambda^{(i)}) \quad J^{(i)\top}\right] \begin{bmatrix} \Delta \hat{u}^{(i+1)} \\ \Delta \lambda^{(i+1)} \end{bmatrix} = -\nabla_{\hat{u}} L(\hat{u}^{(i)}, \lambda^{(i)}). \tag{27}$$

For the feasibility condition, i.e., the second component of $R(u^*, \lambda^*; \mu)$ in Eq. (25), we also linearize the constraint as:

$$R_1(\hat{u}^{(i+1)}, \lambda^{(i+1)}; \mu) = h(\hat{u}^{(i+1)}) = h(\hat{u}^{(i)}) + J^{(i)}(\hat{u}^{(i+1)} - \hat{u}^{(i)}) = 0, \tag{28}$$

using first-order Taylor expansion (Newton's Method). Then,

$$J^{(i)}\Delta\hat{u}^{(i+1)} = -h(\hat{u}^{(i)}). \tag{29}$$

We can then combine Eq. (27) and Eq. (29) to form the following linearized system of KKT conditions:

$$\begin{bmatrix} \nabla^2_{\hat{u}\hat{u}}L(\hat{u}^{(i)},\lambda^{(i)}) & J^{(i)\top} \\ J^{(i)} & 0 \end{bmatrix} \begin{bmatrix} \Delta\hat{u}^{(i+1)} \\ \Delta\lambda^{(i+1)} \end{bmatrix} = - \begin{bmatrix} \nabla_{\hat{u}}L(\hat{u}^{(i)},\lambda^{(i)}) \\ h(\hat{u}^{(i)}) \end{bmatrix}. \tag{30}$$

Note that the system of equations in Eq. (30) is used in Sequential Quadratic Programming (SQP) (Wilson, 1963; Nocedal & Wright, 2006; Gill & Wong, 2012) when there are no inequality constraints. In addition, since our objective $f$ is quadratic, we do not need to compute its second-order Taylor expansion, and only need to linearize the constraints. SQP reduces to Newton's Method when there are no constraints. In particular, Eq. (30) gives the standard unconstrained Newton step $\nabla^2 f(\hat{u}^{(i)})\Delta\hat{u}^{(i+1)} = -\nabla f(\hat{u}^{(i)})$ when $h = 0$.

Now, we use our quadratic objective $f$ in Problem 21 to compute:

$$\begin{aligned} \nabla_{\hat{u}}L(\hat{u}^{(i)},\lambda^{(i)}) &= Q(\hat{u}^{(i)} - \mu) + J^{(i)\top}\lambda^{(i)}, \\ \nabla^2_{\hat{u}\hat{u}}L(\hat{u}^{(i)},\lambda^{(i)}) &= Q + \nabla^2 h(\hat{u}^{(i)})\lambda^{(i)} \approx Q. \end{aligned} \tag{31}$$

Note that $Q$ is symmetric positive definite, but $\nabla^2 h(\hat{u}^{(i)})$ is not guaranteed to be positive definite in the general case, especially at every iterate, which could make the Newton step undefined. Regularization may be needed to ensure that $\nabla^2 h(\hat{u}^{(i)})$ is positive semi-definite. In addition, since $\nabla^2 h(\hat{u}^{(i)}) \in \mathbb{R}^{n \times n \times q}$ is a three-dimensional tensor, it is computationally expensive to compute this matrix of second derivatives, especially on our large-scale problem and through auto-differentiation (Griewank & Walther, 2008; Blondel & Roulet, 2024). Similar to the Gauss-Newton method (Björck, 1996; Nocedal & Wright, 2006) for nonlinear least squares problems, we assume that the constraint $h$ is approximately affine near its optimal point $u^*$, and use only first-order constraint information. Hence, we set $\nabla^2 h(\hat{u}^{(i)}) \approx 0$. We note that even with these approximations for efficiency on large-scale problems, we still show strong performance in the nonlinear constraint results in Table 4. An alternate approach could be to use a low-rank approximation to the Hessian as done in Quasi-Newton, e.g., BFGS methods (Nocedal & Wright, 2006).

Using Eq. (31) with setting $\nabla^2 h(\hat{u}^{(i)}) = 0$, Eq. (30) simplifies to:

$$\begin{bmatrix} Q & J^{(i)\top} \\ J^{(i)} & 0 \end{bmatrix} \begin{bmatrix} \Delta\hat{u}^{(i+1)} \\ \Delta\lambda^{(i+1)} \end{bmatrix} = - \begin{bmatrix} Q(\hat{u}^{(i)} - \mu) + J^{(i)\top}\lambda^{(i)} \\ h(\hat{u}^{(i)}) \end{bmatrix}. \tag{32}$$

Then,

$$Q\Delta\hat{u}^{(i+1)} + J^{(i)\top}(\lambda^{(i+1)} - \lambda^{(i)}) = -Q(\hat{u}^{(i)} - \mu) - J^{(i)\top}\lambda^{(i)}, \tag{33a}$$

$$J^{(i)}\Delta\hat{u}^{(i+1)} = -h(\hat{u}^{(i)}). \tag{33b}$$

We see that the only terms involving $\lambda^{(i)}$ cancel from both sides of the equation. Note that the method does not require tracking the dual variable, so it could also be equivalently reset to $\lambda^{(i)} = 0$ at each iteration, and we compute $\lambda^{(i+1)}$ only for computing the primal update in Eq. (22b).

Since $Q \succ 0$, it is invertible, we can multiply Eq. (33a) by $Q^{-1}$ to obtain:

$$\Delta\hat{u}^{(i+1)} = \hat{u}^{(i+1)} - \hat{u}^{(i)} = -(\hat{u}^{(i)} - \mu) - Q^{-1}J^{(i)\top}\lambda^{(i+1)}. \tag{34}$$

Multiplying both sides of Eq. (34) by $J^{(i)}$ and using Eq. (33b), we can eliminate $\Delta\hat{u}^{(i+1)}$ to obtain:

$$-h(\hat{u}^{(i)}) = -J^{(i)}(\hat{u}^{(i)} - \mu) - J^{(i)}Q^{-1}J^{(i)\top}\lambda^{(i+1)}. \tag{35}$$

Since $Q \succ 0$, $Q^{-1} \succ 0$ and then $J^{(i)}Q^{-1}J^{(i)\top} \succ 0$, and hence it is invertible. We can then solve Eq. (35) for $\lambda^{(i+1)}$ to obtain:

$$\lambda^{(i+1)} = \left(J^{(i)}Q^{-1}J^{(i)\top}\right)^{-1}\left(h(\hat{u}^{(i)}) - J^{(i)}(\hat{u}^{(i)} - \mu)\right), \tag{36}$$

which gives the desired Eq. (22a). Then, solving Eq. (34) for $\hat{u}^{(i+1)}$ gives:

$$u^{(i+1)} = \mu - Q^{-1}J^{(i)\top}\lambda^{(i+1)}, \tag{37}$$

which is the desired Eq. (22b). Lastly, for the first iterate, substituting Eq. (36) into Eq. (37), setting $i = 0$ and $u^{(0)} = \mu$ gives the desired Eq. (23).

We can also solve Eq. (32) efficiently using block Gaussian elimination and the Schur complement $(J^{(i)}Q^{-1}J^{(i)\top})^{-1} \in \mathbb{R}^{q \times q}$ (Golub & Greif, 2003). In particular, the block matrix in Eq. (32) can be factored into a product of elementary matrices as:

$$\begin{bmatrix} Q & J^{(i)\top} \\ J^{(i)} & 0 \end{bmatrix} = \begin{bmatrix} I & 0 \\ J^{(i)}Q^{-1} & I \end{bmatrix} \begin{bmatrix} Q^{-1} & 0 \\ 0 & -J^{(i)}Q^{-1}J^{(i)\top} \end{bmatrix} \begin{bmatrix} I & Q^{-1}J^{(i)\top} \\ 0 & I \end{bmatrix}. \tag{38}$$

Since the matrix factorization in Eq. (38) is a product of elementary matrices and a diagonal matrix, we can easily compute its inverse as:

$$\begin{bmatrix} Q & J^{(i)\top} \\ J^{(i)} & 0 \end{bmatrix}^{-1} = \begin{bmatrix} I & -Q^{-1}J^{(i)\top} \\ 0 & I \end{bmatrix} \begin{bmatrix} Q^{-1} & 0 \\ 0 & -(J^{(i)}Q^{-1}J^{(i)\top})^{-1} \end{bmatrix} \begin{bmatrix} I & 0 \\ -J^{(i)}Q^{-1} & I \end{bmatrix} \tag{39}$$

Then multiplying by the right-hand side in Eq. (32) gives the solution:

$$\begin{aligned}
\begin{bmatrix} \hat{u}^{(i+1)} \\ \lambda^{(i+1)} \end{bmatrix} &= \begin{bmatrix} \hat{u}^{(i)} \\ 0 \end{bmatrix} - \begin{bmatrix} I & -Q^{-1}J^{(i)\top} \\ 0 & I \end{bmatrix} \begin{bmatrix} Q^{-1} & 0 \\ 0 & -(J^{(i)}Q^{-1}J^{(i)\top})^{-1} \end{bmatrix} \begin{bmatrix} I & 0 \\ -J^{(i)}Q^{-1} & I \end{bmatrix} \begin{bmatrix} Q(\hat{u}^{(i)} - \mu) \\ h(\hat{u}^{(i)}) \end{bmatrix} \\
&= \begin{bmatrix} \hat{u}^{(i)} \\ 0 \end{bmatrix} - \begin{bmatrix} I & -Q^{-1}J^{(i)\top} \\ 0 & I \end{bmatrix} \begin{bmatrix} Q^{-1} & 0 \\ 0 & -(J^{(i)}Q^{-1}J^{(i)\top})^{-1} \end{bmatrix} \begin{bmatrix} Q(\hat{u}^{(i)} - \mu) \\ h(\hat{u}^{(i)}) - J^{(i)}(\hat{u}^{(i)} - \mu) \end{bmatrix} \\
&= \begin{bmatrix} \hat{u}^{(i)} \\ 0 \end{bmatrix} - \begin{bmatrix} I & -Q^{-1}J^{(i)\top} \\ 0 & I \end{bmatrix} \begin{bmatrix} \hat{u}^{(i)} - \mu \\ -(J^{(i)}Q^{-1}J^{(i)\top})^{-1}(h(\hat{u}^{(i)}) - J^{(i)}(\hat{u}^{(i)} - \mu)) \end{bmatrix} \\
&= \begin{bmatrix} \mu - Q^{-1}J^{(i)\top}\lambda^{(i+1)} \\ (J^{(i)}Q^{-1}J^{(i)\top})^{-1}(h(\hat{u}^{(i)}) - J^{(i)}(\hat{u}^{(i)} - \mu)) \end{bmatrix}.
\end{aligned}$$

Using the Schur complement reduces the Newton system from an indefinite $(n + q) \times (n + q)$ solve to a $n \times n$ SPD solve with $Q^{-1}$ and $q \times q$ SPD solve with the Schur complement, where $q \leq n$. Similarly, the Jacobian expression in Eq. (24), which we will show next, is obtained by implicitly differentiating the linearized KKT conditions and eliminating the dual block, which also avoids the need to invert a full $(n + q) \times (n + q)$ saddle-point or indefinite matrix.

**2. Jacobian $J_{\mathcal{T}}(\mu)$.** Here, we compute the Jacobian $J_{\mathcal{T}}(\mu) := \partial u^*(\mu)/\partial \mu$ of the transformation $\mathcal{T}(\mu) = u^*(\mu)$ using implicit differentiation. At convergence, the optimal pair $(u^*, \lambda^*)$ satisfies Eq. (19). Differentiating both sides of the first stationarity equation in Eq. (19) w.r.t. $\mu$ gives:

$$\begin{aligned}
\frac{\partial}{\partial \mu} R_0(u^*, \lambda^*; \mu) &= \frac{\partial}{\partial \mu}(Q(u^* - \mu) + \nabla h(u^*)\lambda^*) = 0, \\
&\iff (Q + \nabla^2 h(u^*)\lambda^*)\frac{\partial u^*}{\partial \mu} + \nabla h(u^*)\frac{\partial \lambda^*}{\partial \mu} = Q.
\end{aligned} \tag{40}$$

Similar to Eq. (31), we assume $h(u^*)$ is approximately affine near the optimal point, and we approximate $\nabla^2 h(u^*) \approx 0$.

Similarly differentiating both sides of the second feasibility equation in Eq. (19) w.r.t $\mu$ gives:

$$\frac{\partial}{\partial \mu} R_1(u^*, \lambda^*; \mu) = \frac{\partial}{\partial \mu}h(u^*) = \nabla h(u^*)^\top \frac{\partial u^*}{\partial \mu} = 0. \tag{41}$$

Combining Eq. (40) and Eq. (41) leads to the following block linear system:

$$\begin{bmatrix} Q & J^{*\top} \\ J^* & 0 \end{bmatrix} \begin{bmatrix} \partial u^*/\partial \mu \\ \partial \lambda^*/\partial \mu \end{bmatrix} = \begin{bmatrix} Q \\ 0 \end{bmatrix}.$$

Similar to Eq. (38), we can use the Schur complement to eliminate the dual term via block substitution. Using the block inverse in Eq. (39) with $J^{(i)} = J^*$, we have

$$\begin{bmatrix} \partial u^*/\partial \mu \\ \partial \lambda^*/\partial \mu \end{bmatrix} = \begin{bmatrix} I & -Q^{-1}J^{*\top} \\ 0 & I \end{bmatrix} \begin{bmatrix} Q^{-1} & 0 \\ 0 & -(J^*Q^{-1}J^{*\top})^{-1} \end{bmatrix} \begin{bmatrix} I & 0 \\ -J^*Q^{-1} & I \end{bmatrix} \begin{bmatrix} Q \\ 0 \end{bmatrix}$$

$$= \begin{bmatrix} I & -Q^{-1}J^{*\top} \\ 0 & I \end{bmatrix} \begin{bmatrix} Q^{-1} & 0 \\ 0 & -(J^*Q^{-1}J^{*\top})^{-1} \end{bmatrix} \begin{bmatrix} Q \\ -J^* \end{bmatrix}$$

$$= \begin{bmatrix} I & -Q^{-1}J^{*\top} \\ 0 & I \end{bmatrix} \begin{bmatrix} I \\ (J^*Q^{-1}J^{*\top})^{-1}J^* \end{bmatrix}$$

$$= \begin{bmatrix} I - Q^{-1}J^{*\top}\partial \lambda^*/\partial \mu \\ (J^*Q^{-1}J^{*\top})^{-1}J^* \end{bmatrix}.$$

Hence, the Jacobian is given by the first component as:

$$J_{\mathcal{T}}(\mu) = \frac{\partial u^*(\mu)}{\partial \mu} = I - Q^{-1}J^{*\top} \left( J^*Q^{-1}J^{*\top} \right)^{-1} J^*,$$

which is the desired Eq. (24). $\qquad\qquad\qquad\qquad\qquad\qquad\qquad\qquad\qquad\qquad\qquad\qquad\qquad$ $\square$

## C.3 (NONLINEAR) CONVEX INEQUALITY CONSTRAINTS

In this subsection, we describe how to compute the DPPL in `ProbHardE2E` for nonlinear convex inequality constraints. Convex inequality constraints arise naturally in many scientific and engineering applications. For example, total variation (TV) regularization is widely used to promote smoothness or piecewise-constant structure in spatial fields, e.g., image denoising (Rudin et al., 1992; Boyd & Vandenberghe, 2004) and total variation diminishing (TVD) constraints to avoid spurious artificial oscillations in numerical solutions to PDEs (Harten, 1997; Tezaur et al., 2017; Schein et al., 2021). Other common convex constraints include box constraints, which enforce boundedness of physical or operational quantities (Bertsekas, 1997).

We consider the constrained projection Problem (7), where $Q \succ 0$, and $h : \mathbb{R}^n \to \mathbb{R}^q$, $g : \mathbb{R}^n \to \mathbb{R}^s$ denote smooth functions representing equality and convex inequality constraints, respectively. This is a convex optimization problem due to the strictly convex quadratic objective and the assumption that $g(u)$ is convex. The associated Lagrangian is

$$L(u, \lambda, \nu; z) = \tfrac{1}{2}(u - z)^\top Q(u - z) + \lambda^\top h(u) + \nu^\top g(u),$$

with Lagrange multipliers $\lambda \in \mathbb{R}^q$ for the equality constraints and $\nu \in \mathbb{R}^s$ for the inequality constraints, where $\nu \geq 0$. The KKT optimality conditions are given as:

$$\begin{aligned} \text{(Stationarity)} \quad & Q(u^* - z) + \nabla h(u^*)\lambda^* + \nabla g(u^*)\nu^* = 0, \\ \text{(Primal feasibility)} \quad & h(u^*) = 0, \quad g(u^*) \leq 0, \\ \text{(Dual feasibility)} \quad & \nu^* \geq 0, \\ \text{(Complementary slackness)} \quad & \nu_j^* \cdot g_j(u^*) = 0 \quad \text{for all } j = 1, \ldots, s. \end{aligned} \tag{42}$$

Note the first two conditions are the same as the ones for nonlinear equality constraints with $\nu = 0$, in Eq. (19).

The KKT conditions in Eq. (42) are necessary and sufficient for optimality, under standard constraint qualifications, e.g., Slater's condition (Boyd & Vandenberghe, 2004). Eq. (42) can be solved by various optimization methods, e.g., stochastic trust-region methods with sequential quadratic programming (SQP) (Boyd & Vandenberghe, 2004; Hong et al., 2023) and exact augmented Lagrangian (Boyd & Vandenberghe, 2004; Fang et al., 2024). The augmented Lagrangian balances the need for both constraint satisfaction and computational efficiency, which makes it particularly effective in large-scale optimization problems. While the inequality constraints $g(u) \leq 0$ are convex by assumption, the equality constraints $h(u) = 0$ are typically required to be affine to ensure that the feasible set remains convex (Boyd & Vandenberghe, 2004). Nonlinear equalities generally yield non-convex level sets, which can violate problem convexity even when the objective and inequalities are convex. Although exceptions exist where nonlinear equalities define convex sets, these cases are rare and must be verified explicitly (Bertsekas, 1997; Boyd & Vandenberghe, 2004).

To compute the Jacobian $J_{\mathcal{T}}(\mu) := \partial u^*(\mu)/\partial \mu$ of the projection map with respect to the input $\mu$, we could, in principle, apply implicit differentiation to the KKT conditions in Eq. (42). For general

constrained problems with nonlinear equality and convex inequality constraints, the derivation becomes analytically complex, particularly due to active set variability and non-affine structure. In the special case of quadratic programs with affine constraints, OptNet (Amos & Kolter, 2017) provides an explicit expression for the derivatives via KKT conditions. In addition, CVXPYLayers (Agrawal et al., 2019) enables gradient-based learning for general convex cone programs by canonicalizing them into a standard conic form. In our implementation, we use CVXPYLayers to enforce the constraints during the projection step. Since CVXPYLayers does not currently support full Jacobian extraction or higher-order derivatives, we estimate the variance of the projection map using Monte Carlo methods by applying random perturbations to the inputs and computing empirical statistics over repeated forward passes.

## D  SPECIAL CASES OF PROBHARDE2E

In this section, we show applications of `ProbHardE2E` in two seemingly unrelated but technically related domains: (1) hierarchical time series forecasting with coherency constraints (Rangapuram et al., 2021; Olivares et al., 2024a); (2) solving partial differential equations (PDEs) with global conservation constraints (Hansen et al., 2023; Mouli et al., 2024). Both are special cases of `ProbHardE2E` with linear equality constraints, and orthogonal ($Q = I$) and oblique ($Q = \Sigma^{-1}$) projections, respectively. Fig. 2 illustrates the wide variety of cases that our framework covers.

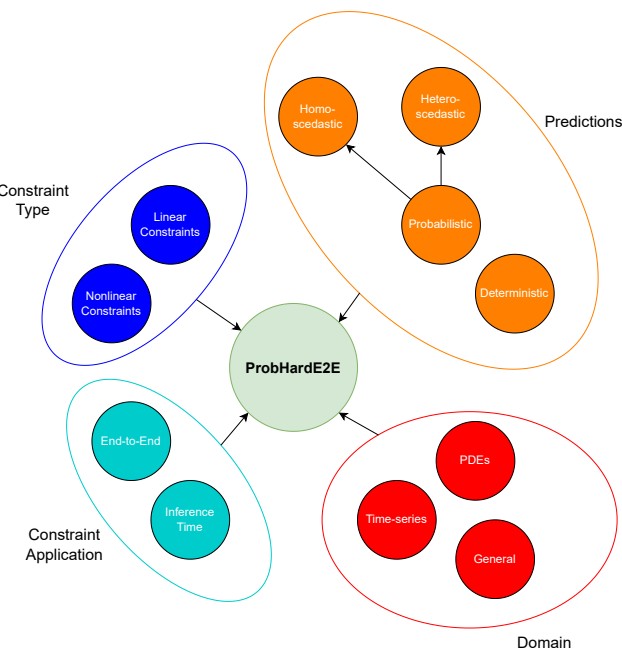

Figure 2: `ProbHardE2E` serves as a probabilistic unified framework for learning with hard constraints.

### D.1  ENFORCING COHERENCY IN HIERARCHICAL TIME SERIES FORECASTING

Hierarchical time series forecasting is abundant in several applications, e.g., retail demand forecasting and electricity forecasting. In retail demand forecasting, the sales are tracked at various granularities, including item, store, and region levels. In electricity forecasting, the consumption demand is tracked at individual and regional levels. Each time series at time $t$ can be separated into bottom and aggregate levels. Bottom-levels aggregate into higher-level series at each time point through known

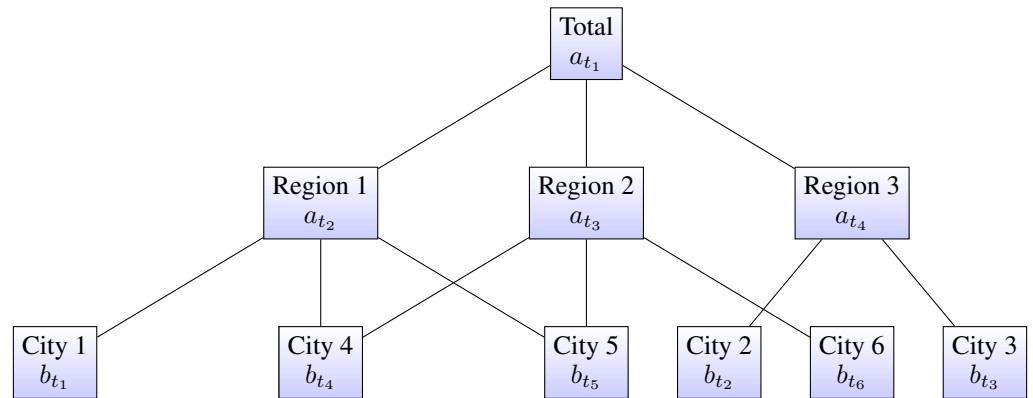

Figure 3: Example hierarchical time series structure with $a_t \in \mathbb{R}^4$, $b_t \in \mathbb{R}^6$ and $S_{\text{sum}} = \begin{bmatrix} 1 & 1 & 1 & 1 & 1 & 1 \\ 1 & 0 & 0 & 1 & 1 & 0 \\ 0 & 0 & 0 & 1 & 1 & 1 \\ 0 & 1 & 1 & 0 & 0 & 0 \end{bmatrix}$.

relationships, which can be represented as dependency graphs. Let $z_t = [a_t \quad b_t]^\top \in \mathbb{R}^n$, where $a_t \in \mathbb{R}^q$ denotes the aggregate entries, $b_t \in \mathbb{R}^{\tilde{q}}$ denotes the bottom-level entries, and $n = q + \tilde{q}$. Let $S_{\text{sum}} \in \{0,1\}^{q \times \tilde{q}}$ denote the summation matrix, which defines the relationship between the bottom and aggregate levels as $a_t = S_{\text{sum}} b_t$. This coherency constraint can be equivalently expressed as:

$$[I_q \quad -S_{\text{sum}}] \begin{bmatrix} a_t \\ b_t \end{bmatrix} = 0 \Leftrightarrow A z_t = 0, \quad \forall t, \tag{43}$$

where $I_q$ denotes the $q \times q$ identity matrix. See Hyndman et al. (2011); Rangapuram et al. (2021); Olivares et al. (2024b) and the references therein for details, and Fig. 3 for an illustration.

`HierE2E` (Rangapuram et al., 2021) enforces the coherency constraint in Eq. (43) by projecting the multivariate samples $z_t$ onto the null space of the constraint, i.e., $A z_t = 0$. It uses the following projection:

$$u^*(z_t) = \underbrace{\left(I - A^\top (A A^\top)^{-1} A\right)}_{P_I} z_t = (I - A^\dagger A) z_t, \tag{44}$$

where $A^\dagger = A^\top (A A^\top)^{-1}$ denotes the right psuedoinverse, and $P_I = P_I^2 = P_I^\top$ denotes an orthogonal projector.

We show that `HierE2E` can be formulated in our `ProbHardE2E` framework with the following posterior mean and covariance:

$$\hat{\mu}_{\text{HierE2E}} = (I - A^\dagger A)\mu, \tag{45a}$$

$$\hat{\Sigma}_{\text{HierE2E}} = \Sigma - A^\dagger A \Sigma - \Sigma A^\dagger A + A^\dagger A \Sigma A^\dagger A, \tag{45b}$$

where $P_I$ is defined in Eq. (44). In particular, we show in Proposition D.1 that the `HierE2E` posterior update in Eq. (45) is a special linear constraint case of our `ProbHardE2E` method, which uses an orthogonal projection with $Q = I$ and $b = 0$.

**Proposition D.1.** *The projected mean and covariance for `HierE2E` in Eq. (45) is given by the solution to Problem (7) with linear constraints in Proposition C.1, i.e., $h(u) = Au = 0$, $b = 0$, where $Q = I$ for an orthogonal projection and $\mathbf{Z} \sim \mathcal{N}(\mu, \Sigma)$ is a multivariate Gaussian.*

*Proof.* The oblique projection in Eq. (16) used in `ProbHardE2E` for linear constraints is given as $P_{Q^{-1}} = I - Q^{-1} A^\top (A Q^{-1} A^\top)^{-1} A$. Setting $Q = I$, the expression simplifies to $P_{Q^{-1}} = P_I = I - A^\top (A A^\top)^{-1} A = I - A^\dagger A$.

The posterior mean $\hat{\mu}$ for `ProbHardE2E` with linear constraints is given in Eq. (18a) with $b = 0$ as:

$$
\begin{aligned}
\hat{\mu} &= P_{Q^{-1}}\mu \\
&= P_I \mu \\
&= (I - A^\dagger A)\mu \\
&= \hat{\mu}_{\text{HierE2E}},
\end{aligned}
\tag{46}
$$

which is the desired expression in Eq. (45a).

Similarly, the posterior covariance $\hat{\Sigma}$ for `ProbHardE2E` in Eq. (18b) is given as:

$$
\begin{aligned}
\hat{\Sigma} &= P_{Q^{-1}}\Sigma P_{Q^{-1}}^\top \\
&= P_I \Sigma P_I^\top \\
&= P_I \Sigma P_I \\
&= (I - A^\dagger A)\Sigma(I - A^\dagger A) \\
&= (I - A^\dagger A)(\Sigma - \Sigma A^\dagger A) \\
&= \Sigma - A^\dagger A\Sigma - \Sigma A^\dagger A + A^\dagger A\Sigma A^\dagger A \\
&= \hat{\Sigma}_{\text{HierE2E}},
\end{aligned}
\tag{47}
$$

which is the desired expression in Eq. (45b). $\qquad\square$

Note that `HierE2E` does not directly project the distribution parameters, even though a closed form exists, as shown in Eq. (46) and Eq. (47). Instead, it directly projects the samples in Eq. (44). An improvement to `HierE2E` (that we do in `ProbHardE2E`) is to eliminate the computationally expensive sampling in the training loop. (See Section 3.5.) `HierE2E` samples from the parametric distribution generated by `DeepVAR` (Salinas et al., 2019; 2020; Alexandrov et al., 2019), reconciles these samples, and computes the loss over time using the Continuous Ranked Probability Score (CRPS). Generally, for unknown distributions, the CRPS evaluation requires sampling, which may explain its necessity in their framework. For many standard distributions, e.g., the multivariate Gaussian distribution in `HierE2E`, the CRPS can be computed analytically (Matheson & Winkler, 1976; Taillardat et al., 2016) using the mean and covariance of the output distribution.

## D.2 Enforcing Conservation Laws in PDEs

In addition to hierarchical forecasting, another (at first seemingly-unrelated) application of `ProbHardE2E` is enforcing conservation laws in solutions to partial differential equations (PDEs). A conservation law is given as $u_t + \nabla \cdot F(u) = 0$, for unknown $u(t, x)$ and nonlinear flux function $F(u)$ (LeVeque, 1990). Hansen et al. (2023) propose the `ProbConserv` method to enforce the integral form of conservation laws from finite volume methods (LeVeque, 2002) as a linear constraint $Au = b$ for specific problems that satisfy a boundary flux linearity assumption. In particular, `ProbConserv` proposes the following update equations for the posterior mean and covariance matrix:

$$
\hat{\mu}_{\text{ProbConserv}} = \mu - \Sigma A^\top (A\Sigma A^\top)^{-1}(A\mu - b),
\tag{48a}
$$

$$
\hat{\Sigma}_{\text{ProbConserv}} = \Sigma - \Sigma A^\top (A\Sigma A^\top)^{-1}A\Sigma,
\tag{48b}
$$

given the mean $\mu$ and the covariance matrix $\Sigma$ estimated from a black-box probabilistic model, e.g., Gaussian Process, probabilistic Neural Operators (Mouli et al., 2024) or Attentive Neural Process (ANP) (Hansen et al., 2023) or DeepVAR (Salinas et al., 2019) used in the hierarchical forecasting case.

In `ProbConserv`, the posterior mean $\hat{\mu}$ in Eq. (48a) is shown to be the solution to the constrained least squares problem:

$$
\hat{\mu}_{\text{ProbConserv}} = \underset{\substack{\tilde{\mu} \in \mathbb{R}^n \\ A\tilde{\mu} = b}}{\arg\min} \frac{1}{2}||\tilde{\mu} - \mu||^2_{\Sigma^{-1}}.
$$

We formulate this optimization problem more generally, and show that by assuming that $z \sim \mathbf{Z} \sim \mathcal{N}(\mu, \Sigma)$ is a multivariate Gaussian, a constrained sample $u^*(z) \sim \mathbf{Y} \sim \mathcal{N}(\hat{\mu}, \hat{\Sigma})$ in `ProbConserv`

is a solution to our Problem (7) with $Q = \Sigma^{-1}$ and linear constraints. In particular, we show in Proposition D.2 that the `ProbConserv` posterior update in Eq. (48) is a special linear constraint case of our `ProbHardE2E` method, which uses an oblique projection with $Q = \Sigma^{-1}$.

**Proposition D.2.** *The projected mean and covariance for* `ProbConserv` *in Eq.* (48) *is given by the solution to Problem* (7) *with linear constraints in Proposition C.1, i.e.,* $h(u) = Au - b = 0$, *where* $Q = \Sigma^{-1}$ *for an oblique projection and* $\mathbf{Z} \sim \mathcal{N}(\mu, \Sigma)$ *is a multivariate Gaussian.*

*Proof.* The oblique projection in Eq. (16) used in `ProbHardE2E` for linear constraints is given as $P_{Q^{-1}} = P_\Sigma = I - Q^{-1}A^\top(AQ^{-1}A^\top)^{-1}A$. Setting $Q = \Sigma^{-1}$, we have that $P_{Q^{-1}} = I - \Sigma A^\top(A\Sigma A^\top)^{-1}A = P_\Sigma$.

The posterior mean $\hat\mu$ for `ProbHardE2E` with linear constraints is given in Eq. (18a) as:

$$
\begin{aligned}
\hat\mu &= P_{Q^{-1}}\mu + (I - P_{Q^{-1}})A^\dagger b \\
&= (I - \Sigma A^\top(A\Sigma A^\top)^{-1}A)\mu + (I - (I - \Sigma A^\top(A\Sigma A^\top)^{-1}A))A^\dagger b \\
&= (I - \Sigma A^\top(A\Sigma A^\top)^{-1}A)\mu + \Sigma A^\top(A\Sigma A^\top)^{-1}\underbrace{AA^\dagger}_{I} b \\
&= \mu - \Sigma A^\top(A\Sigma A^\top)^{-1}(A\mu - b) \\
&= \hat\mu_{\text{ProbConserv}},
\end{aligned}
$$

which is equal to the desired expression in Eq. (48a).

Similarly, the posterior covariance $\hat\Sigma$ for `ProbHardE2E` in Eq. (18b) is given as:

$$
\begin{aligned}
\hat\Sigma &= P_{Q^{-1}}\Sigma P_{Q^{-1}}^\top \\
&= (I - \Sigma A^\top(A\Sigma A^\top)^{-1}A)\Sigma(I - A^\top(A\Sigma A^\top)^{-1}A\Sigma) \\
&= (I - \Sigma A^\top(A\Sigma A^\top)^{-1}A)(\Sigma - \Sigma A^\top(A\Sigma A^\top)^{-1}A\Sigma) \\
&= \Sigma - 2\Sigma A^\top(A\Sigma A^\top)^{-1}A\Sigma + \Sigma A^\top(A\Sigma A^\top)^{-1}(A\Sigma A^\top)(A\Sigma A^\top)^{-1}A\Sigma \\
&= \Sigma - \Sigma A^\top(A\Sigma A^\top)^{-1}A\Sigma \\
&= \hat\Sigma_{\text{ProbConserv}},
\end{aligned}
$$

which is equal to the desired expression in Eq. (48b). $\qquad\square$

Note that the projected distribution parameters in Eq. (48) are applied only at inference time in `ProbConserv`. In `ProbHardE2E`, we show the benefits of imposing the constraints at training time as well in an end-to-end manner.

## E  FLEXIBILITY IN THE CHOICE OF $Q$ AND ITS STRUCTURE

In this section, we discuss the modeling choices for the projection matrix $Q$ in our DPPL, which defines the energy norm in the objective in the constrained least squares Problem (7). Its specification significantly influences both the learning dynamics and the inductive biases of the model. Selecting or learning $Q$ offers a principled mechanism to reflect the statistical structure of the data, particularly in settings involving multivariate regression or heteroscedastic noise (Kendall et al., 2018; Stirn et al., 2023). Table 6 summarizes common structure choices for $Q$ and their trade-offs. Of course, in many applications, there is a single goal for the choice of Q—to optimize accuracy.

In practice, the space of symmetric positive definite (SPD) matrices is too large to be explored (and "learned") without additional structure, especially in high-dimensional settings. To address this, structural constraints are often imposed on $Q$, reducing the number of parameters, and acting as a form of regularization (Willette et al., 2021). These structures encode modeling assumptions, e.g., output independence, sparsity, or low-rank correlations, and they trade off statistical expressivity against computational efficiency.

In many cases, the choice of $Q$ (or the form of $Q$) should ideally reflect (knowledge or assumptions or hope about) the structure of the underlying data distribution. The simplest choice, $Q = I$,

| Structure of $Q$ | Example Form | Merits and Demerits |
|---|---|---|
| **Identity** | $Q = I$ | + Simplest choice, no parameters
+ Strong regularization
– Ignores uncertainty and correlations |
| **Diagonal (learned)** | $Q = \mathrm{diag}(q_1, \dots, q_n)$ | + Captures heteroscedasticity
+ Efficient to compute and invert
– Ignores correlations |
| **Low-rank (learned $L$)** | $Q = LL^\top, L \in \mathbb{R}^{n \times d}$ | + Captures dominant correlations
+ Fewer parameters than full
– Still computationally involved |
| **Full (learned $L$)** | $Q = LL^\top, L \in \mathbb{R}^{n \times n}$ | + Fully expressive
– High memory and compute cost
– Prone to overfitting |

Table 6: Several structure choices for the matrix $Q$ and their associated trade-offs.

assumes isotropy across output dimensions, and is often used for its regularization benefits and ease of implementation. This choice neglects any correlation structure in the data, and it tends to perform poorly in the presence of strong heteroscedasticity. A diagonal matrix $Q = \mathrm{diag}(q_1, \dots, q_n)$ introduces per-dimension weighting, and is well-suited to heteroscedastic tasks where the variance differs across outputs (Kendall & Gal, 2017; Skafte et al., 2019). Low-rank approximations provide a compromise between model complexity and expressivity, by capturing dominant correlation directions (Willette et al., 2021). Full-rank matrices allow flexibility and often require strong priors or large datasets to avoid overfitting (Weinberger & Saul, 2009).

We focus on two concrete realizations of $Q$: the identity matrix $Q = I$ that is used in the `HierE2E` (Rangapuram et al., 2021) (see Appendix D.1), and a diagonal matrix defined as the inverse of a predicted diagonal covariance, $Q = \Sigma^{-1}$ that is used in `ProbConserv` (Hansen et al., 2023) (see Appendix D.2), where $\Sigma = \mathrm{diag}(\sigma_1^2, \dots, \sigma_d^2)$ denote the empirical variances output by the model. This latter choice corresponds to a heteroscedastic formulation that scales residuals based on their predicted precision, which emphasizes more confident predictions, and down-weights less certain ones (Stirn et al., 2023; Le et al., 2005; Hansen et al., 2023).

## F  PROOF OF THEOREM 3.1

In this section, we begin by first restating Theorem 3.1, which provides a closed-form update for our DPPL in Eq. (8) for a prior distribution that belongs to a multivariate local-scale family of distributions; and then we provide its proof.

**Theorem 3.1.** *Let $\mathbf{Z} \sim \mathcal{F}(\mu, \Sigma)$ be a random variable, where the underlying distribution $\mathcal{F}$ belongs to a multivariate location-scale family of distributions, with mean $\mu$ and covariance $\Sigma$; and let $\mathcal{T}$ be a function with continuous first derivatives, such that $J_{\mathcal{T}}(\mu) \Sigma J_{\mathcal{T}}(\mu)^\top$ is symmetric positive semi-definite. Then, the transformed distribution $\mathbf{Y} = \mathcal{T}(\mathbf{Z})$ converges in distribution with first-order accuracy to $\mathcal{F}(\hat{\mu}, \hat{\Sigma})$ with mean $\hat{\mu} = \mathcal{T}(\mu)$ and covariance $\hat{\Sigma} = J_{\mathcal{T}}(\mu) \Sigma J_{\mathcal{T}}(\mu)^\top$, where $J_{\mathcal{T}}(\mu) = \nabla \mathcal{T}(\mu)^\top$ denotes the Jacobian of $\mathcal{T}$ with respect to $z$ evaluated at $\mu$.*

*Proof.* Recall that a family of probability distributions is said to be a location-scale family if for any random variable $\mathbf{Z}$ whose distribution belongs to the family $\mathbf{Z} \sim \mathcal{F}(\mu, \Sigma)$, then there exists a transformation (re-parameterization) of the form

$$\mathbf{Y} \stackrel{d}{=} A\mathbf{Z} + B,$$

where $A$ denotes a scale transformation matrix, $B$ denotes the location parameter, and $\stackrel{d}{=}$ denotes equality in distribution.

Let $\mathbf{Y} = \mathcal{T}(\mathbf{Z})$ be a nonlinear transformation. We calculate the first-order Taylor series expansion to linearize the function about the mean $\mu$ as:

$$\mathbf{Y} = \mathcal{T}(\mathbf{Z}) \approx \mathcal{T}(\mu) + J_{\mathcal{T}}(\mu)(\mathbf{Z} - \mu) \tag{49}$$
$$= \underbrace{J_{\mathcal{T}}(\mu)\mathbf{Z}}_{A} + \underbrace{(\mathcal{T}(\mu) - J_{\mathcal{T}}(\mu)\mu)}_{B}.$$

Then, since $\mathbf{Z}$ belongs to the location-scale family of distributions, the linearization of $\mathbf{Y} \sim \mathcal{F}(\hat{\mu}, \hat{\Sigma})$ also belongs to the family with mean $\hat{\mu}$ and covariance $\hat{\Sigma}$, which we compute below.

Taking the expectation of both sides of Eq. (49) we get:

$$\begin{aligned}
\hat{\mu} = \mathbb{E}[\mathcal{T}(\mathbf{Z})] &\approx \mathbb{E}[\mathcal{T}(\mu) + J_{\mathcal{T}}(\mu)(\mathbf{Z} - \mu)] \\
&= \mathbb{E}[\mathcal{T}(\mu)] + \mathbb{E}[J_{\mathcal{T}}(\mu)(\mathbf{Z} - \mu)] \text{ (by linearity of expectation)} \\
&= \mathcal{T}(\mu) + J_{\mathcal{T}}(\mu) \underbrace{(\mathbb{E}[\mathbf{Z}] - \mu)}_{0} \text{ (since } \mu \text{ is not a random variable)} \\
&= \mathcal{T}(\mu). 
\end{aligned} \tag{50}$$

Then, the covariance $\hat{\Sigma}$ is given as:

$$\begin{aligned}
\hat{\Sigma} &= \mathbb{E}[(\mathcal{T}(\mathbf{Z}) - \mathbb{E}[\mathcal{T}(\mathbf{Z})])(\mathcal{T}(\mathbf{Z}) - \mathbb{E}[\mathcal{T}(\mathbf{Z})])^{\top}] \\
&= \mathbb{E}[(\mathcal{T}(\mathbf{Z}) - \mathcal{T}(\mu))(\mathcal{T}(\mathbf{Z}) - \mathcal{T}(\mu))^{\top}] \text{ (by Eq. (50))} \\
&\approx \mathbb{E}[(\mathcal{T}(\mu) + J_{\mathcal{T}}(\mu)(\mathbf{Z} - \mu) - \mathcal{T}(\mu))(\mathcal{T}(\mu) + J_{\mathcal{T}}(\mu)(\mathbf{Z} - \mu) - \mathcal{T}(\mu))^{\top}] \text{ (by Eq. (49))} \\
&= \mathbb{E}[(J_{\mathcal{T}}(\mu)(\mathbf{Z} - \mu))(J_{\mathcal{T}}(\mu)(\mathbf{Z} - \mu))^{\top}] \\
&= J_{\mathcal{T}}(\mu)\mathbb{E}[(\mathbf{Z} - \mu)(\mathbf{Z} - \mu)^{\top}]J_{\mathcal{T}}(\mu)^{\top} \\
&= J_{\mathcal{T}}(\mu)\Sigma J_{\mathcal{T}}(\mu)^{T}.
\end{aligned}$$

$\square$

Importantly, the approximation error between the nonlinear transformation and its linearization converges to zero in probability (Van der Vaart, 2000), which ensures the validity of this approach asymptotically. We note that this result is closely related to the Multivariate Delta Method (Casella & Berger, 2001), which shows that for a nonlinear function $\mathcal{T}$, the sample mean of $\mathcal{T}(z_1, \ldots, z_n)$ also converges in distribution, under mild conditions. Specifically, if the sample mean of $n$ i.i.d. draws from $\mathbf{Z}$ converges to a multivariate Gaussian (by the CLT), then the same linearization argument and Slutsky's theorem imply that the sample mean of the projected samples converges to a multivariate Gaussian, with parameters given in Eq. (8). Second-order approximations (via a quadratic expansion of $\mathcal{T}$) yield higher-order corrections, and can lead to non-Gaussian outcomes (e.g., chi-squared) (Casella & Berger, 2001).

## G BENCHMARKING DATASETS

In this section, we detail the benchmarking datasets in both applications domains, i.e., PDEs and probabilistic time series forecasting.

### G.1 PDEs

We consider a series of conservative PDEs with varying levels of difficulties, where the goal is to learn an approximation of the solution that satisfies known conservation laws. We follow the empirical evaluation protocol from Hansen et al. (2023). The PDEs we study are conservation laws, which take the following differential form:

$$u_t + \nabla \cdot F(u) = 0, \tag{51}$$

for some nonlinear flux function $F(u)$. These equations can be written in their conservative form as:

$$\frac{d}{dt} \int_{\Omega} u(t, x)d\Omega = F(u(t, x_0)) - F(u(t, x_N)), \tag{52}$$

by applying the divergence term in 1D over the domain $\Omega = [x_0, x_N]$ (LeVeque, 1990; Hansen et al., 2023). This global conservation law states that the rate of change of total mass or energy in this system is given by the difference of the flux into the domain and the flux out of the domain. Note that in higher dimensions, the flux difference on the right-hand side of Eq. (52) can be written as a surface integral along the boundary of the domain. This conservative form is at the heart of numerical finite volume methods (LeVeque, 2002), which discretize the domain into control volumes and solve this equation locally in each control volume, to enforce local conservation, i.e., so that the flux into a control volume is equal to the flux out of it. In the following, we summarize the PDE test cases with their initial and boundary conditions, exact solutions, and derived linear conservation constraints from Hansen et al. (2023).

### G.1.1 GENERALIZED POROUS MEDIUM EQUATION (GPME)

The Generalized Porous Medium Equation (GPME) is given by the following degenerate parabolic PDE:

$$u_t - \nabla \cdot (k(u)\nabla u) = 0, \tag{53}$$

where the flux in Eq. (51) is given as $F(u) = -k(u)\nabla u$, and $k(u)$ denotes the diffusivity parameter. This diffusivity parameter $k(u)$ may depend nonlinearly and/or discontinuously on the solution $u$. We consider three representative cases within the GPME family, by changing this parameter $k(u)$. Each instance of the GPME increases in difficulty based on the regularity of the solution and the presence of shocks or discontinuities.

**Heat Equation ("Easy").** The classical parabolic heat equation arises when the diffusivity is constant, i.e., $k(u) = k$ in Eq. (53). We use the heat equation with the following sinusoidal initial condition and periodic boundary conditions from Krishnapriyan et al. (2021); Hansen et al. (2023):

$$\begin{aligned}
u_t &= k\Delta u, &\forall x \in \Omega = [0, 2\pi], \ \forall t \in [0, 1], \\
u(0, x) &= \sin(x), &\forall x \in [0, 2\pi], \\
u(t, 0) &= u(2\pi, t), &\forall t \in [0, 1],
\end{aligned} \tag{54}$$

respectively. The exact solution, which can be solved using the Fourier Transform, is given as:

$$u_{\text{exact}}(t, x) = e^{-kt}\sin(x).$$

The solution is a smooth sinusoidal curve that exponentially decays or dissipates over time, and has an infinite speed of propagation. With these specific initial and boundary conditions in Eq. (54), the global conservation law in Eq. (52) reduces to the following linear equation:

$$\int_0^{2\pi} u(t, x)\, dx = 0, \quad \forall t \in [0, 1], \tag{55}$$

since the net flux on the boundaries is 0.

**Porous Medium Equation (PME) ("Medium").** The PME is a nonlinear degenerate subclass of the GPME, where the diffusivity is a nonlinear, monomial of the solution, i.e., $k(u) = u^m$ in Eq. (53). It has been using in modeling nonlinear heat transfer (Vázquez, 2007; Maddix et al., 2018a). We use the PME with the following initial condition and growing in time left Dirichlet boundary condition from Lipnikov et al. (2016); Maddix et al. (2018a); Hansen et al. (2023):

$$\begin{aligned}
u_t - \nabla \cdot (u^m \nabla u) &= 0, &\forall x \in \Omega = [0, 1], \ \forall t \in [0, 1], \\
u(0, x) &= 0, &\forall x \in [0, 1], \\
u(t, 0) &= (mt)^{1/m}, &\forall t \in [0, 1].
\end{aligned} \tag{56}$$

The exact solution is given as:

$$u_{\text{exact}}(t, x) = (m\,\text{ReLU}(t - x))^{1/m}.$$

For small values of $k(u)$, this degenerate parabolic equation behaves hyperbolic in nature. The solution exhibits a sharp front at the degeneracy point $t = x$ with a finite speed of propagation. With these specific initial and boundary conditions in Eq. (56), the global conservation law in Eq. (52) reduces to the following linear equation:

$$\int_0^1 u(t, x)\, dx = \frac{(mt)^{1+1/m}}{m + 1}, \quad \forall t \in [0, 1]. \tag{57}$$

**Stefan Equation ("Hard").** The Stefan equation has been used in foam modeling (van der Meer et al., 2016) and crystallization (Sethian & Strain, 1992), and models phase transitions with the following discontinuous diffusivity:

$$k(u) = \begin{cases} 1, & u \geq u^* \\ 0, & u < u^* \end{cases}, \quad u^* \geq 0,$$

in Eq. (53). We use the Stefan equation with the following initial condition and Dirichlet boundary conditions from Maddix et al. (2018b); Hansen et al. (2023):

$$\begin{aligned} u_t - \nabla \cdot (k(u)\nabla u) &= 0, & \forall x \in \Omega = [0,1],\ t \in [0,1], \\ u(0,x) &= 0, & \forall x \in [0,1], \\ u(t,0) &= 1, & \forall t \in [0,1]. \end{aligned} \tag{58}$$

The exact solution is given as:

$$u_{\text{exact}}(t,x) = \mathbf{1}_{u \geq u^*} \left[ 1 - \frac{1-u^*}{\text{erf}(\alpha/2)} \text{erf}\left(\frac{x}{2\sqrt{t}}\right) \right],$$

where $\mathbf{1}$ denotes the indicator function, $\text{erf}(z) = (2/\sqrt{\pi})\int_0^z \exp(-y^2)dy$ denotes the error function, and $\alpha = 2\tilde{\alpha}$ and $\tilde{\alpha}$ satisfies the following nonlinear equation:

$$\frac{1-u^*}{\sqrt{\pi}} = u^* \text{erf}(\tilde{\alpha})\tilde{\alpha}e^{\tilde{\alpha}^2}.$$

The solution is a rightward moving shock. With these specific initial and boundary conditions in Eq. (58), the global conservation law in Eq. (52) reduces to the following linear equation:

$$\int_0^1 u(t,x)\,dx = \frac{2(1-u^*)}{\text{erf}(\alpha/2)}\sqrt{\frac{t}{\pi}}, \quad \forall t \in [0,1]. \tag{59}$$

### G.1.2 HYPERBOLIC LINEAR ADVECTION EQUATION

The hyperbolic linear advection equation models fluids transported at a constant velocity, and is given by Eq. (51) with linear flux $F(u) = \beta u$. We use the 1D linear advection problem with the following step-function initial condition and inflow Dirichlet boundary conditions from Hansen et al. (2023):

$$\begin{aligned} u_t + \beta u_x &= 0, & \forall x \in \Omega = [0,1], \forall t \in [0,1], \\ u(0,x) &= \mathbf{1}_{x \leq 0.5}, & \forall x \in [0,1], \\ u(t,0) &= 1, & \forall t \in [0,1]. \end{aligned} \tag{60}$$

The exact solution is given as:

$$u(x,t) = h(x - \beta t),$$

where $h(x) = \mathbf{1}_{x \leq 0.5}$ denotes the initial condition. The solution remains a shock, which travels to the right with a finite speed of propagation $\beta$. With these specific initial and boundary conditions in Eq. (60), the global conservation law in Eq. (52) reduces to the following linear equation:

$$\int_0^1 u(x,t)\,dx = \frac{1}{2} + \beta t, \tag{61}$$

which shows that the total mass increases linearly with time due to the fixed inflow.

### G.2 PROBABILISTIC TIME SERIES FORECASTING

In addition to PDEs, we also evaluate `ProbHardE2E` on five hierarchical time series forecasting benchmark datasets, where the goal is to generate probabilistic predictions that are coherent with known aggregation constraints across cross-sectional hierarchies (Rangapuram et al., 2021).

Table 7 provides an overview of the time series datasets used in our empirical evaluation. For each benchmarking dataset, it details the total number of series, the number of bottom level series (i.e., the leaf nodes in the hierarchy), the number of series aggregated from the bottom-level series, the depth

of the hierarchy in terms of the number of levels, the number of time series observations, and the prediction horizon $\tau$.

We adopt the same dataset configurations as in Rangapuram et al. (2021), from which we use the hierarchical forecasting benchmarks and pre-processing pipeline. These datasets are available in GluonTS package (Alexandrov et al., 2019). The LABOUR dataset (Australian Bureau of Statistics, 2019) contains monthly Australian employment statistics from 1978 to 2020, organized into a 57-series hierarchy. The TRAFFIC dataset (Ben Taieb & Koo, 2019) includes sub-hourly freeway lane occupancy data, aggregated into daily observations forming a 207-series structure. TOURISM (Tourism Australia, Canberra, 2005) consists of quarterly tourism counts across 89 Australian regions (1998–2006), and the extended TOURISM-L dataset (Wickramasuriya et al., 2019) comprises 555 grouped series based on both geography and travel purpose. Lastly, WIKI contains daily page view counts from 199 Wikipedia pages collected over two years (Anava et al., 2018).

Table 7: A summary of the time-series datasets. TOURISM-L has two hierarchies, defined by geography and travel purpose; consequently, it has different numbers of bottom series and different depths in each hierarchy.

| Dataset | Total | Bottom | Aggregated | Levels | Obs. | Horizon $\tau$ | Frequency |
|---------|-------|--------|------------|--------|------|----------------|-----------|
| TOURISM | 89 | 56 | 33 | 4 | 36 | 8 | Quarterly |
| TOURISM-L | 555 | 76; 304 | 175 | 4; 5 | 228 | 12 | Monthly |
| LABOUR | 57 | 32 | 25 | 4 | 514 | 8 | Monthly |
| TRAFFIC | 207 | 200 | 7 | 4 | 366 | 1 | Daily |
| WIKI | 199 | 150 | 49 | 5 | 366 | 1 | Daily |

## H IMPLEMENTATION DETAILS

In this section, we provide the implementation details of `ProbHardE2E`. Fig. 4 illustrates the overall pipeline of `ProbHardE2E`, which integrates probabilistic modeling, constraint enforcement, and loss-based calibration into a unified differentiable architecture. The core contribution lies in the DPPL, which acts as a "corrector" to the "predictor," which is the unconstrained distribution predicted by a wide class of models. Conceptually, this layer parallels classical predictor-corrector and primal-dual methods from numerical optimization (Boyd & Vandenberghe, 2004; Bertsekas, 1997), where a candidate solution is refined to satisfy known constraints before evaluation.

We evaluate `ProbHardE2E` on two scientific domains: (1) PDEs, where structured physical constraints, e.g., conservation laws and boundary conditions, must be enforced (see Appendix G.1), and (2) probabilistic hierarchical time series forecasting, where aggregation coherency is required (see Appendix G.2). We show that `ProbHardE2E` is model-agnostic by using a base probabilistic model (predictor) from each application domain, i.e., `VarianceNO` (Mouli et al., 2024) for PDEs and `DeepVAR` (Salinas et al., 2019) for forecasting. We then enforce the corresponding constraint with our DPPL (corrector). We provide the experimental details for each application in the following subsections.

### H.1 PDEs

All the experiments are performed on a single NVIDIA V100 GPU. We use a probabilistic Fourier Neural Operator (FNO) (Li et al., 2021), i.e., `VarianceNO` (Mouli et al., 2024) to learn a mapping from PDE parameters to solutions, e.g., the diffusivity mapping $k(u) \mapsto u(t, x)$ in the (degenerate) parabolic Generalized Porous Medium Equation (GPME), or the velocity mapping $\beta \mapsto u(t, x)$ in the hyperbolic linear advection equation. (See Appendix G.1 for details on the datasets.)

#### H.1.1 DATASET GENERATION

Table 8 provides an overview of the PDE data generation. For each PDE in Appendix G.1, we generate a dataset of $N = 200$ parameter-solution pairs $\{\phi^{(i)}, u^{(i)}\}_{i=1}^{N} \sim \mathcal{D}$, where $\phi^{(i)}$ denotes the input PDE parameters, e.g., $k, m, u^*, \beta$, and $u^{(i)}$ denotes the corresponding spatiotemporal solution

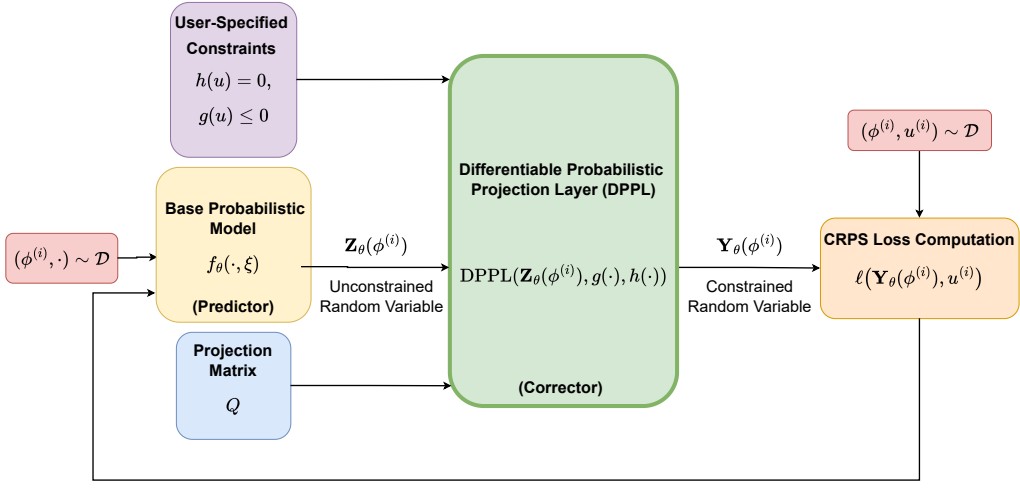

Figure 4: Schematic representation of `ProbHardE2E` (see Algorithm 1). Here, a known pathwise-differentiable probabilistic model is chosen to predict a (unconstrained) prior distribution. (Optionally, the projection matrix can be specified as a part of the prediction from the probabilistic model or modeled separately.) Next, we transform the distribution with our DPPL to obtain the transformed distribution, done empirically or via the Delta Method (see Section 3.3), which enforces the constraints. Lastly, we choose an appropriate loss function, e.g., CRPS, to calibrate the transformed distribution with the target variable.

field. Each solution $u^{(i)}(t, x)$ is simulated over a grid of 100 equidistant points in both space and time, yielding a total of $100 \times 100$ observations per instance. During evaluation, we predict the final 20 equidistant time slices while conditioning on the earlier time steps.

Table 8: Overview of PDE dataset generation. Each dataset contains 200 samples with a fixed 160/40 train-test split.

| PDE | Parameter range | Spatial domain | Time domain | Train/Test (%) |
|---|---|---|---|---|
| Heat | $k \in [1, 5]$ | $[0, 2\pi]$ | $[0, 1]$ | 80/20 |
| PME | $m \in [2, 3]$ | $[0, 1]$ | $[0, 1]$ | 80/20 |
| Stefan | $u^* \in [0.6, 0.65]$ | $[0, 1]$ | $[0, 1]$ | 80/20 |
| Linear Advection | $\beta \in [1, 2]$ | $[0, 1]$ | $[0, 1]$ | 80/20 |

### H.1.2 ARCHITECTURAL DETAILS

We use `VarianceNO` (Mouli et al., 2024) as our base unconstrained probabilistic model. `VarianceNO` is an augmented Fourier Neural Operator (FNO) (Li et al., 2021) that updates the last layer to output two prediction heads instead of one, i.e, one for the mean and the other for the variance of the multivariate Gaussian distribution. Table 9 details the model hyperparameters.

### H.1.3 TRAINING AND TESTING SETUP

We follow the standard training procedure for FNO-based models as proposed by Li et al. (2021). Specifically, we use the Adam optimizer (Kingma & Ba, 2015) with weight decay and train using mini-batches of fixed size $B = 20$. A step-based learning rate scheduler is applied, which reduces the learning rate by half every 50 epochs. During evaluation, we uniformly sample parameters from the specified parameter ranges in Table 8 to construct test sets and compute the evaluation metrics.

Table 9: Hyperparameters for the `VarianceNO` model.

| Hyperparameter | Values |
|---|---|
| `VarianceNO` | |
|     Number of Fourier layers | 4 |
|     Channel width | $\{32, 64\}$ |
|     Number of Fourier modes | 12 |
|     Batch size | 20 |
|     Learning rate | $\{10^{-4}, 10^{-3}, 10^{-2}\}$ |

## H.2 PROBABILISTIC TIME SERIES FORECASTING

The experiments are performed on an Intel(R) Xeon(R) CPU E5-2603 v4 @ 1.70GHz.

### H.2.1 DATASET GENERATION

We adopt the hierarchical forecasting benchmarks and preprocessing pipeline introduced in Ranga-puram et al. (2021), using five standard datasets: LABOUR, TRAFFIC, TOURISM, TOURISM-L, and WIKI. Each dataset contains a hierarchy of time series with varying depth and number of aggregation levels (see Table 7). The train/test splits, seasonal resolutions, and prediction horizons follow the standardized setup provided in Rangapuram et al. (2021).

### H.2.2 ARCHITECTURAL DETAILS

We use `DeepVAR` (Salinas et al., 2019) as our base unconstrained probabilistic model, which is aligned with `Hier-E2E`. `DeepVAR` is a probabilistic autoregressive LSTM-based model that leverages a multivariate Gaussian distribution assumption on the multivariate target. `DeepVAR` models the joint dynamics of all the time series in the hierarchy through latent temporal dependencies, and outputs both the mean and scale of the predictive distribution, by optimizing the negative log likelihood (NLL). Our `ProbHardE2E` model in the time series application is developed based on `Hier-E2E` in GluonTS (Alexandrov et al., 2019). We use the default base model architecture `DeepVAR`, and make further modifications to `Hier-E2E`. Specifically, we tune the hyperparameters in Table 10, and adjust the loss to CRPS for structured probabilistic evaluation. We disable sampling-based projection during training because `ProbHardE2E` optimizes the closed-from CRPS for Gaussian distributions, and our projection methodology ensures that linear constraints are met probabilistically. During inference, we report CRPS through samples, in order to align the evaluation definition with the various hierarchical forecasting baselines.

Table 10: Key hyperparameters for `DeepVAR` across hierarchical forecasting datasets.

| Dataset | Epochs | Batch Size | Learning Rate | Context Length | No. of Prediction Samples |
|---|---|---|---|---|---|
| LABOUR | 5 | 32 | 0.01 | 24 | 400 |
| TRAFFIC | 10 | 32 | 0.001 | 40 | 400 |
| TOURISM | 10 | 32 | 0.01 | 24 | 200 |
| TOURISM-L | 10 | 4 | 0.001 | 36 | 200 |
| WIKI | 25 | 32 | 0.001 | 15 | 200 |

### H.2.3 TRAINING AND TESTING SETUP

We follow the standard GluonTS (Alexandrov et al., 2019) training setup using the Adam opti-mizer (Kingma & Ba, 2015) and mini-batch updates. Each epoch consists of 50 batches, with batch size set according to Table 10. We run our evaluation five times and report the mean and variance of the CRPS values in Table 3.

Unlike `Hier-E2E` (Rangapuram et al., 2021), which samples forecast trajectories during training and projects them to ensure structural coherence on samples, our method operates entirely in the parameter space during training. We avoid sampling and instead minimize the closed-form CRPS

loss (Gneiting et al., 2005) directly on the predicted mean and variance. This makes the training process sampling-free and reduces training time, similar to the PDE case discussed later in Figure 5. This key distinction avoids the use of `coherent_train_samples`, as described in the Appendix of Rangapuram et al. (2021).

At inference time, because the reported CRPS is computed on the samples in the hierarchical baselines, we enable structured projection by drawing predicted samples from the learned distribution, and we apply our DPPL to ensure that they satisfy the hierarchical aggregation constraints. This setup parallels the `coherent_pred_samples` mode in `HierE2E`, and we implement the inference step with this approach for experimentation simplicity. Table 10 shows the number of prediction samples in evaluation to compute the CRPS and calibration metrics over the projected outputs. Alternatively, we can also evaluate the CRPS using samples from the projected distribution.

### H.3 METRICS

We evaluate `ProbHardE2E` and the various baselines using the following metrics. We denote the exact solution or ground truth observations as $u$, and we report the metrics on the mean $\hat{\mu}$, covariance $\hat{\Sigma}$, and samples $\{u_i^*\}_{i=1}^N$ drawn from the constrained multivariate Gaussian distribution $\mathcal{N}(\hat{\mu}, \hat{\Sigma})$.

**Mean Squared Error (MSE).** The MSE measures the mean prediction accuracy and is given as:

$$\text{MSE}(\hat{\mu}) = \frac{1}{n} \|u - \hat{\mu}\|_F^2,$$

where the Frobenius norm is taken over all the datapoints $n$ in $\hat{\mu}$.

**Constraint Error (CE).** The CE measures the error in the various equality constraints $h(u^*) = 0$, i.e., conservation laws for PDEs and coherency for hierarchical time series forecasting, on the samples, and is given as:

$$\text{CE}(u^*) = \sum_{i=1}^N \|h(u_i^*)\|_2^2,$$

where we compute the average error over $N = 100$ samples $\{u_i^*\}_{i=1}^N$.

**Continuous Ranked Probability Score (CRPS).** The CRPS (Gneiting & Raftery, 2007) measures the quality of uncertainty quantification by comparing a predictive distribution to a ground-truth observation. For a multivariate Gaussian distribution with independent components $\mathcal{N}(\mu, \text{diag}(\hat{\sigma}^2))$, where $\hat{\sigma}_{ii}^2$ denotes the $i$-th diagonal entry of the predictive covariance $\hat{\Sigma}$, the CRPS is given in closed-form as:

$$\text{CRPS}_{\mathcal{N}}(\hat{\mu}, \hat{\sigma}; u) = \sum_{i=1}^n \hat{\sigma}_{ii} \left[ z_i \left(2P(z_i) - 1\right) + 2p(z_i) - \frac{1}{\sqrt{\pi}} \right],$$

where $z_i = (u_i - \hat{\mu}_i)/\hat{\sigma}_{ii}$, $p(z_i) = (1/\sqrt{2\pi}) \exp(-z_i^2/2)$ denotes the standard normal PDF, and $P(z_i) = \int_{-\infty}^{z_i} p(y) dy$ denotes the standard normal CDF (Gneiting et al., 2005; Taillardat et al., 2016).

## I ADDITIONAL EMPIRICAL RESULTS AND DETAILS

In this section, we include additional empirical results and details for `ProbHardE2E` with various constraint types, i.e., linear equality, nonlinear equality and convex inequality.

### I.1 LINEAR EQUALITY CONSTRAINTS

In this subsection, we show additional results and details for `ProbHardE2E` with linear equality constraints in both PDEs and hierarchical time series forecasting.

### I.1.1 PDEs with Conservation Law Constraints

Here, we impose the discretized form of the simplified linear global conservation laws given in Appendix G.1 for the heat equation (Eq. (55)), PME (Eq. (57)), Stefan (Eq. (59)) and linear advection equation (Eq. (61)). We use the trapezoidal discretizations of the integrals from Hansen et al. (2023).

Fig. 5 shows the analogous training time per epoch to Fig. 1(a) for PDE datasets. Models trained with 100 posterior samples per training step incur a 3.5–3.6× increase in epoch time relative to our `ProbHardE2E`, which avoids sampling altogether by using a closed-form CRPS loss. See Table 2 for the accuracy results.

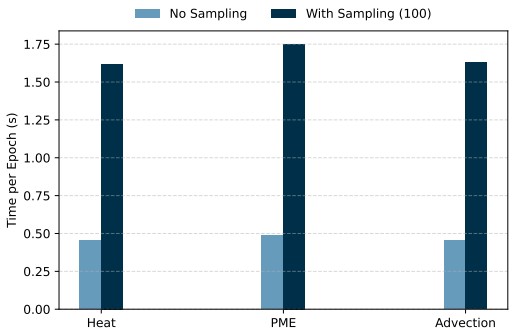

Figure 5: `ProbHardE2E`: PDE timing comparisons for our sampling-free approach.

### I.1.2 Hierarchical Time Series Forecasting with Coherency Constraints

Here, we test `ProbHardE2E` on probabilistic hierarchical forecasting with coherency constraints. (See Appendix G.2 for details and Table 3 for the results.) We compare the two variants of `ProbHardE2E`, i.e., with oblique $Q = \Sigma^{-1}$ (`ProbHardE2E-Ob`) and with orthogonal $Q = I$ (`ProbHardE2E-Or`) projection to the following baselines:

- `DeepVAR` (Salinas et al., 2019) is the base unconstrained probabilistic model, which assumes a multivariate Gaussian distribution for $\mathbf{Z} \sim \mathcal{N}(\mu, \Sigma)$ with mean $\mu$ and diagonal covariance $\Sigma$.
- `Hier-E2E` (Rangapuram et al., 2021) uses `DeepVAR` as the base model, and enforces the exact coherency constraint by applying the orthogonal projection directly on the samples in an end-to-end manner. Another difference from their approach is that we use the closed-form CRPS expression rather than the approximate weighted quantile loss.
- `ProbConserv` (Hansen et al., 2023) enforces the coherency constraint as an oblique projection at inference time only.
- `ARIMA-NaiveBU` and `ETS-NaiveBU` are two simple baseline models that use ARIMA and exponential smoothing (ETS), respectively. These methods use a naive bottom-up approach of deriving aggregated level forecasts (Hyndman & Athanasopoulos, 2018).
- `PERMBU-MINT` (Taieb et al., 2017) is a hierarchical probabilistic forecasting model that is based on a linear projection method MINT (Wickramasuriya et al., 2019). It generates probabilistic forecasts for aggregated series using permuted bottom-level forecasts.

We do not include `DPMN` (Olivares et al., 2024a) or `CLOVER` (Olivares et al., 2024b) in our experiments because the implementations are proprietary. Given that `Hier-E2E` is the best open-access hierarchical forecasting model, through GluonTS (Alexandrov et al., 2019), to the best of our knowledge, we use the same base model to `Hier-E2E` (i.e., `DeepVAR`), and we evaluate forecast accuracy compared to `Hier-E2E` to assess the added value of our `ProbHardE2E`.

### I.2 Nonlinear Equality Constraints

In this subsection, we impose the discretized form of the general nonlinear global linear conservation laws from Eq. (52) in Appendix G.1 for the PME with various ranges for the parameter $m$. (See Table 4 for the results and Fig. 1(b) for the solution profile.) For the PME, the flux in Eq. (52) is

given as $F(u) = -k\nabla u$, where $k(u) = u^m$. Substituting this flux into Eq. (52) and integrating in time gives the general conservation law for the PME as:

$$\int_\Omega u(t,x)d\Omega = \int_0^t [u^m(t,x_0)\nabla u(t,x_0) - u^m(t,x_N)\nabla u(t,x_N)]dt, \quad \forall t \in [0,1].$$

Similar to the linear equality constraint case, we discretize the integral using the trapezoidal rule. Unlike `ProbConserv` (Hansen et al., 2023), which requires an analytical flux expression to evaluate the right-hand side, our `ProbHardE2E` can enforce arbitrary (nonlinear) conservation laws directly. In addition, `ProbHardE2E` with nonlinear constraints can be applied to arbitrary PDEs with any initial or boundary conditions. We impose the initial and boundary conditions as additional linear constraints and enforce positivity on the solution. We test on various training and testing ranges for the parameter $m$, i.e., $m \in [2,3]$, $[3,4]$ and $[4,5]$. As the exponent $m$ is increased, the degeneracy increases, and as a result the solution becomes sharper and more challenging to solve (Maddix et al., 2018a; Hansen et al., 2023). We see in Table 4 that across all values of $m$, either our oblique `ProbHardE2E-Ob` or orthogonal projection `ProbHardE2E-Or` variants of our method perform better than all the baselines.

### I.3 (NONLINEAR) CONVEX INEQUALITY CONSTRAINTS

In this subsection, we impose a convex total variation diminishing (TVD) inequality constraint. (See Fig. 1(c) for the solution profile.) TVD numerical schemes have been commonly using in solving hyperbolic conservation laws with shocks to minimize numerical oscillations from dispersion (Harten, 1997; LeVeque, 1990; Tezaur et al., 2017). The total variation (TV) is defined in its continuous form as:

$$\text{TV}(u(t,\cdot)) = \int_\Omega \left|\frac{\partial u}{\partial x}\right| d\Omega.$$

This integral can be approximated as the discrete form of the total variation (TV) used in image processing as:

$$\text{TV}(u(t)) = \text{TV}(u(t,\cdot,)) = \sum_{i=1}^{N_x} |u(t,x_{i+1}) - u(t,x_i)|, \tag{62}$$

where we discretize the spatial domain $\Omega = [x_1, \ldots, x_{N_x}]$ into $N_x$ gridpoints. A numerical scheme is called TVD if:

$$\text{TV}(u(t_{n+1})) \leq \text{TV}(u(t_n)), \quad \forall\, n = 1, \ldots, N_t, \tag{63}$$

where we discretize the temporal domain $[0,T] = [t_1, \ldots, t_{N_t}]$ into $N_t$ gridpoints, and TV denotes the discretized form defined in Eq. (62).

The TVD constraint in Eq. (63) is a nonlinear inequality constraint, and enforcing it as a hard constraint is challenging with current frameworks, e.g., DCL (Agrawal et al., 2019). To address this, we perform a convex relaxation of the constraint by imposing:

$$\text{TVD} = \sum_{n=1}^{N_t} \sum_{i=1}^{N_x} |u(t_n,x_{i+1}) - u(t_n,x_i)|,$$

as a regularization term. This approach is analogous to total variation denoising in signal processing (Rudin et al., 1992; Boyd & Vandenberghe, 2004).

Fig. 1(c) demonstrates the application of the modified TVD constraint, resulting in more physically-meaningful solutions by decreasing both the artificial oscillations and probability of negative samples, which violate the monotonicity and positivity properties of the true solution, respectively. In addition, `ProbHardE2E` leads to improved (tighter) uncertainty estimates.

## J THE USE OF LARGE LANGUAGE MODELS (LLMS)

LLMs are used for grammatical corrections and minor formatting of the paper. They are not used in the conceptualization and implementation of research.

