# OpenReview forum: "End-to-End Probabilistic Framework for Learning with Hard Constraints"
_ICLR.cc/2026/Conference — ICLR 2026 Poster_

### Official Review · Reviewer_MxEb · 2025-10-23

**Soundness:** 3
**Presentation:** 2
**Contribution:** 3
**Rating:** 6
**Confidence:** 3

**Summary:**

The paper presents *ProbHardE2E*, an end-to-end probabilistic prediction framework that incorporates hard constraints. The approach first trains an unconstrained probabilistic backbone using the CRPS, chosen for its robustness and existence properties compared to the NLL. The main contribution is the *differentiable probabilistic projection layer (DDPL)*, which projects unconstrained predictions onto constrained probabilistic outputs (mean and covariance) during training and onto corresponding random samples during inference.

For a location–scale predictive family, the authors derive closed-form solutions of the push-forward through the optimal constrained objective under linear, nonlinear, and convex constraints, allowing sample-free training. The DDPL thus produces uncertainty-aware predictions that satisfy hard constraints. Numerical experiments across time-series and PDE tasks demonstrate improved accuracy and uncertainty metrics over several baselines.

Overall, this is a well-written and valuable paper with a clear contribution, sound theoretical derivations, and strong empirical validation. I have a few main comments that, if clarified, could further strengthen the paper. Given satisfactory responses, I would be willing to raise my score.

**Strengths:**

The paper provides an interesting combination of probabilistic learning and optimization under hard constraints. To my best judgement, the author have a good grasp of the theory and provide detailed proofs for their theoretical claims. I particularly enjoyed the idea of analyzing the push-forward distribution of the constrained optimization problem and showing ways to derive the anyltical solution for a location-scale family.
The numerical experiments support the author's claims, showing improved performance while fulfilling the hard constraints.

**Weaknesses:**

I believe, the presentation and readability could be improved, and some details of the method might need further clarification, see questions.

**Questions:**

## CRPS notation
Equation (2) uses the CRPS from Gneiting & Raftery (2007):
$$
\mathrm{CRPS}(Y,y)=\tfrac12\mathbb{E}|Y-Y'|-\mathbb{E}|Y-y|.
$$
However, this corresponds to a proper scoring rules that is defined such that $S(Q,Q)\ge S(P,Q)$, i.e., the true distribution maximizes the score. This conflicts with the minimization objectives in (3) and (4).
Since the closed-form in §3.5 is derived for minimization, as opposed to (2), the sign in (2) should be flipped for consistency. This is likely a notational issue and easily corrected.

---

## Notation in §3.1
The notation in §3.1 is somewhat inconsistent.
You define $\mathcal{Y} \subset \mathbb{R}^{k}$ as the space of predicted distribution parameters,(l. 104), but later (l. 111) use $\mathcal{Y}$ for the space of constrained distribution samples.
Moreover, ${\mathbf{Y}\_{\theta}} \in \mathbb{R}^{n}$ is introduced as a realization in $\mathbb{R}^{n}$, which is a notation conflict with the parameter space $\mathcal{Y}$. The way I understood it, $\mathbf{Y}\_{\theta}$ is supposed to be a random variable.

However, with the formulation $\mathbf{Y}_{\theta}(\phi^{(i)})$ it appears like a functional, although it is probably meant as the conditional predictive distribution given $\phi^{(i)}$. Clarity would improve by reserving distinct symbols: e.g., use $\Theta$ for the parameter space, $\Phi$ for network parameters, and define all variables upon first use. This would make it explicit that the network outputs distribution parameters, which then define a predictive measure from which realizations can be sampled.


---

## Framing of log-likelihood and CRPS
Line 19 states that likelihood-based objectives are “heavily biased by their distributional assumptions.” This phrasing feels too strong since the proposed approach also assumes a distributional form (e.g., location-scale or Gaussian), for which the log-likelihood would be an equally valid objective. The CRPS is indeed advantageous, especially under misspecification, but I suggest softening the claim in the abstract and §3.1 or framing it in terms of the observed empirical performance benefits.
Additionally, it could be interesting to evaluate *ProbHardE2E* with alternative proper scoring rules—such as the energy or kernel scores (Gneiting & Raftery 2007)—as you did for NLL in Q3. Showing that the approach generalizes to various scores would reinforce its flexibility.

---

## Discussion of unique solutions
A brief discussion on the *existence and uniqueness* of the constrained optimization problem (eqs 1 and 7) would enhance completeness.
If the feasible set is empty, the algorithm naturally terminates; but if the solution is unique, uncertainty vanishes and the probabilistic framework offers no benefit.
For instance, with linear equalities where $q=n$ and $A$ is full rank, the projection $P_{Q^{-1}}=0$ yields the deterministic solution
$$
\hat{\mu}=A^{\dagger}b,\quad \Sigma=0,
$$
which is a degenerate distribution and technically not part of the underlying distributional family anymore (maybe depending a bit on the chosen definition). Certainly, in this case uncertainty quantification is not possible and the approach offers no advantage against non-probabilistic solvers. While this is not a limitation of the method, clarifying that *ProbHardE2E* mainly targets under-determined $q<n$ or non-unique settings would improve conceptual precision.
More generally, for any constraint, the transformation $\mathcal{T}$ projects the distribution onto the subspace induced by the constraints and $\Sigma$ does in general not have full rank, i.e. the underlying distribution is degenerate (by design of course). Maybe it would be worth to mention and/or analyze some implication of this, for example that uncertainty cannot be (meaningfully) analyzed across the whole space of the initially learned distribution, but only across the subspace of the transformed distribution.


---

## Minor comments
- $\theta$ appears in l. 106 before being defined.
- The domains of $\phi^{(i)}$ and $u^{(i)}$ (l. 106) are missing.
- The spaces of $\xi$ and $z_\theta$ (l. 168–169) should be specified.

---

## Reference
Gneiting, T., & Raftery, A. E. (2007). *Strictly Proper Scoring Rules, Prediction, and Estimation.* **JASA**, 102 (477), 359–378.
<https://doi.org/10.1198/016214506000001437>

---

> ### Author Response · Authors · 2025-11-21
>
> We thank the reviewer for the careful reading and constructive feedback. We are especially grateful for the positive assessment that our work is “a well-written and valuable paper with a clear contribution, sound theoretical derivations, and strong empirical validation.” We also appreciate the comment that the analysis of the push-forward distribution and the derivation of analytic solutions for location–scale families were particularly effective. We address the reviewer’s comments and questions below.
>
> We appreciate the reviewer’s thorough analysis and attention to notation and presentation. We have updated these in the revised manuscript and address your questions below.
>
> ### CRPS notation
>
> We thank the reviewer for catching this and agree that the sign in equation (2) should be flipped for consistency, as we refer to the minimization of CRPS. We have updated this in our revised manuscript.
>
> ### Notation in §3.1
>
> Yes, $\mathcal{Y}$ denotes the space of constrained distribution parameters, which is the output of the neural network $f\_{\theta}$, where $\theta \in \Theta$ denotes the network parameters and $\phi^{(i)} \in \Phi $ denotes the input training data. Our notation $\mathbf{Y}\_{\theta}(\phi^{(i)})$ denotes the conditional predictive distribution given $\phi^{(i)}$. You are also correct that $\mathbf{Y}_{\theta}$ is a random variable that is the output of the reparameterization function $r(\hat{f}\_\theta(\phi^{(i)}), \xi)$ in Eqn. 6, for noise $\xi \sim p(\xi) \in \mathbb{R}^n$. Any constrained sample drawn from $\mathbf{Y}\_{\theta}$ is in $\mathbb{R}^n$. We have clarified this and corrected the notation in Section 3.1 of the revised version.
>
>
> ### Framing of log-likelihood and CRPS
>
> We agree that our motivation to use CRPS is for its advantageous properties, especially under misspecification. We have rephrased this sentence in the abstract and Section 3.1 of the revised version to focus more on these advantages and observed empirical gains. The past baseline methods, e.g., ProbConserv and Operator-ProbConserv in the PDEs community, had been using NLL, and we empirically showed in our work the benefit of training with CRPS.
>
> Yes, it is a very interesting direction to test ProbHardE2E with other proper scoring rules, e.g., energy or kernel scores, and leave this as an exciting direction for future work.
>
> ### Discussion of unique solutions
>
> We agree that ProbHardE2E with linear constraints is designed for underdetermined systems.  On lines 243-244, we stated that the $q \times n$ matrix needs to be full row-rank $q$, where $q \le n$. We have clarified that ProbHardE2E applies to underdetermined systems when $q < n$ in the revised version. In addition, we have added a discussion on the uniqueness and degeneration to point predictions when $q=n$ to the end of Appendix C.1. In the case of $q = n$, other baselines, e.g., HardC from ProbConserv, have not projected the covariance. We have experimental results with this HardC baseline in Table 2, which show that it mainly performs worse than ProbHardE2E-Or or ProbHardE2E-Ob.
>
> ### Minor comments
>
> We appreciate the reviewers’ comment to help improve the notational clarity. We have clarified these notations and domains of the variables in our revised version.
>
> We hope we have addressed your comments and that you will consider raising your score, as mentioned.

---

> > ### Comment · Reviewer_MxEb · 2025-11-26
> >
> > Thank you for addressing my comments and incorporating the suggested notation improvements into the revised submission. The authors have fully resolved the minor issues (CRPS notation, missing mathematical spaces) and have reformulated parts of the manuscript in line with our discussion (uniqueness of linear systems, framing of log-likelihood vs. CRPS).
> > All points have therefore been satisfactorily addressed, and I consider this a strong and well-suited contribution for the venue. Therefore, I raise my score from 6 to 8 and recommend acceptance of the paper.

---

> > > ### Author Response · Authors · 2025-11-27
> > >
> > > We thank the reviewer for their continued positive perception and support of the paper,  highlighting it being a "strong and well-suited contribution for the venue". We are glad that our responses satisfactorily addressed your concerns, and your consideration in increasing our rating of the paper. Your comments and questions were helpful in improving the quality and thoroughness of the paper.  Thank you again for your meticulous review and constructive feedback.

---

### Official Review · Reviewer_971x · 2025-10-26

**Soundness:** 4
**Presentation:** 3
**Contribution:** 4
**Rating:** 8
**Confidence:** 3

**Summary:**

The authors present ProbHardE2E, a probabilistic forecasting framework that incorporates hard operational and physical constraints while providing uncertainty quantification. Their methodology introduces a novel differentiable probabilistic projection layer (DPPL) that can be integrated with a wide range of neural network architectures. The DPPL enables the model to learn the system in an end-to-end manner, in contrast to other approaches where constraints are enforced only through post-processing or during inference.

ProbHardE2E optimizes a strictly proper scoring rule without making any distributional assumptions about the target, allowing it to produce robust distributional estimates. This stands in contrast to existing likelihood-based methods, which are often biased by their distributional assumptions and model choices. Moreover, the framework can incorporate a variety of non-linear constraints, thereby enhancing modeling power and flexibility.

The authors demonstrate the effectiveness of ProbHardE2E in learning partial differential equations with uncertainty estimates and in probabilistic time-series forecasting, establishing it as a broadly applicable general framework that bridges these seemingly disparate domains.

**Strengths:**

1. The proposed method uses strictly proper scoring rules (e.g., CRPS) instead of log-likelihood objectives, reducing the learning bias caused by incorrect distributional assumptions.
2. The training process can be sample-free, offering potential efficiency advantages.
3. In both time-series forecasting and PDE-solving tasks, the proposed method demonstrates strong empirical performance.

**Weaknesses:**

1. During inference, the authors indicate that a projection must be computed at every step. Does this imply that the computational cost during inference could be substantial? Could the authors provide an analysis of the time complexity for both training and inference? Similarly, although the training process is sampling-free, repeatedly computing the Jacobian matrix can also increase computational time, especially when dealing with high-dimensional data or scenarios involving multiple constraints.

**Questions:**

please refer to the weakness

---

> ### Author Response · Authors · 2025-11-21
>
> We thank the reviewer for recognizing the novelty of our “broadly applicable general” framework that “bridges seemingly disparate domains.” We are grateful that the reviewer highlighted the contribution of our differentiable probabilistic projection layer (DPPL), the benefits of optimizing strictly proper scoring rules such as CRPS, and the advantages of our sample-free training procedure. We address your questions below.
>
>
> ### Inference cost
>
> We agree with the reviewer’s interpretation that, indeed, during inference, the projection must be computed at every step. However, the computational cost is not as large as incorporating sampling during training, where backpropagation through many samples introduces substantial compute and memory overhead. This design is intentional: ProbHardE2E keeps DPPL training sampling-free for efficiency, while at inference DPPL projects samples to ensure strict feasibility, which is important in safety-critical applications. Certain structures (e.g., linear constraints) further alleviate this cost, and we explicitly derive closed-form expressions for these cases. Solving a nonlinear system via Newton’s method typically requires $(O(n^3))$ per iteration, while linear constraints reduce this to $(O(n^2))$. All projections are parallelized across batch dimensions. We are not aware of any work that models arbitrary nonlinear constraints as hard requirements while being computationally more efficient.
>
> ### Training cost and comparison to sampling
>
> We agree that computing the Jacobian matrix increases computational time. In our case, we compute the Jacobian using reverse-mode automatic differentiation, which is significantly faster than finite-difference or forward-mode approaches. By design, we automatically handle multiple constraints; even for nonlinear PME we use IC, BC, and conservation constraints together, forming a full row-rank $q \times n$ structure ($q < n$), where reverse-mode AD requires only $q$ evaluations. The computational complexity matches that of the projection itself: $O(n^3)$ for nonlinear constraints and $O(n^2)$ for linear constraints. The additional memory cost for backpropagating through the linearized KKT system is approximately $O(n^2)$ to store intermediate matrices. For our Gauss-Newton case, the linear solve is cheaper at only $O(q^3)$. (See Appendix C.2 for details).
>
> By contrast, sampling-based UQ methods incur $O(Sq^3)$ computation per iteration, where $S$ (often 50–200) denotes the number of samples, since each sample requires both projection and differentiation. They also require storing and backpropagating through all sample paths, incurring $O(S,n^2)$ memory. Avoiding this multiplicative sampling factor is exactly why DPPL training is kept sampling-free. We further parallelize Jacobian computation using vectorized maps. Algorithms such as Jacobian-free Newton–Krylov methods [1], which rely only on Jacobian-vector products, could further reduce compute and memory costs; we leave this for future work.
>
> ### Empirical timing
>
> We provide timing comparisons of sampling vs. non-sampling approaches during training and inference in Figures 1(a) (time series) and Figure 5 (PDEs), which show that removing the $(O(S))$ sampling factor yields meaningful speedups of approximately **3–5×** while still enforcing constraints strictly.
>
>
> ### References
> [1] Knoll, Dana A., and David E. Keyes. "Jacobian-free Newton–Krylov methods: a survey of approaches and applications." *Journal of Computational Physics* 193.2 (2004): 357–397.

---

> > ### Comment · Reviewer_971x · 2025-11-28
> >
> > The authors have effectively addressed my concerns regarding both inference and training costs, and have provided a theoretical comparison and thorough analysis of the experimental results. Therefore, I will maintain my positive score and recommend accepting this paper.

---

### Official Review · Reviewer_8rdZ · 2025-10-30

**Soundness:** 3
**Presentation:** 3
**Contribution:** 3
**Rating:** 6
**Confidence:** 4

**Summary:**

The paper proposes ProbHardE2E, a unified probabilistic framework for learning under hard constraints with uncertainty quantification. The core module DPPL turns an unconstrained probabilistic predictor (e.g., DeepVAR for time series or VarianceNO for PDEs) into a constraint-satisfying probabilistic model in an end-to-end pipeline. Training is sample-free: the model predicts location–scale parameters $(\mu, \Sigma)$ and DPPL projects these to constrained parameters $(\hat{\mu}, \hat{\Sigma})$ via a delta-method linearization of a projection map $T$, and the paper optimizes CRPS on the constrained distribution. At inference, they sample from the prior and exactly project each sample by solving a constrained least-squares problem to enforce feasibility. The approach aims to handle linear equalities, nonlinear equalities, and convex inequalities within one recipe.

**Strengths:**

- The method is clearly formulated in a principled “predictor–corrector” view.
- The authors provide an exact handling for linear constraints.
- The training objective is close-formed and sampling free.

**Weaknesses:**

- First-order DPPL approximation with no error bounds (risk under strong nonlinearity/variance).
- Loss/evaluation assume independence while the projected posterior is generally correlated.
- Practical experiments lock $Q$ to diagonal/identity, limiting the benefits of oblique projections and potentially biasing outcomes.
- Nonlinear equality projection claims “strict feasibility,” but non-convexity and active-set issues are largely handled numerically without theoretical or robustness analysis.
- Inequality-constraint metrics are thin (CE defined for equalities), so support for those claims is more qualitative than quantitative.

**Questions:**

1. Can you provide error bounds (or at least empirical error studies) for the delta-method approximation vs exact sampling across varying nonlinearity/variance.
2. Can you report multivariate calibration checks (e.g., energy score/logS), not just diagonal-CRPS, given the fact that you are doing multivariate forecasting, where CRPS is not strictly proper.
3. Include ablations on $Q$ (identity vs diagonal vs low-rank) with runtime/accuracy trade-offs, since $Q$ defines the projection geometry.
4. The claim "We demonstrate the importance of using a strictly proper scoring rule for evaluating probabilistic predictions, e.g., the CRPS, rather than negative log-likelihood (NLL)." is completely wrong, NLL is strictly proper.

---

> ### Author Response · Authors · 2025-11-21
> **Author's Response Part (1/3)**
>
> We thank the reviewer for the detailed and insightful evaluation. We appreciate the recognition of our method as a principled “predictor–corrector” framework, the acknowledgment of our “exact handling for linear constraints,” and the appreciation of our “closed-form, sampling-free training objective.” We are also grateful that the reviewer highlighted the clarity of our formulation and the generality of DPPL across linear, nonlinear, and convex constraints. We address the reviewer’s questions and concerns point-by-point below.
>
> ### Linearization of KKT Conditions
>
> Yes, our method uses a linearization of the KKT conditions, similar to approaches in nonlinear optimization, e.g., Sequential Quadratic Programming (SQP) and Gauss-Newton [1]. We understand the concern of first-order approximation of the Jacobian and the estimation errors, and that they are only locally accurate. We emphasize that our method does not rely solely on linearization. The delta method is used **only during training**, where it provides closed-form gradients for location-scale families  and avoid sampling variance, which enables stable optimization and **significant speedups** (**3–5×** faster for linear constraints, as shown in Figure 1a). At inference, we do not linearize but instead solve a constrained optimization problem for each sample via a Newton solver, enforcing hard constraints in their exact (nonlinear) form, which is **crucial in strongly nonlinear regimes**.  Our use of the delta method is a deliberate model design decision that balances training speed with accuracy.
>
> To appease these concerns, we have added additional experiments in Table A below on PME with $m \in [2,3]$ that directly project the samples and avoid our linear approximation with the delta method during training. We see that ProbHardE2E is approximately **9-45x** faster for approximately the same accuracy. We see that while the accuracy of the sampling approach does increase with more samples $S$, the runtime increases linearly with $S$, and it is already significantly higher than that for our ProbHardE2E. This trade-off between computational efficiency and approximation accuracy is particularly important in high-dimensional spatio-temporal scientific ML problems, where training time is a bottleneck.
>
> ### Table A: Performance comparison on the PME with nonlinear conservation law constraint for ProbHardE2E versus sampling-based baselines with 10 and 50 samples.
>
> | PME m ∈ [2,3] | Metric | ProbHardE2E | Sampling-10 | Sampling-50 |
> |---------------|--------|-------------|-------------|-------------|
> |               | MSE    | **5.04E-06**    | 1.35E-02    | 2.93E-05    |
> |               | CE     | 0           | 0           | 0           |
> |               | CRPS   | **8.648E-04**   | 5.01E-02    | 2.03E-03    |
> |               | Time   | **1.54 s**      | 14.17 s     | 70.65 s     |
>
> In terms of theoretical guarantees, Theorem 3.1, whose proof is in Appendix F, shows that the approximation error between the nonlinear transformation and its linearization converges to zero in probability [2]. This linearization facilitates efficient training via closed-form CRPS gradients.
>
>
> In practice, our DPPL can be used with any optimization method, including more advanced ones as a forward pass and generating backward pass via implicit differentiation. We specifically chose to implement the Gauss-Newton Method mainly due to our requirements in the framework, e.g., differentiability and batching support. Naive differentiation through optimization programs can result in substantially long training times [7]. We use implicit differentiation to write a custom backward pass, with a forward pass that could use any method to obtain the feasible point. We find that explicit batching and implicit differentiation for optimization methods with nonlinear equality constraints are only available in a limited and fragmented manner within the PyTorch ecosystem, or the tools themselves available for implementation within the ecosystem are not yet mature. (See Appendix C.2 for details). Methods, e.g., augmented Lagrangian and second-order (Hessian) approaches may have better convergence and are good directions for future work. In addition, while second-order methods, e.g., BFGS [1], which use a higher second-order Taylor expansions may improve fidelity, they require the Hessian computation, which significantly increases the computational and memory complexity. We discuss this in Appendix F, and have mentioned this in the conclusion as a direction of future work.

---

> ### Author Response · Authors · 2025-11-21
> **Author's Response Part (2/3)**
>
> ### Diagonal vs. Low-Rank $Q$ Matrix and Energy-Score Metric
> We clarify that our use of a diagonal covariance is a **deliberate and common design choice** in scalable probabilistic modeling frameworks. Notably, prior works, e.g., **ProbConserv**, **Operator-ProbConserv** and **HierE2E** employ diagonal Σ and still achieve strong performance on constrained forecasting and simulation tasks. In our setting, the diagonal assumption allows efficient training and inference in high-dimensional PDE regimes, where full Σ estimation would be prohibitively expensive.
>
> Moreover, we highlight in **Appendix E (Table 6\)** that diagonal Σ can serve as an **implicit regularizer**, which avoids overfitting in data-sparse regimes. We also discuss the practical trade-offs: while full or low-rank covariance may increase expressiveness, it introduces substantial computational and memory overhead, and may not translate into improved constraint satisfaction or UQ in practice.
>
> We have added an additional experiment in Table B below, which compares training with a diagonal to low-rank $Q$ matrix, where the rank $r=100$. In this experiment, we also report the Energy Score, as suggested, as another metric for multivariate distributions. We see that on some more challenging problems with shocks, e.g., linear advection and Stefan, there are benefits to the low-rank covariance. We plan to further explore structured approximations in future work with low-rank and dense parameterizations, as mentioned in the conclusion section.
> ### Table B: PDE performance with diagonal vs. low-rank $Q$ matrix.
>
> | Dataset         | Metric       | Diagonal       | Low-rank  |
> |-----------------|--------------|----------------|----------------|
> | **Heat**        | MSE          | **5.1532e-07**     | 4.6246e-06     |
> |                 | Energy Score | **0.0063**         | 0.0180         |
> | **PME**         | MSE          | **5.3510e-05**     | 0.0001         |
> |                 | Energy Score | **0.1562**         | 0.2123         |
> | **Linear Adv.** | MSE          | 0.0020         | **0.0007**         |
> |                 | Energy Score | 0.8699         | **0.5367**         |
> | **Stefan**      | MSE          | 0.0030         | **0.0019**         |
> |                 | Energy Score | 0.5629         | **0.4567**         |
>
>
> ### Feasibility of the Constraints in Nonlinear Optimization
> We thank the reviewer for highlighting the challenges associated with non-convexity and active-set behavior. These issues are well known in the nonlinear optimization literature, and our setting inherits the same characteristics. As in classical nonlinear programming, feasibility and convergence depend on local curvature, initialization, and conditioning of the KKT system: properties that also arise in widely used methods such as IPOPT, line-search, and trust-region Newton methods, and other constrained Newton-type solvers [1].
> In our implementation, DPPL calls a standard constrained optimizer inside the forward pass. This aligns with the literature on differentiable optimization layers [2,3,4], where the forward pass solves a nonlinear program and the backward pass differentiates the KKT system. Because DPPL does not depend on a specific solver, any classical nonlinear constrained method (interior-point, SQP, line-search Newton, etc.) can be used interchangeably, and the theoretical behavior follows directly from the established analysis in that literature. Even implementing these advanced optimization methods for nonlinear constraints that support batching, GPU, and differentiability is an active research area, and we have developed an implementation that works suitably for our use cases. (See Appendix C.2 for details).
>
> Importantly, in our applications, the constraints arise from physics- or operator-derived structure that is already implicitly satisfied by the data, and they are smooth and low-dimensional. As a result, we did not observe convergence failures or active-set instability in practice.
>
> Finally, we note that a full theoretical analysis for general non-convex constraints is an open problem in the broader optimization-as-layers community and is an interesting direction for future work. We do provide universal approximation guarantees under convex constraints in Appendix B, which offers formal grounding in the settings, where such analysis is tractable.
>
>
> ### Inequality Constraints
> The TVD inequality constraint is used to reduce the Gibb's phenomenon at shocks, i.e., the artificial oscillations as done in numerical methods  for hyperbolic conservation laws [5]. As a result, we do see narrower uncertainty intervals and importantly a smaller probability of drawing non-physical negative samples compared to the other approaches. In addition, the maximum and minimum of the uncertainty interval is decreased with our TVD constraint, which reduces the magnitude of the oscillations, as designed.

---

> ### Author Response · Authors · 2025-11-21
> **Author's Response (3/3)**
>
> ### NLL Clarifications
>
> We agree with the reviewer that the NLL is a strictly proper scoring rule, and have rephrased that sentence in the revised version. To clarify, we purposely propose to use the CRPS since it has some better properties, e.g., it is robust to probabilistic model misspecification. This is echoed by **Reviewer 971x**, who states our main motivation that with the CRPS “ProbHardE2E optimizes a strictly proper scoring rule without making any distributional assumptions about the target, allowing it to produce robust distributional estimates. This stands in contrast to existing likelihood-based methods, which are often biased by their distributional assumptions and model choices.” While it is true that CRPS and NLL are proper scoring rules, CRPS has theoretical advantages over NLL under distributional miscalibration [6]. This does not automatically imply superior performance in constrained generative settings. For instance, NLL can still perform well even if the predictive distribution is sharp and well-calibrated, even under hard constraints.
>
> Therefore, we do not claim that CRPS is universally superior; rather, we empirically validate that CRPS leads to better constraint satisfaction and predictive accuracy in our constrained settings. CRPS has been commonly used in the time series literature [7], and is especially true in scientific ML applications (e.g., PDE forecasting), where the feasibility of individual samples is critical. Our experiments in Table 2 confirm this advantage of the CRPS objective (See our Q3).
>
> ### References
> [1] Bertsekas, Dimitri P. "Nonlinear programming." Journal of the Operational Research Society 48.3 (1997): 334-334.
>
> [2] Amos, Brandon, and J. Zico Kolter. "Optnet: Differentiable optimization as a layer in neural networks." International conference on machine learning. PMLR, 2017.
>
> [3] Pineda, Luis, et al. "Theseus: A library for differentiable nonlinear optimization." Advances in Neural Information Processing Systems 35 (2022): 3801-3818.
>
> [4] Agrawal, Akshay, et al. "Differentiable convex optimization layers." Advances in neural information processing systems 32 (2019).
>
> [5] LeVeque, Randall J. Finite volume methods for hyperbolic problems. Vol. 31. Cambridge university press, 2002.
>
> [6] Gneiting, Tilmann, and Raftery, Adrian E.. "Strictly proper scoring rules, prediction, and estimation." Journal of the American statistical Association 102.477 (2007): 359-378.
>
> [7] Hyndman, Rob J., and George Athanasopoulos. Forecasting: principles and practice. OTexts, 2018.

---

> ### Comment · Reviewer_8rdZ · 2025-11-25
>
> Thank you for addressing my concerns. The authors fully addressed my concerns over the risk around the delta-method approximation and optimization behavior. My concerns over inequality-constraint claims, global robustness guarantees, and multivariate calibration are partially resolved. I maintain my positive view on this paper.

---

> > ### Author Response · Authors · 2025-11-26
> >
> > We thank the reviewer for the follow-up. We are glad that our revisions satisfactorily addressed the core concerns, and we appreciate the reviewer's positive assessment of the paper. Thank you again for the careful reading and constructive feedback.

---

### Official Review · Reviewer_LXk5 · 2025-10-31

**Soundness:** 3
**Presentation:** 4
**Contribution:** 3
**Rating:** 6
**Confidence:** 4

**Summary:**

This paper proposes an end-to-end probabilistic framework that strictly enforces hard constraints during training while producing full predictive distributions (mean + covariance) for uncertainty quantification (UQ). The technical core is the Differentiable Probabilistic Projection Layer (DPPL), which projects an unconstrained probabilistic prediction onto the constraint manifold. For equality/linear constraints, the projection/transformation has closed forms. For nonlinear/convex constraints, the DPPL layer uses Newton/KKT iterations to compute the projection and implicit differentiation (linearized KKT systems) to get the Jacobian for covariance propogation. Then, using CRPS and its closed forms for common local-scale families, it further enables sample-free and end-to-end training, which is faster compared to the sampling-based UQ baselines. The method is shown to imporve forecasting/UQ metrics on several benchmarks in hierarchical time series forecasting and PDE solving problems.

**Strengths:**

1. The main nolvety of this paper is to propose an end-to-end enforcement of hard constraints, where constraints are optimized as part of learning, not only enforced at post-processing or inference as in convention.

2. It also enables joint UQ and constraint satisfaction, thus suitable for safety/physics/constrained engineering tasks.

3. The sample-free & closed-form CRPS, combined with analytic projection propagations gives substantial training speedups.

4. Generality: the framework covers linear equalities, nonlinear equalities and convex inequalities (with iterative solvers for the nonlinear/convex cases). In addition, it clearly demonstrates how the method works in each scenario with examples and application areas provided.

**Weaknesses:**

1. The major concern is that the method relies on first-order approximations to linearize the nonlinear function transformation, the KKT system, and the constraints, which together may bring too much estimation errors. Specifically, the covariance propagation relies on a first-order approximation of the Jacobian, thus the UQ under strong nonlinearity constraint can be misestimated. Also the linearized KKT system is only locally valid and can fail to converge or even diverge if the constraint is highly nonlinear or the initial point is far from feasible.

2. The closed-form sample-free CRPS loss depends on using tractable distribution families (e.g., Gaussian). For multimodal/complex posteriors, it still requires sampling or approximations, thus the advantage of computational efficiency is not guaranteed.

3. In some places, the experimental results are not consistent with the claims made in the paper.

**Questions:**

1. Is it possible to provide a synthetic analysis of the effect of the first-order approximations on the results, or compute the approximation errors? Have you considered other methods than the linearized KKT, such as Gauss-Newton augmented Lagrangian for nonlinear constraints? The experimental results are not strong enough to support the superior performance of the proposed method (details in the 2nd question below), so maybe relaxing some of the approximations would improve the peformance.

2. The paper overclaims its performance based on the experimental results in some places. When analyzing these results,  sometimes it is hard to relate the numbers in the tables to the conclusions stated in the paper. For example,
   - When addressing Q1, the paper stated "Specifically, when measured against two accuracy metrics across four PDE datasets in Table 2, our method with either oblique (ProbHardE2E-Ob) or orthogonal (ProbHardE2E-Or) projection consistently outperforms both ProbConserv…". However, in Table 2, when comparing the results on Heat, the MSEs for ProbHardE2E-Ob/ProbHardE2E-Or trained with CRPS are higher than ProbConserv trained with CRPS (i.e., 0.036 and 0.031 compared to 0.027); when comparing the results on PME, the MSEs for ProbHardE2E-Ob/ProbHardE2E-Or trained with CRPS are higher than the other methods trained with CRPS (i.e., 9.59 and 9.01 compared to 8.801, 8.187 and 7.945). These results conflict with your statement that your method "consistently outperforms" the other methods.
   - In Q3, the target is to compare performance using CRPS as training objective to that using NLL, but Table 3 does not provide such experiments or information. I do not see the improvement upon HierE2E addressed Q3, either.
   - In Section 4.2.1, the paper stated "We see an even larger MSE performance improvement of ≈ 15 − 17× when trained on CRPS, and CRPS performance improvement of ≈ 2.5× over the various baselines". This statement only holds true for the case when $m\in[2,3]$, but not for the other two cases.
   - In Figure 1(b), it seems to me that the uncertainty estimates from the three methods are roughly the same (maybe the blue region is a bit wider than the others; it is hard to tell the difference between the red and purple regions). In Figure 2\(c\), it seems that all the three curves perform roughly the same, none of them captured the ground truth curve.

3. Typos/suggested corrections in the manuscript:
   - In Algorithm 1, it would be more clear if you indicate what $\mathbf{Y}_\theta(\phi)$ and $\mathbf{Z}_\theta(\phi)$ represent (constrained/unconstrained random variable), even though you have explained them in the main text. In addition, I think at the end of step 7, "where $z_\theta(\phi)\sim\mathbf{Z}_\theta(\phi)$" should be moved to the end of step 8.
   - The paper frequently mentioned "Problem 7" while I think it would be better to use "Problem (7)" as it appears.
   - In line 322-323 on page 6, the classical statistical approaches should be listed separately instead of being listed under "(ii) HierE2E".
   - Be consistent with using "HierE2E" and "Hier-E2E" (the paper is using both).
   - In line 424, "approximating" should be "approximately".
   - In lines 1327-1332, $\nabla L_{\hat{u}}$ should be $\nabla_{\hat{u}}L$.
   - In line 1448, in the last vector, $-J^{(*)}$ should be $J^{*}$.
   - In Eq. (44), add definition of $P_I=I-A^{\top}(AA^{\top})A$ (as you mentioned $P_I$ later but it was undefined).

---

> ### Author Response · Authors · 2025-11-21
> **Author's Response Part (1/2)**
>
> We thank the reviewer for their thoughtful evaluation and for highlighting the core strengths of our work. We appreciate the recognition that “the main novelty of this paper is to propose an end-to-end enforcement of hard constraints” and that our approach “enables joint UQ and constraint satisfaction.” We are also grateful for the reviewer’s comments on the “substantial training speedups” from sampling-free CRPS and analytic projections, as well as the “generality” of our framework across linear, nonlinear, and convex constraints. Below, we will try address the your concerns.
>
>
> ### Linearization of KKT Conditions
>
> Yes, our method uses a linearization of the KKT conditions, similar to approaches in nonlinear optimization, e.g., Sequential Quadratic Programming (SQP) and Gauss-Newton [1]. We understand the concern of first-order approximation of the Jacobian and the estimation errors, and that they are only locally accurate. We emphasize that our method does not rely solely on linearization. The delta method is used **only during training**, where it provides closed-form gradients for location-scale families  and avoid sampling variance, which enables stable optimization and **significant speedups** (**3–5×** faster for linear constraints, as shown in Figure 1a). At inference, we do not linearize but instead solve a constrained optimization problem for each sample via a Newton solver, enforcing hard constraints in their exact (nonlinear) form, which is **crucial in strongly nonlinear regimes**.  Our use of the delta method is a deliberate model design decision that balances training speed with accuracy.
>
> To appease these concerns, we have added additional experiments in Table A below on PME with $m \in [2,3]$ that directly project the samples and avoid our linear approximation with the delta method during training. We see that ProbHardE2E is approximately **9-45x** faster for approximately the same accuracy. We see that while the accuracy of the sampling approach does increase with more samples $S$, the runtime increases linearly with $S$, and it is already significantly higher than that for our ProbHardE2E. This trade-off between computational efficiency and approximation accuracy is particularly important in high-dimensional spatio-temporal scientific ML problems, where training time is a bottleneck.
>
> ### Table A: Performance comparison on the PME with nonlinear conservation law constraint versus sampling-based baselines with 10 and 50 samples.
>
> | PME m ∈ [2,3] | Metric | ProbHardE2E | Sampling-10 | Sampling-50 |
> |---------------|--------|-------------|-------------|-------------|
> |               | MSE    | **5.04E-06**    | 1.35E-02    | 2.93E-05    |
> |               | CE     | 0           | 0           | 0           |
> |               | CRPS   | **8.648E-04**   | 5.01E-02    | 2.03E-03    |
> |               | Time   | **1.54 s**      | 14.17 s     | 70.65 s     |
>
> In terms of theoretical guarantees, Theorem 3.1, whose proof is in Appendix F, shows that the approximation error between the nonlinear transformation and its linearization converges to zero in probability [2]. This linearization facilitates efficient training via closed-form CRPS gradients.
>
>
> In practice, our DPPL can be used with any optimization method, including more advanced ones as a forward pass and generating backward pass via implicit differentiation. We specifically chose to implement the Gauss-Newton Method mainly due to our requirements in the framework, e.g., differentiability and batching support. Naive differentiation through optimization programs can result in substantially long training times [7]. We use implicit differentiation to write a custom backward pass, with a forward pass that could use any method to obtain the feasible point. We find that explicit batching and implicit differentiation for optimization methods with nonlinear equality constraints are only available in a limited and fragmented manner within the PyTorch ecosystem, or the tools themselves available for implementation within the ecosystem are not yet mature. (See Appendix C.2 for details). Methods, e.g., augmented Lagrangian and second-order (Hessian) approaches may have better convergence and are good directions for future work. In addition, while second-order methods, e.g., BFGS [1], which use a higher second-order Taylor expansions may improve fidelity, they require the Hessian computation, which significantly increases the computational and memory complexity. We discuss this in Appendix F, and have mentioned this in the conclusion as a direction of future work.

---

> ### Author Response · Authors · 2025-11-21
> **Author's Response Part (2/2)**
>
> ### Closed-form CRPS
>
> We thank the reviewer for acknowledging that the CRPS training loss can be generalized to a broad class of probabilistic models. Among continuous distributions, we experimented with Gaussian variables, which have closed-form CRPS.  We show in our experimental results on PDEs and time series that even when the underlying probabilistic distribution is unknown, our ProbHardE2E performs well and the Gaussian assumption is robust. Beyond tractable univariate distributions, as the reviewer pointed out, sample-based projection is agnostic to distributional form.  Thus, the framework can be extended to skewed, bounded, or heavy-tailed distributions using, e.g., normalizing flows or rejection sampling, albeit at additional computational cost during training. This represents an exciting direction of future work, where we can incorporate sampling or other distributions into our framework.
>
>
> ### Experimental Results Performance
>
> We did not intend to overclaim and have clarified the following points in the revised version:
>
> - We clarify in the revised version that we consistently outperform in our target metric of CRPS, which is the preferred metric for probabilistic forecasts [3] . Note our approach is not guaranteed to improve MSE and we are targeting the uncertainty quantification metrics. For the PDE datasets, we clarify that all existing baselines in their original papers were trained with NLL, including ProbConserv [4]. To make it comparable in our ablations and test the effect of CRPS vs. NLL in Q3, we also train it with CRPS, as described in Evaluation Paragraph of Section 4 (lines 331-333).  We see that these other methods have improved performance when trained using CRPS, but they are trained using NLL in their original form.
> - We use Table 3 to discuss the computational efficiencies in Q4, and do not use Table 3 to address the CRPS vs. NLL ablation in Q3, which is conducted in Table 2. We also do not perform this ablation in Table 3, primarily because it is well-established in the hierarchical time series community [6] that CRPS is an effective training loss (e.g., from HierE2E). We do not claim accuracy improvement over HierE2E (See Appendix D.1 for its connection to our method). In particular, we show that we can improve the computational performance of HierE2E by projecting directly on the parameter distributions rather than projecting on the samples, as done in HierE2E, during training, and in the linear constraint case, an exact closed form solution exists. This leads to the  **3-5x** improvement in training time in Figure 1a.
> - We have clarified in the revised version that these specific performance numbers are upper bounds and occur when $m \in [2,3]$. We still see improved performance compared to the baselines for the other powers of $m$.
> - We clarify the results in Figure 1b-1c.  In Figure 1b, our ProbHardE2E (shown in purple) is directly on top of the exact solution, and the variance is so small that it is difficult to visualize in the figure. The wide uncertainty intervals on the plot correspond to ProbConserv (red) and ProbHardInf (blue). We can see this also by the significantly smaller CRPS and MSE  in this case $m \in [3,4]$ in Table 4 compared to the baselines. The TVD constraint is used to reduce the Gibb's phenomenon at shocks, i.e., the artificial oscillations as done in numerical methods  for hyperbolic conservation laws [6]. As a result, we do see narrower uncertainty intervals and importantly a smaller probability of drawing non-physical negative samples compared to the other approaches. In addition, the maximum and minimum of the uncertainty interval is decreased with our TVD constraint, which reduces the magnitude of the oscillations, as designed.
>
> ### Minor Comments
>
> We appreciate the reviewers’ comment to help improve the notational clarity. We have updated the algorithm and corrected them in our revised manuscript with the changes shown in red.
>
> ### References
>
> [1] Nocedal, J. and Wright, S.J., Numerical Optimization, 2nd Edition, Springer-Verlag, Berlin, 2006.
>
> [2] van der Vaart, A. W., Asymptotic Statistics, Theorem 3.1. Cambridge University Press, 1998.
>
> [3] Gneiting, Tilmann, and Raftery, Adrian E.. "Strictly proper scoring rules, prediction, and estimation." Journal of the American statistical Association 102.477 (2007): 359-378.
>
> [4] Hansen, Derek, et al. Learning physical models that can respect conservation laws. ICML 2023.
>
> [5] Rangapuram, Syama Sundar, et al. "End-to-end learning of coherent probabilistic forecasts for hierarchical time series." International Conference on Machine Learning. PMLR, 2021.
>
> [6] LeVeque, Randall J. Finite volume methods for hyperbolic problems. Vol. 31. Cambridge university press, 2002.
>
> [7] Blondel, Mathieu, et al. "Efficient and modular implicit differentiation." Advances in neural information processing systems 35 (2022): 5230-5242.

---

> ### Comment · Reviewer_LXk5 · 2025-11-28
>
> Many thanks to the authors for their careful response. They have addressed my concerns. Therefore, I still maintain a positive score. A minor typo is that in the revised version, at around line 1463, in the last vector, $-J^{(\*)}$ should be $-J^{\*}$.

---

> > ### Author Response · Authors · 2025-12-01
> >
> > We are happy that we have addressed your concerns and that you maintain your positive score.  We have fixed that remaining typo in the revised version. Thank you again for your detailed review that helped improve the paper's clarity.

---

### Author Response · Authors · 2025-11-21
**Summary**

We appreciate the reviewers’ valuable feedback to help improve the clarity of our manuscript. We are glad that the reviewers found our work an important and novel contribution to the field and all voted for acceptance. To summarize:
 - **Reviewer LXk5** highlights the “novelty” of our “end-to-end enforcement of hard constraints”, and “generality” of the various constraint types supported by our framework.
 - **Reviewer 8rdZ** acknowledges that our “method is clearly formulated in a principled “predictor–corrector” view.”
- **Reviewer 971x** highlights that our work “​​introduces a novel differentiable probabilistic projection layer (DPPL) that can be integrated with a wide range of neural network architectures” and that our “proposed method demonstrates strong empirical performance.”
- **Reviewer MxEb** states that our work “is a well-written and valuable paper with a clear contribution, sound theoretical derivations, and strong empirical validation.”
We address the reviewers’ comments below, and have also updated the manuscript with the changes highlighted in red in the rebuttal version.

---

### Author Response · Authors · 2025-12-01
**Summary of our rebuttal**

Dear AC, PC, and SAC,

We understand that this is a challenging period, and that the new ICLR policies have significantly increased your workload. We would like to highlight that all four reviewers expressed a positive assessment of the submission both pre- and post-rebuttal, and we successfully further addressed all comments and concerns during the rebuttal. We are grateful for the thoughtful evaluations provided by all four reviewers, which have helped to improve the quality and thoroughness of the paper. We have added the changes to the revised manuscript in red. To help minimize the time you need to spend, we have summarized each reviewer's feedback along with our corresponding rebuttal below:
- **Reviewer LXk5** emphasizes the novelty of our end-to-end enforcement of hard constraints and the generality of the constraint families supported by our framework, and maintained a positive score of **6** after our clarifications.
- **Reviewer 8rdZ** highlights that our method is clearly formulated in a principled predictor–corrector view, and confirmed that our rebuttal fully addressed the core concerns, maintaining a positive evaluation of **6**.
- **Reviewer 971x** underscores that our work introduces a novel differentiable probabilistic projection layer (DPPL) that integrates seamlessly with diverse architectures, and that our approach shows strong empirical performance; they confirmed that all questions regarding computation and analysis were effectively addressed and retained their positive score of **8**.
- **Reviewer MxEb** highlighted our work as having a clear contribution, sound theoretical derivations, and strong empirical validation. They acknowledged that our responses fully resolved the minor issues around notation and problem formulation and increased their score from **6 to 8** after our rebuttal, describing the paper as "a strong and well-suited contribution for the venue."

Across reviewers, there is **unanimous agreement** that the paper presents a novel and valuable framework (ProbHardE2E), offers strong theoretical grounding, and demonstrates solid empirical results. We hope this summary is helpful, and we appreciate the committee's efforts under the unusual circumstances.

Best regards,

The Authors

---

### Public Comment · ~Mingchuang_Li1 · 2026-04-14

End-to-End Probabilistic Framework is similar to [DistPred in KDD 2025](https://dl.acm.org/doi/10.1145/3690624.3709286).

---

### Meta-Review · Area_Chair_5G2J · 2026-01-02

**Summary:**

This paper presents ProbHardE2E, a probabilistic forecasting framework that incorporates hard operational/physical constraints and provides uncertainty quantification. The methodology uses a novel differentiable probabilistic projection layer (DPPL) that can be combined with a wide range of neural network architectures. DPPL allows the model to learn the system in an end-to-end manner, compared to other approaches where constraints are satisfied either through a post-processing step or at inference. ProbHardE2E optimizes a strictly proper scoring rule, without making any distributional assumptions on the target, which enables it to obtain robust distributional estimates and it can incorporate a range of non-linear constraints. The authors apply ProbHardE2E in learning partial differential equations with uncertainty estimates and to probabilistic time-series forecasting, showcasing it as a broadly applicable general framework that connects these seemingly disparate domains. Most reviewers agree that this is a nice paper with several strong points, including novelty, substantial training speedups, and generality. The reviewers also indicate that the paper is clearly written, the proposed method reduces learning bias, and that the training process can be sample-free, offering efficiency advantages. They also state that the method demonstrates strong empirical performance. Moreover, there are detailed proofs for the theoretical claims. Weakness are minor, but include that the method relies on first-order approximations, the CRPS loss depends on using tractable distribution families, and that in some places, the experimental results are not consistent with the claims made in the paper. Finally, some reviewers indicate a margin for presentation and readability improvement. Overall, I think that this is a nice paper that may receive the attention of the community.

**Reviewer Concerns:**

The rebuttal resolved most of the concerns (minor) about the paper by all reviewers. Specifically, the rebuttal cited extra relevant papers for clarification, added experiments, and improved notation.

**Reviewer Scores:**

I do not think that the scores would have significantly changed. However, reviewer MxEb says that they increased the score from 6 to 8.

---

### Decision · Program_Chairs · 2026-01-26

Accept (Poster)